# Does momentum Change the Implicit Bias on separable data?

## Abstract

The momentum acceleration technique is widely adopted in many optimization algorithms. However, the theoretical understanding on how the momentum affects the generalization performance of the optimization algorithms is still unknown. In this paper, we answer this question through analyzing the implicit bias of momentum-based optimization. We prove that on the linear classification problem with separable data, both SGD with momentum and Adam (without stochasticity) converge to the $L_2$ max-margin solution for exponential-tailed loss, which is the same as vanilla gradient descent. That means, these optimizers with momentum acceleration still converge to a model with low complexity, which provides guarantees on their generalization. Technically, to overcome the difficulty brought by the error accumulation in analyzing the momentum, we construct new Lyapunov functions as a tool to analyze the gap between the model parameter and the max-margin solution.

## 1 Introduction

It is widely believed that the optimizers have implicit bias in terms of selecting output parameters among all the local minima on the landscape (Neyshabur et al., 2015; Keskar et al., 2017; Wilson et al., 2017). It is shown in the analysis of Adaboost that the *coordinate descent* would converge to the $L^1$ max-margin solution for the linear classification task with exponetial-tailed loss ((Schapire & Freund, 2013; Telgarsky, 2013)). Latter, Soudry et al. (2018) shows that *gradient descent* would converge to the $L^2$ max-margin solution under the same setting, which mirrors its good generalization property in practice. Since then, many efforts have been taken on analyzing the implicit bias of various local-search optimizers, including stochastic gradient descent (Nacson et al., 2019), steepest descent (Gunasekar et al., 2018a), AdaGrad (Qian & Qian, 2019) and optimizers for homogeneous neural networks (Lyu & Li, 2019; Ji & Telgarsky, 2020; Wang et al., 2021).

However, though the momentum acceleration technique is widely adopted in the optimization algorithms in both convex and non-convex learning tasks (Sutskever et al., 2013; Vaswani et al., 2017; Tan & Le, 2019), the understanding on how the momentum would affect generalization performance of the optimization algorithms is still unclear. A natural question is:

*Can we theoretically analyze the implicit bias of momentum-based optimizers?*

In this paper, we take the first step to analyze the convergence of momentum based optimizers and unveil their implicit bias. Specifically, we study the classification problem with linear model and exponential-tailed loss using Stochastic Gradient Descent with Momentum (SGDM) and Adam optimizers. We consider the optimizers with constant learning rate and constant momentum hyper-parameters, which are widely adopted in practice, e.g., the default setting in popular machine learning frameworks (Paszke et al., 2019) and in experiments (Xie et al., 2017). We note that Gradient Descent with Momentum (GDM) can be viewed as a special case of SGDM, and naturally share the properties for SGDM. Our main results are summarized in Theorem 1.

**Theorem 1** (informal). *With linear separable dataset $\boldsymbol{S}$, for SGDM and Adam (without stochasticity, abbreviated as w/s latter), the loss converges to $0$ with rate $\mathcal{O}(\frac{1}{t})$ where $t$ is the number of iterations, the parameter norm diverges to infinity, and direction of parameters converges to the direction of the $L^2$ max-margin solution.*

Theorem 1 states SGDM converges to the $L^2$ max-margin solution, which is the same as SGD, indicating that momentum does not affect the convergent direction. The good generalization behavior of the output parameters of SGDM is well validated as the margin of a classifier is positively correlated with its generalization error (Jiang et al., 2019). This is supported by existing experimental observations (c.f. Figure 1, (Soudry et al., 2018) and Figure 2, (Nacson et al., 2019)). Similar claims hold for Adam (w/s), which is also well supported by empirical results in Wang et al. (2021).

Our contributions are significant in terms of the following aspects:

- We establish the implicit bias of the momentum based optimizers, an open problem since the initial work Soudry et al. (2018). The momentum based optimizers are widely used in practice and our theoretical characterization deepens the understanding on their generalization property, which is important by its own.
- Technically, we propose a *new Lyapunov function* to analyze the convergence of SGDM, which helps to bound the sum of squared gradients along the training trajectory. Compared to the usual one, the new Lyapunov function depends on a middle variable of an alternative update rule of SGDM, which helps to capture the historical dependence in the momentum update. To our knowledge, such a technique has not been exploited ever before and can be of independent interest for convergence analysis of momentum-based optimizers. We then construct a new Lyapunov function to bound the difference of learned parameters and the scaled max-margin solution, which finally leads to the direction convergence. This Lyapunov function provides a direct way to establish the convergence to the desired direction.

**Organization of This Paper.** Section 2 collects further related works on the implicit bias of first order optimizers and convergence of momentum-based optimizers. Section 3 shows basic settings and assumptions which will be used throughout this paper. Section 4 studies the implicit bias of GDM as a warm up, while Section 5 and Section 6 explore respectively the implicit bias of SGDM and Adam (w/s). Discussions of these results are put in Section 7.

## 2 FURTHER RELATED WORKS

**Implicit Bias of First-order Optimization Methods.** Soudry et al. (2018) prove that gradient descent on linear classification problem with exponential-tailed loss converges to the direction of the max $L^2$ margin solution of the corresponding hard-margin Support Vector Machine. Nacson et al. (2019) extend the results in (Soudry et al., 2018) to the stochastic case, proving that the convergent direction of SGD is the same as GD almost surely. Qian & Qian (2019) go beyond the vanilla gradient descent methods and consider the AdaGrad optimizer instead. They prove that the convergent direction of AdaGrad has a dependency on the optimizing trajectory, which varies according to the initialization. Ji & Telgarsky (2021) propose a primal-dual analysis framework for the linear classification models, and prove a faster convergent rate of the margin by increasing the learning rate according to the loss. Based on (Ji & Telgarsky, 2021), (Ji et al., 2021) design another algorithm with an even faster convergent rate of margin by applying the Nesterov's Acceleration Method on the dual space. However, the corresponding form of the algorithm on the primal space is no longer a Nesterov's Acceleration Method nor GDM, which is significantly different from our settings.

On the other hand, there is another line of work trying to extend the result in the linear case to deep neural networks. Ji & Telgarsky (2018); Gunasekar et al. (2018b) study the deep linear network and Soudry et al. (2018) study the two-layer neural network with ReLU activation. Lyu & Li (2019) propose a framework to analyze the asymptotic direction of GD on homogeneous neural networks, proving that given there exists a time the network achieves $100\%$ training accuracy, GD will converge to some KKT point of the $L^2$ max-margin problem. Wang et al. (2021) extend the framework of Lyu & Li (2019) to adaptive optimizers, and prove RMSProp and Adam without momentum have the same convergent direction as GD, while AdaGrad doesn't. The results (Lyu & Li, 2019; Wang et al., 2021) indicate that results in the linear model can be extended to deep homogeneous neural networks, and suggest that the linear model is a proper start point to study the implicit bias. There are also works on the implicit bias of regression problems with bounded optimal points and interesting readers can refer to (Rosasco & Villa, 2015; Lin & Rosasco, 2017; Ali et al., 2020) etc. for details.

**Convergence of Momentum-Based Optimization Methods.** For convex optimization problems, the convergence rate of Nesterov's Acceleration Method has been proved in Nesterov (1983). In

contrast, although GDM (Polyak's Heavy-Ball Method) was proposed in (Polyak, 1964) prior to the Nesterov's Acceleration Method, the convergence of GDM on convex loss with Lipschitz gradient was not solved until Ghadimi et al. (2015) provides an ergodic convergent result for GDM, i.e., the convergent result for the running average of the iterates. However, the ergodic result is undesired under many learning scenarios, e.g., in classification tasks, the optimization algorithms usually output the parameters of the last step. To the best of our knowledge, there is only two works on the non-ergodic analysis of (S)GDM: Sun et al. (2019) and Tao et al. (2021). Sun et al. (2019) prove that if the training loss is coercive (the training loss goes to infinity whenever parameter norm goes to infinity), convex, and globally smooth, then, with constant momentum hyper-parameter, the training loss converges to minima with rate $\mathcal{O}(t^{-1})$. Tao et al. (2021) analyze a case when momentum coefficient increases to 1 and the gradient is bounded all over the parameter space, showing SGDM can improve the convergence rate of SGD by a factor $\log(t)$.

There are also works on the gradient norm convergence of SGDM under various settings (Yan et al., 2018; Yu et al., 2019; Liu et al., 2020) and on the investigation of momentum-based method from the view point of dynamics (Sarao Mannelli & Urbani, 2021). However, there is no existing work on the implicit bias of momentum-based optimizers for the classification problem, which is first analyzed by this paper.

## 3 PRELIMINARIES

In this paper, we focus on the linear model with exponential-tailed loss. We first derive the results for binary classification, then we show that the methodology can be easily extended to the multi-class classification problem.

**Problem setting.** The dataset used for training is defined as $\boldsymbol{S} = (\boldsymbol{x}_i, \boldsymbol{y}_i)_{i=1}^N$, where $\boldsymbol{x}_i \in \mathbb{R}^d$ is the $i$-th feature, and $\boldsymbol{y}_i \in \mathbb{R}$ is the $i$-th label ($i = 1, 2, \cdots, N$). We will use the linear model to fit the label: for any feature $\boldsymbol{x} \in \mathbb{R}^d$ and parameter $\boldsymbol{w} \in \mathbb{R}^d$, the prediction is given by $\langle \boldsymbol{w}, \boldsymbol{x} \rangle$.

For binary classification, given any data $\boldsymbol{z}_i = (\boldsymbol{x}_i, \boldsymbol{y}_i) \in \boldsymbol{S}$, the individual loss for parameter $\boldsymbol{w}$ is given as $\ell(\boldsymbol{y}_i \langle \boldsymbol{w}, \boldsymbol{x}_i \rangle)$. A common setting is to ensemble the feature and label together as $\tilde{\boldsymbol{x}}_i = \boldsymbol{y}_i \boldsymbol{x}_i$ (there is a mapping $\mathcal{T} : (\boldsymbol{x}, \boldsymbol{y}) \to \boldsymbol{y}\boldsymbol{x}$, and $\tilde{\boldsymbol{x}}_i$ is $\mathcal{T}((\boldsymbol{x}_i, \boldsymbol{y}_i))$). The individual loss can be rewritten as

$$\tilde{\ell}(\boldsymbol{w}, (\boldsymbol{x}_i, \boldsymbol{y}_i)) = \ell(\boldsymbol{y}_i \langle \boldsymbol{w}, \boldsymbol{x}_i \rangle) = \ell(\langle \boldsymbol{w}, \tilde{\boldsymbol{x}}_i \rangle).$$

The optimization target is defined as the averaged loss:

$$\mathcal{L}(\boldsymbol{w}) = \frac{\sum_{i=1}^N \tilde{\ell}(\boldsymbol{w}, (\boldsymbol{x}_i, \boldsymbol{y}_i))}{N}.$$

Without loss of generality, we consider the case with normalized data[1], that is, $\|\tilde{\boldsymbol{x}}_i\| \leq 1, \forall i \in [N]$.

**Optimizer.** Here we will introduce the update rules of SGDM and Adam (w/s). SGDM can be viewed as a stochastic version of GDM by randomly choosing a subset of the dataset to update. Specifically, the update rule of SGDM is given as

$$(SGDM): \ \boldsymbol{w}(t+1) - \boldsymbol{w}(t) = -\eta \nabla \mathcal{L}_{\boldsymbol{B}(t)}(\boldsymbol{w}(t)) + \beta(\boldsymbol{w}(t) - \boldsymbol{w}(t-1)), \ \forall t \geq 1, \qquad (1)$$

where $\boldsymbol{B}(t)$ is a subset of $\boldsymbol{S}$ with size $b$ which is sampled independently and uniformly with replacement, and $\mathcal{L}_{\boldsymbol{B}(t)}$ is defined as $\mathcal{L}_{\boldsymbol{B}(t)}(\boldsymbol{w}) = \frac{\sum_{\boldsymbol{z} \in \boldsymbol{B}(t)} \tilde{\ell}(\boldsymbol{w}, \boldsymbol{z})}{b}$. We also define $\mathcal{F}_t$ as the sub-sigma field such that $\{\boldsymbol{w}(t)\}_{t=1}^\infty$ is adapted with respect to Filtration $\{\mathcal{F}_t\}_{t=1}^\infty$.

---

[1]The proof can be naturally applied to unnormalized data by letting $\ell(x) = \ell(\max_{\tilde{\boldsymbol{x}} \in \mathcal{T}(\boldsymbol{S})} \|\tilde{\boldsymbol{x}}\| \cdot x)$ and $\tilde{\boldsymbol{x}}_i = \frac{\tilde{\boldsymbol{x}}_i}{\max_{\tilde{\boldsymbol{x}} \in \mathcal{T}(\boldsymbol{S})} \|\tilde{\boldsymbol{x}}\|}$.

The Adam (w/s) can be viewed as a variant of GDM in which the preconditioner is adopted, whose form is characterized as follows:

$$\boldsymbol{m}(0) = \boldsymbol{0}, \boldsymbol{m}(t) = \beta_1 \boldsymbol{m}(t-1) + (1-\beta_1)\nabla\mathcal{L}(\boldsymbol{w}(t)), \hat{\boldsymbol{m}}(t) = \frac{1}{1-\beta_1^t}\boldsymbol{m}(t), \ \forall t \geq 1,$$

$$\nu(0) = 0, \nu(t) = \beta_2\nu(t-1) + (1-\beta_2)\left(\nabla\mathcal{L}(\boldsymbol{w}(t))\right)^2, \hat{\nu}(t) = \frac{1}{1-\beta_2^t}\nu(t), \ \forall t \geq 1,$$

$$(Adam\ (w/s)):\ \boldsymbol{w}(t) = \boldsymbol{w}(t-1) - \eta\frac{\hat{\boldsymbol{m}}(t-1)}{\sqrt{\hat{\nu}(t-1) + \varepsilon\mathbb{1}_d}}, \ \forall t \geq 1, \tag{2}$$

where $\frac{1}{\sqrt{\hat{\nu}(t-1)+\varepsilon\mathbb{1}_d}})$ is called the preconditioner at step $t$.

**Assumptions:** The analysis of this paper are based on three common assumptions in existing literature (first proposed by (Soudry et al., 2018)), respectively on the separability of the dataset, the individual loss behaviour at the tail , and the smoothness of the individual loss. We list them as follows:

**Assumption 1** (Linearly Separable Dataset). *There exists one parameter $\boldsymbol{w} \in \mathbb{R}^d$, such that*

$$\langle \boldsymbol{w}, \tilde{\boldsymbol{x}}_i \rangle > 0, \ \forall i \in [N].$$

**Assumption 2** (Exponential-tailed Loss). *The individual loss $\ell$ is exponential-tailed, i.e.,*

- *Differentiable and monotonically decreasing to zero, with its derivative also converging to zero, i.e., $\lim_{x\to\infty} \ell(x) = \lim_{x\to\infty} \ell'(x) = 0$, and $\ell'(x)<0$ $\forall x$;*

- *Close to exponential loss when $x$ is large enough, i.e., there exist positive constants $c, a, K, \mu_+, \mu_-, x_+, x_-$ and $x_0$, such that,*

$$\forall x > x_+ : -\ell'(x) \leq c(1 + e^{-\mu_+ x})e^{-ax}, \tag{3}$$

$$\forall x > x_- : -\ell'(x) \geq c(1 - e^{-\mu_- x})e^{-ax}. \tag{4}$$

**Assumption 3** (Smooth Loss). *Either of the following assumptions holds regarding the case:*

*(D): (Deterministic Case) The individual loss $\ell$ is locally smooth, i.e., for any $s_0 \in \mathbb{R}$, there exists a positive real $H_{s_0}$, such that $\forall x, y \geq s_0, |\ell'(x) - \ell'(y)| \leq H_{s_0}|x - y|$.*

*(S): (Stochastic Case) The individual loss $\ell$ is globally smooth, i.e., there exists a positive real $H$, such that $\forall x, y \in \mathbb{R}, |\ell'(x) - \ell'(y)| \leq H|x - y|$.*

We provide explanations of these three assumptions respectively. Based on Assumption 1, we can formally define the margin and the maximum margin solution of an optimization problem, which is introduced in Definition 1

**Definition 1.** *Let the margin $\gamma(\boldsymbol{w})$ of parameter $\boldsymbol{w}$ defined as the lowest score of the prediction of $\boldsymbol{w}$ over the dataset $S$, i.e., $\gamma(\boldsymbol{w}) = \min_{\tilde{\boldsymbol{x}}_i \in \mathcal{T}(S)}\langle \boldsymbol{w}, \tilde{\boldsymbol{x}}_i \rangle$. By Assumption 1 and the positive homogeneous of $\gamma$, $\gamma(\frac{\hat{\boldsymbol{w}}}{\gamma(\hat{\boldsymbol{w}})}) = 1$, and thus we define the maximum margin solution $\hat{\boldsymbol{w}}$ and the $L^2$ max margin $\gamma$ of the dataset $S$ as follows:*

$$\hat{\boldsymbol{w}} \overset{\triangle}{=} \arg \min_{\gamma(\boldsymbol{w}) \geq 1} \|\boldsymbol{w}\|^2, \ \gamma \overset{\triangle}{=} \frac{1}{\|\hat{\boldsymbol{w}}\|}$$

Since $\|\cdot\|^2$ is strongly convex and set $\{\boldsymbol{w} : \gamma(\boldsymbol{w}) \geq 1\}$ is convex, $\hat{\boldsymbol{w}}$ is uniquely defined.

Assumption 2 constraints the loss to be exponential-tailed, which is satisfied by many popular choices of $\ell$, including the ($\ell_{exp}(x) = e^{-x}$) and the logistic loss ($\ell_{log}(x) = \log(1 + e^{-x})$). Also, as $c$ and $a$ can be respectively absorbed by resetting the learning rate and data as $\eta = c\eta$ and $\boldsymbol{x}_i = a\boldsymbol{x}_i$, without loss of generality, in this paper we only analyze the case that $c = a = 1$.

The globally smooth assumption (Assumption 3. (S)) is strictly stronger than the locally smooth assumption (Assumption 3. (D)). One can easily verify that both the exponential loss and the logistic loss meet Assumption 3. (**D**), and the logistic loss also meets Assumption 3. (**S**).

## 4   WARM UP: IMPLICIT BIAS OF GDM

In this section, we study the implicit bias of GDM as a warm up. The update rule of GDM is

$$\boldsymbol{w}(t+1) - \boldsymbol{w}(t) = -\eta \nabla \mathcal{L}(\boldsymbol{w}(t)) + \beta(\boldsymbol{w}(t) - \boldsymbol{w}(t-1)), \ \forall t \geq 1 \tag{5}$$

which is a determined case of SGDM, with batch size $b = N$ and thus without randomness. Although the analysis of GDM is essentially easier than that of SGDM, it helps us gain a better understanding of this problem, and helps us to demonstrate several techniques which will be applied latter. The formal theorem of the implicit bias of GDM is as follows:

**Theorem 2.** *Let Assumptions 1, 2, and 3. (**D**) hold. Let $\beta \in [0,1)$ and $\eta < 2\frac{1-\beta}{H_{\ell^{-1}(N\mathcal{L}(\boldsymbol{w}_1))}}$. Then, for almost every data set $S^2$, with arbitrary initialization point $\boldsymbol{w}(1)$, GDM (Eq. (5)) satisfies that $\boldsymbol{w}(t) - \log(t)\hat{\boldsymbol{w}}$ is bounded as $t \to \infty$, and $\lim_{t\to\infty} \frac{\boldsymbol{w}(t)}{\|\boldsymbol{w}(t)\|} = \frac{\hat{\boldsymbol{w}}}{\|\hat{\boldsymbol{w}}\|}$.*

Before we move on to the proof, we would like to provide some explanations of the results in Theorem 2. To begin with, Theorem 2 adopts an constant momentum hyper-parameter, which agrees with the practice use (e.g., $\beta$ is set to be $0.9$ (Xie et al., 2017)). Also, Theorem 2 puts no restriction on the range of $\beta$, which allows wider choices of hyper-parameter tuning. Furthermore, Theorem 2 shows the implicit bias of GDM agrees with GD in linear classification with exponential-tailed loss (c.f. Soudry et al. (2018) for results on GD), and this consistency can be verified by existing results (c.f. Section 7 for detailed discussions).

We then present a proof sketch of Theorem 2, which is divided into two parts: we first prove that the sum of squared gradients is bounded, which indicates both the loss and the norm of gradient converge to 0; we then show the difference between $\boldsymbol{w}(t)$ and $\log(t)\hat{\boldsymbol{w}}$ is bounded, and therefore, the direction of $\hat{\boldsymbol{w}}$ dominates as $t \to \infty$.

**Stage I: Bound the sum of squared gradients.** The core of Stage I is to select a proper Lyapunov function $\xi(t)$, which is required to correlated with the training loss $\mathcal{L}$ and be non-increasing along the optimization trajectory. For GD, since $\mathcal{L}$ itself is non-increasing with properly chosen learning rate, we can just pick $\xi(t) = \mathcal{L}(t)$. However, as the update of GDM doesn't align with the direction of negative gradient, training loss $\mathcal{L}(t)$ in GDM is no longer monotonously decreasing, and the Lyapunov function requires special construction. A choice of $\xi(t)$ is proposed as the follows:

**Lemma 1.** *Let all conditions in Theorem 2 hold. Define $\xi(t) \triangleq \mathcal{L}(\boldsymbol{w}(t)) + \frac{\beta}{2\eta}\|\boldsymbol{w}(t) - \boldsymbol{w}(t-1)\|^2$. Let $C_1$ be a positive real, s.t., $\eta = 2\frac{1-\beta}{H_{\ell^{-1}(N\mathcal{L}(\boldsymbol{w}_1))}}C_1$. We then have*

$$\xi(t) \geq \xi(t+1) + \frac{(1-\beta)(1-C_1)}{\eta}\|\boldsymbol{w}(t+1) - \boldsymbol{w}(t)\|^2. \tag{6}$$

**Remark 1.** *Although this Lyapunov function is obtained by (Sun et al., 2019) by directly examining the Taylor's expansion at $\boldsymbol{w}(t)$, the proof here is non-trivial as we only requires the loss to be locally smooth instead of globally smooth in (Sun et al., 2019), and the Taylor's expansion can only be applied to $\boldsymbol{w}(t+1)$ if it's ensured that all parameters on the line $\alpha\boldsymbol{w}(t) + (1-\alpha)\boldsymbol{w}(t+1)$ ($\alpha \in (0,1)$) have training loss no larger than $\mathcal{L}(\boldsymbol{w}(0))$.*

To prove Lemma 1, we define $\boldsymbol{w}(t+\alpha) = \alpha\boldsymbol{w}(t) + (1-\alpha)\boldsymbol{w}(t+1)$ for any $t \in \mathbb{Z}^+$ and $\alpha \in (0,1)$, and prove a generalized version of Eq. (6):

$$\xi(t) \geq \xi(t+\alpha) + \frac{(1-\beta)(1-c)\alpha^2}{\eta}\|\boldsymbol{w}(t+1) - \boldsymbol{w}(t)\|^2. \tag{7}$$

The proof idea is that as long as Eq. (7) holds for all time in $[1, t+\alpha)$, the training loss across $[1, t+\alpha]$ will be smaller than $\mathcal{L}(\boldsymbol{w}(1))$, and the strict inequality in Eq. (7) holds. Consequently, there exists some small enough positive real $\varepsilon$, such that Eq. (7) holds for all time in $[1, t+\alpha+\varepsilon)$, and we are able to extend the feasible set where Eq. (7) holds. The formal description of the generalized lemma and its corresponding proof is deferred to Appendix B.1.1.

---

[2]Here "almost everywhere" means the conclusion holds except a zero-measure set in $\mathbb{R}^{d \times N}$.

By Lemma 1, we have that $\xi(t)$ is monotonously decreasing by gap $\frac{(1-\beta)(1-c)\alpha^2}{\eta}\|\boldsymbol{w}(t+1)-\boldsymbol{w}(t)\|^2$. As $\xi(1)=\mathcal{L}(\boldsymbol{w}(1))$ is a finite number, we have $\sum_{t=1}^{\infty}\|\boldsymbol{w}(t+1)-\boldsymbol{w}(t)\|^2<\infty$, which further leads to the following corollary by the negative derivative of the individual loss and separable data:

**Corollary 1.** *Let all conditions in Theorem 2 hold. We have,* $\sum_{t=1}^{\infty}\|\nabla\mathcal{L}(\boldsymbol{w}(t))\|^2<\infty$.

The proof of Corollary 1 can be found in Appendix B.1.1. Resulted from Corollary 1, we have $\|\nabla\mathcal{L}(\boldsymbol{w}(t))\|^2\to 0$ as $t\to\infty$, which by the exponential-tailed loss assumption further leads to $\lim_{t\to\infty}\mathcal{L}(\boldsymbol{w}(t))=0$.

**Stage II. Bound the difference between $\boldsymbol{w}(t)$ and $\log(t)\hat{\boldsymbol{w}}$.** It seems natural to directly bound $\|\boldsymbol{w}(t)-\log(t)\hat{\boldsymbol{w}}\|$ across all iterations. Soudry et al. (2018) and Nacson et al. (2019) indeed follow this routine by showing the norm of $\boldsymbol{r}(t)\triangleq\boldsymbol{w}(t)-\log(t)\hat{\boldsymbol{w}}-\tilde{\boldsymbol{w}}$ is bounded, where $\tilde{\boldsymbol{w}}$ is some constant vector satisfying $e^{\langle\tilde{\boldsymbol{w}},\tilde{\boldsymbol{x}}\rangle}$ recovers the coefficient of support vector $\tilde{\boldsymbol{x}}$ in $\hat{\boldsymbol{w}}$. However, their analyses rely on the fact that $\boldsymbol{w}(t+1)-\boldsymbol{w}(t)$ equals $-\eta\nabla\mathcal{L}(\boldsymbol{w}(t))$, which has a simple form, i.e.,

$$For\ GD: \|\boldsymbol{r}(t+1)\|^2-\|\boldsymbol{r}(t)\|^2=\|\boldsymbol{r}(t+1)-\boldsymbol{r}(t)\|^2-2\langle\log\frac{t+1}{t}\hat{\boldsymbol{w}}+\eta\nabla\mathcal{L}(\boldsymbol{w}(t)),\boldsymbol{r}(t)\rangle.$$

However, it no longer holds for GDM, as $\boldsymbol{w}(t+1)-\boldsymbol{w}(t)$ is a exponentially-decayed sum of gradients at $\boldsymbol{w}(s),\ s\le t$. To handle this problem, we propose a novel Lyapunov's function for the direction convergence, concluded as the following Lemma.

**Lemma 2.** *Let all conditions in Theorem 2 hold. Then, $\|\boldsymbol{r}(t)\|$ is bounded if and only if the function $g(t)$ is upper bounded, where $g:\mathbb{Z}^+\to\mathbb{R}$ is defined as*

$$g(t)\triangleq\frac{1}{2}\|\boldsymbol{r}(t)\|^2+\frac{\beta}{1-\beta}\langle\boldsymbol{r}(t),\boldsymbol{w}(t)-\boldsymbol{w}(t-1)\rangle-\frac{\beta}{1-\beta}\sum_{\tau=2}^{t}\langle\boldsymbol{r}(\tau)-\boldsymbol{r}(\tau-1),\boldsymbol{w}(\tau)-\boldsymbol{w}(\tau-1)\rangle. \quad (8)$$

*Furthermore, we have $\sum_{t=1}^{\infty}(g(t+1)-g(t))$ is upper bounded.*

The first claim in Lemma 2 provides an alternative approach to verify that $\|\boldsymbol{r}(t)\|$ is bounded, while the second claim shows this approach can be fulfilled. The motivation of this lemma is to construct a easy-to-verify criterion (i.e., $g(t)$ is upper bounded) of $\|\boldsymbol{r}(t)\|$ is upper bounded. To demonstrate how $g(t)$ is constructed intuitively, we consider the continuous dynamics approximation of GDM, i.e.,

$$\frac{\beta}{1-\beta}\frac{\mathrm{d}^2\boldsymbol{w}(t)}{\mathrm{d}t^2}+\frac{\mathrm{d}\boldsymbol{w}(t)}{\mathrm{d}t}+\frac{\eta}{1-\beta}\nabla\mathcal{L}(\boldsymbol{w}(t))=0.$$

We then calculate the $\frac{1}{2}\|\boldsymbol{r}(t)\|^2-\frac{1}{2}\|\boldsymbol{r}(1)\|^2$ by applying the integral of the dynamics as

$$\int_1^t\frac{1}{2}\frac{\mathrm{d}\|\boldsymbol{r}(s)\|^2}{\mathrm{d}s}\mathrm{d}s=\int_1^t\left\langle\boldsymbol{r}(s),-\frac{\eta}{1-\beta}\nabla\mathcal{L}(\boldsymbol{w}(s))-\frac{1}{s}\hat{\boldsymbol{w}}\right\rangle\mathrm{d}s+\int_1^t\frac{\beta}{1-\beta}\left\langle\boldsymbol{r}(s),-\frac{\mathrm{d}^2\boldsymbol{w}(s)}{\mathrm{d}s^2}\right\rangle\mathrm{d}s$$

$$=\int_1^t\left\langle\boldsymbol{r}(s),-\frac{\eta}{1-\beta}\nabla\mathcal{L}(\boldsymbol{w}(s))-\frac{1}{s}\hat{\boldsymbol{w}}\right\rangle\mathrm{d}s+\frac{\beta}{1-\beta}\left(\left\langle\boldsymbol{r}(t),-\frac{\mathrm{d}\boldsymbol{w}(t)}{\mathrm{d}t}\right\rangle-\left\langle\boldsymbol{r}(1),-\frac{\mathrm{d}\boldsymbol{w}(t)}{\mathrm{d}t}\Big|_{t=1}\right\rangle+\int_1^t\left\langle\frac{\mathrm{d}\boldsymbol{r}(s)}{\mathrm{d}s},\frac{\mathrm{d}\boldsymbol{w}(s)}{\mathrm{d}s}\right\rangle\mathrm{d}s\right),$$

where the last equation is due to Integration by part. We denote $\tilde{g}(t)=\frac{\beta}{1-\beta}(\langle\boldsymbol{r}(t),\frac{\mathrm{d}\boldsymbol{w}(t)}{\mathrm{d}t}\rangle-\int_1^t\langle\frac{\mathrm{d}\boldsymbol{r}(s)}{\mathrm{d}s},\frac{\mathrm{d}\boldsymbol{w}(s)}{\mathrm{d}s}\rangle\mathrm{d}s)+\frac{1}{2}\|\boldsymbol{r}(t)\|^2$, and it can be verified the derivative of $\tilde{g}(t)$ is $\langle\boldsymbol{r}(t),-\frac{\eta}{1-\beta}\nabla\mathcal{L}(\boldsymbol{w}(t))-\frac{1}{t}\hat{\boldsymbol{w}}\rangle$. By replacing $\frac{\mathrm{d}\boldsymbol{w}(t)}{\mathrm{d}t}$ as $\boldsymbol{w}(t)-\boldsymbol{w}(t-1)$, and $\frac{\mathrm{d}\boldsymbol{r}(t)}{\mathrm{d}t}$ as $\boldsymbol{r}(t)-\boldsymbol{r}(t-1)$ in $\tilde{g}(t)$, we obtain $g(t)$, and $g(t+1)-g(t)$ has the form $\langle\boldsymbol{r}(t),-\frac{\eta}{1-\beta}\nabla\mathcal{L}(\boldsymbol{w}(t))-\log\frac{t+1}{t}\hat{\boldsymbol{w}}\rangle$, which can be analyzed following the similar routine as (Soudry et al., 2018). The proof of the second claim is completed.

On the other hand, the core of the proof of Lemma 2 is that $\frac{1}{2}\|\boldsymbol{r}(t)\|^2+\frac{\beta}{1-\beta}\langle\boldsymbol{r}(t),\boldsymbol{w}(t)-\boldsymbol{w}(t-1)\rangle$ is bounded if and only if $\frac{1}{2}\|\boldsymbol{r}(t)\|^2$ is bounded, while $\langle\boldsymbol{r}(\tau)-\boldsymbol{r}(\tau-1),\boldsymbol{w}(\tau)-\boldsymbol{w}(\tau-1)\rangle$ is an absolute convergent based on $\sum_{t=1}^{\infty}\|\boldsymbol{w}(t+1)-\boldsymbol{w}(t)\|^2<\infty$.

## 5 TACKLE THE DIFFICULTY BROUGHT BY RANDOM SAMPLING

In this section, we analyze the implicit bias of SGDM. The randomness introduced by random sampling makes the analysis of GDM no longer work for SGDM. As an example, if we want to

follow the same routine of the GDM case to show $\mathcal{L}(\boldsymbol{w}(t)) + \frac{\beta}{2\eta}\mathbb{E}\|\boldsymbol{w}(t) - \boldsymbol{w}(t-1)\|^2$ is a Lyapunov function of SGDM, we can only have

$$\mathbb{E}\mathcal{L}(\boldsymbol{w}(t)) + \frac{\beta}{2\eta}\mathbb{E}\|\boldsymbol{w}(t) - \boldsymbol{w}(t-1)\|^2 \geq \mathbb{E}\mathcal{L}(\boldsymbol{w}(t+1)) + \frac{\beta}{2\eta}\mathbb{E}\|\mathbb{E}\left[\boldsymbol{w}(t+1) - \boldsymbol{w}(t)|\mathcal{F}_t\right]\|^2$$
$$+ \frac{(1-\beta)(1-c)}{\eta}\mathbb{E}\|\mathbb{E}\left[\boldsymbol{w}(t+1) - \boldsymbol{w}(t)|\mathcal{F}_t\right]\|^2.$$

As the squared first momentum is smaller than the second momentum, $\mathbb{E}\mathcal{L}(\boldsymbol{w}(t)) + \frac{\beta}{2\eta}\mathbb{E}\|\boldsymbol{w}(t) - \boldsymbol{w}(t-1)\|^2$ may no longer be monotonously decreasing for every $\beta \in [0,1)$. Furthermore, if the margin $\gamma$ is small, $\beta$ will be required to be upper bounded by a small number, which contradicts to the relatively large choice of $\beta$ in practice, e.g. $0.9$ (We defer a detailed discussion to Appendix B.2.3). To tackle this problem, we propose a new Lyapunov function, with which we derive the following theorem for the implicit bias of SGDM:

**Theorem 3.** *Let Assumption 1, 2, and 3. (S) hold. Let $\beta \in [0,1)$ and $\eta < \min\{\frac{1-\beta}{1+\frac{HN}{b\gamma^2}}, \frac{(1-\beta)^3\gamma^4 b}{2H^2 N^3 \beta^2}\}$. Then, for almost every data set $S$, with arbitrary initialization point $\boldsymbol{w}(1)$, SGDM (Eq. (1)) satisfies $\boldsymbol{w}(t) - \log(t)\hat{\boldsymbol{w}}$ is bounded as $t \to \infty$ and $\lim_{t\to\infty} \frac{\boldsymbol{w}(t)}{\|\boldsymbol{w}(t)\|} = \frac{\hat{\boldsymbol{w}}}{\|\hat{\boldsymbol{w}}\|}$, almost surely (a.s.).*

To the best of our knowledge, this is the first convergence analysis of SGDM with no restriction on the gradient norm and with constant learning rates. Also, as both $\frac{1-\beta}{1+\frac{HN}{b\gamma^2}}$ and $\frac{(1-\beta)^3\gamma^4 b}{2H^2 N^3}$ are monotonously increasing with respect to batch size $b$, Theorem 3 also sheds light on the learning rate tuning, i.e., the larger the batch size is, the larger the learning rate is. Furthermore, similar to the GDM case, Theorem 3 shows the implicit bias of SGDM under this setting is consistent with SGD (c.f. (Nacson et al., 2019) for the implicit bias of SGD). This matches the observations in practice (c.f. Section 7 for details).

**Remark 2.** *In practice, mini-batch SGDM are more widely adopted, whose update rule differs from SGDM only by the way obtaining $\boldsymbol{B}(t)$. Specifically, within the $T$-th epoch $E(T) = \{KT + 1, \cdots, KT+T\}$ ($K$ is the length of one epoch), $\{\boldsymbol{B}(t)\}_{t \in E(T)}$ is a randomly and uniformly partition of $\boldsymbol{S}$. One may wonder whether the same result hold for mini-batch SGDM. The answer is "Yes", with the a.s. condition removed in this case. We defer the detailed description of the corresponding theorem together with the proof to Appendix D.1.*

The proof sketch follows the same framework as that for GDM by dividing the proof into two stages. However, the implementations in both stages differ, among which the proof of Stage I is significantly distinctive, as a new Lyapunov function is proposed.

**Stage I: Bound the sum of squared gradients.** To obtain the new Lyapunov function, we first present a lemma to provide an alternative form of update rule of SGDM:

**Lemma 3.** *Define $\tilde{\eta} \triangleq \frac{\eta}{1-\beta}$, and $\boldsymbol{u}(1) \triangleq \boldsymbol{w}(1)$. The update rule of SGDM (Eq. (1)) is equivalent to*

$$\boldsymbol{u}(t+1) = -\tilde{\eta}\nabla\mathcal{L}_{\boldsymbol{B}(t)}(\boldsymbol{w}(t)) + \boldsymbol{u}(t),$$
$$\boldsymbol{w}(t+1) = \beta\boldsymbol{w}(t) + (1-\beta)\boldsymbol{u}(t+1). \tag{9}$$

By a simple rearrangement of the second equation, we have $\boldsymbol{u}(t+1) = \boldsymbol{w}(t) + \frac{1}{1-\beta}(\boldsymbol{w}(t+1) - \boldsymbol{w}(t))$, which differs from $\boldsymbol{w}(t+1)$ only by a larger step size updated from $\boldsymbol{w}(t)$. The following lemma then indicates that $\mathbb{E}\mathcal{L}(\boldsymbol{u}(t))$ is a proper selection of Lyapunov function.

**Lemma 4.** *Let all conditions in Theorem 3 hold. Then, we have*

$$\mathbb{E}[\mathcal{L}(\boldsymbol{u}(t+1))] \leq \mathbb{E}\mathcal{L}(\boldsymbol{u}(t)) - \frac{\tilde{\eta}}{2}\mathbb{E}\|\nabla\mathcal{L}(\boldsymbol{w}(t))\|^2 + \frac{1-\beta}{4}\tilde{\eta}\left(\sum_{s=1}^{t-1}\beta^{t-1-s}\mathbb{E}\|\nabla\mathcal{L}(\boldsymbol{w}(s))\|^2\right),$$

*and*

$$\mathbb{E}[\mathcal{L}(\boldsymbol{u}(t+1))] \leq \mathcal{L}(\boldsymbol{u}(1)) - \sum_{s=1}^{t}\frac{\tilde{\eta}}{4}\mathbb{E}\|\nabla\mathcal{L}(\boldsymbol{w}(s))\|^2.$$

$\mathbb{E}\mathcal{L}(\boldsymbol{u}(t))$ may be not monotonously decreasing. However, the upper bound of $\mathbb{E}\mathcal{L}(\boldsymbol{u}(t))$, i.e., $\mathcal{L}(\boldsymbol{u}(1)) - \sum_{s=1}^{t} \frac{\tilde{\eta}}{4} \, ` \mathbb{E}\|\nabla\mathcal{L}(\boldsymbol{w}(s))\|^2$, is monotonously decreasing by $\frac{\tilde{\eta}}{4}\mathbb{E}\|\nabla\mathcal{L}(\boldsymbol{w}(t))\|^2$ at step $t$, and leads to the sum of expected squared gradients being finite along the trajectory (which further indicates the sum of squared gradients is finite, a.s.).

The proof idea of Lemma 4 is the expectation of the first order Taylor's expansion of $\mathcal{L}$ at $\boldsymbol{w}(t)$ has the form

$$\mathbb{E}[\mathcal{L}(\boldsymbol{u}(t+1))|\mathcal{F}_t] \leq \mathcal{L}(\boldsymbol{u}(t)) - \tilde{\eta}\langle\nabla\mathcal{L}(\boldsymbol{u}(t)), \nabla\mathcal{L}(\boldsymbol{w}(t))\rangle + \frac{L\tilde{\eta}^2}{2}\mathbb{E}[\|\nabla\mathcal{L}_{\boldsymbol{B}(t)}(\boldsymbol{w}(t))\|^2|\mathcal{F}_t].$$

As $\mathcal{L}$ is $H$ smooth, we have $\nabla\mathcal{L}(\boldsymbol{u}(t)) = \nabla\mathcal{L}(\boldsymbol{w}(t)) + \mathcal{O}(\|\boldsymbol{u}(t) - \boldsymbol{w}(t)\|)$, where $\boldsymbol{u}(t) - \boldsymbol{w}(t) = \frac{\beta}{1-\beta}(\boldsymbol{w}(t) - \boldsymbol{w}(t-1)) = -\eta\frac{\beta}{1-\beta}\sum_{s=1}^{t-1}\beta^{t-1-s}\nabla\mathcal{L}_{\boldsymbol{B}(s)}(\boldsymbol{w}(s))$ is proportional to $\eta$. Therefore, when $\eta$ is small, $-\langle\nabla\mathcal{L}(\boldsymbol{u}(t)), \nabla\mathcal{L}(\boldsymbol{w}(t))\rangle$ is close to $-\|\nabla\mathcal{L}(\boldsymbol{w}(t))\|^2$, and the coefficient of $-\|\nabla\mathcal{L}(\boldsymbol{w}(t))\|^2$ becomes dominant.

**Stage II. Bound the difference between $\boldsymbol{w}(t)$ and $\log(t)\hat{\boldsymbol{w}}$.** The Lyapunov function used by Stage II of SGDM is the same as the $g(t)$ in the GDM case. As we have proved the sum of squared gradient along the trajectory is bounded, one can easily obtain that $\sum_{t=1}^{\infty}\|\boldsymbol{w}(t+1) - \boldsymbol{w}(t)\|^2$ is bounded, and $\|\boldsymbol{r}(t)\|$ being bounded is still equivalent to that $g(t)$ is upper bounded. However, when it comes to the detailed calculation of analyzing $g(t+1) - g(t)$, it differs from the GDM case due to the random subset. We defer the proof to Appendix B.2.2.

## 6 ANALYZE THE EFFECT OF PRECONDITIONERS

When it comes to the analysis of Adam (w/s), the effect of the preconditioner should be taken into consideration when designing the corresponding Lyapunov functions. To tackle this problem, we incorporate the preconditioner into the Lyapunov function, and obtain the implicit bias of Adam (w/s) as follows:

**Theorem 4.** *Let Assumption 1, 2, and 3. (D) hold. Let $1 > \beta_2 > \beta_1^4 \geq 0$, and the learning rate $\eta$ is a small enough constant (The upper bound of learning rate is complex, and we defer it to Appendix C.1). Then, for almost every data set $S$, with arbitrary initialization point $\boldsymbol{w}(1)$, Adam (w/s) (Eq. (2)) satisfies that $\boldsymbol{w}(t) - \log(t)\hat{\boldsymbol{w}}$ is bounded as $t \to \infty$, and $\lim_{t\to\infty} \frac{\boldsymbol{w}(t)}{\|\boldsymbol{w}(t)\|} = \frac{\hat{\boldsymbol{w}}}{\|\hat{\boldsymbol{w}}\|}$.*

**Remark 3.** *Existing literature usually assume a time-decaying hyperparameter choice of $\beta_1$ or $\beta_2$ (c.f., (Kingma & Ba, 2014; Chen et al., 2018)). To the best of our knowledge, the only work analyzing Adam with constant $\beta_1$ and $\beta_2$ is (Reddi et al., 2019), which assumes $\beta_2 > \beta_1^2$ (stronger than our assumption). Our result indicates that the ranges of $\beta_1$ and $\beta_2$ can be broader in hyper-parameter selection.*

We still start the proof sketch by proving the sum of squared gradients is finite in Stage I. Compared to GDM, the difference is that in Stage I, we will also prove the loss converges to $0$ with rate $\Theta(t^{-1})$. This property will be used in Stage II to analyze the effect of preconditioner on implicit bias.

**Stage I: Bound the sum of squared gradients.** The next lemma can be viewed as an extension of Lemma 1 by taking the preconditioner into consideration, and characterizes the one-step loss update in Adam (w/s).

**Lemma 5.** *Let all conditions in Theorem 4 hold. Then, for any $t \geq 1$,*

$$\mathcal{L}(t+1) + \frac{1}{2}\frac{1-\beta_1^t}{\eta(1-\beta_1)}\left\|\sqrt[4]{\varepsilon\mathbb{1}_d + \hat{\nu}(t)} \odot (\boldsymbol{w}(t+1) - \boldsymbol{w}(t))\right\|^2$$

$$\leq \mathcal{L}(\boldsymbol{w}(t)) + \frac{1-\beta_1^{t-1}}{2c\eta(1-\beta_1)}\frac{1-(c\beta_1)^t}{1-(c\beta_1)^{t-1}}\left\|\sqrt[4]{\varepsilon\mathbb{1}_d + \hat{\nu}(t-1)} \odot (\boldsymbol{w}(t) - \boldsymbol{w}(t-1))\right\|^2. \quad (10)$$

The difference between the proof of Lemma 5 and Lemma 1 is that we need to handle the gap between the preconditioners at step $t$ and step $t+1$, this leads to a amplifying factor $\sqrt[4]{\frac{1-\beta_2^t}{\beta_2(1-\beta_2^{t-1})}}$ of term $\left\|\sqrt[4]{\varepsilon\mathbb{1}_d + \hat{\nu}(t-1)} \odot (\boldsymbol{w}(t) - \boldsymbol{w}(t-1))\right\|^2$, which, however, is smaller than the shrinking factor $\frac{1-\beta_1^t}{\beta_2(1-\beta_2^{t-1})}$ introduced by the gap between the coefficients of momentum

$\hat{\boldsymbol{m}}(t)$ and $\hat{\boldsymbol{m}}(t-1)$. As $t \to \infty$, both $\beta_1^t \to 0$ and $(c\beta_1)^t \to 0$, and we can obtain that $\xi(t) \stackrel{\triangle}{=} \mathcal{L}(\boldsymbol{w}(t)) + \frac{1}{2\sqrt[2]{c}\eta(1-\beta_1)} \left\| \sqrt[4]{\varepsilon\mathbb{1}_d + \hat{\nu}(t-1)} \odot (\boldsymbol{w}(t) - \boldsymbol{w}(t-1)) \right\|^2$ is a Lyapunov function. Based on Lemma 5, we can prove the following asymptotic rate of $\mathcal{L}$.

**Lemma 6.** *Let all conditions in Theorem 4 hold. Then,*

$$\mathcal{L}(\boldsymbol{w}(t)) = \Theta\left(t^{-1}\right), \|\boldsymbol{w}(t)\| = \Theta(\log(t)), \text{ and } \|\boldsymbol{w}(t) - \boldsymbol{w}(t-1)\| = \Theta(t^{-1}).$$

The proof idea of Lemma 6 is that by the exponential-tailed property, when time is large enough, the training loss can be bounded by the gradient, and further bounded by $\|\sqrt[4]{\varepsilon\mathbb{1}_d + \hat{\nu}(t-1)} \odot (\boldsymbol{w}(t) - \boldsymbol{w}(t-1))\|$. Consequently, $\xi(t+1) - \xi(t) \le -\mathcal{O}(\xi(t))$, and $\xi(t) = \mathcal{O}(t^{-1})$.

**Stage II. Bound the difference between $\boldsymbol{w}(t)$ and $\log(t)\hat{\boldsymbol{w}}$.** The Lyapunov function for the direction difference is also modified according to the preconditioner, as introduced in Lemma 7:

**Lemma 7.** *Let all conditions in Theorem 4 hold. Then, $\|\boldsymbol{r}(t)\|$ is bounded if and only if $g(t)$ is upper bounded, where $g(t)$ is defined as follows.*

$$g(t) \stackrel{\triangle}{=} \frac{\sqrt{\varepsilon}}{2}\|\boldsymbol{r}(t)\|^2 + \frac{\beta_1}{1-\beta_1}\left\langle \boldsymbol{r}(t), (1-\beta_1^{t-1})\sqrt{\varepsilon\mathbb{1}_d + \hat{\nu}(t-1)} \odot (\boldsymbol{w}(t) - \boldsymbol{w}(t-1)) \right\rangle$$

$$- \frac{\beta_1}{1-\beta_1}\sum_{\tau=2}^{t}\langle \boldsymbol{r}(\tau) - \boldsymbol{r}(\tau-1), (1-\beta_1^{\tau-1})\sqrt{\varepsilon\mathbb{1}_d + \hat{\nu}(\tau-1)} \odot (\boldsymbol{w}(\tau) - \boldsymbol{w}(\tau-1))\rangle.$$

*Furthermore, we have $\sum_{s=1}^{\infty}(g(t+1) - g(t))$ is upper bounded.*

The first claim of Lemma 7 is similar to that in GDM and SGDM case. However, when we come to the second claim, simple calculation of $g(t+1) - g(t)$ leads to

$$\langle \boldsymbol{r}(t), (\sqrt{\varepsilon} - (1-\beta_1^t)\sqrt{\varepsilon\mathbb{1}_d + \hat{\nu}(t)}) \odot (\boldsymbol{w}(t+1) - \boldsymbol{w}(t))\rangle + \langle \boldsymbol{r}(t), -\sqrt{\varepsilon}\log(\frac{t+1}{t})\hat{\boldsymbol{w}} - \frac{\eta}{1-\beta}\nabla\mathcal{L}(\boldsymbol{w}(t))\rangle.$$

This is where we need Lemma 6, which bounds the gap between $\sqrt{\varepsilon\mathbb{1}_d + \hat{\nu}(t)}$ and $\sqrt{\varepsilon}$ by $\mathcal{O}(t^{-2})$ and makes $\sum_{t=1}^{\infty}\langle \boldsymbol{r}(t), (\sqrt{\varepsilon} - (1-\beta_1^t)\sqrt{\varepsilon\mathbb{1}_d + \hat{\nu}(t)}) \odot (\boldsymbol{w}(t+1) - \boldsymbol{w}(t))\rangle$ an absolute continuous sequence. The second term $\langle \boldsymbol{r}(t), -\sqrt{\varepsilon}\log(\frac{t+1}{t})\hat{\boldsymbol{w}} - \frac{\eta}{1-\beta}\nabla\mathcal{L}(\boldsymbol{w}(t))\rangle$ can be tackled by following the same routine as the GDM/SGDM case. Consequently, the proof of Lemma 7 is completed.

Till now, we have obtained the implicit bias for GDM, SGDM and Adam (w/s), one may wonder whether the implicit bias of stochastic Adam can be obtained. Here, we make some discussions here and put its further investigation for future works. First, our analysis can be extended to prove that the stochastic adaptive heavy-ball algorithm converges to the $L^2$ max margin solution, which is proposed by (Tao et al., 2021) and has the following form[3]:

$$\boldsymbol{w}(t+1) = \boldsymbol{w}(t) - \eta\frac{\nabla\mathcal{L}_{\boldsymbol{B}(t)}(\boldsymbol{w}(t))}{\sqrt{\varepsilon\mathbb{1}_d + \hat{\nu}(t)}} + \beta(\boldsymbol{w}(t) - \boldsymbol{w}(t-1)), t \ge 1,$$

where $\hat{\nu}(t)$ is defined as Eq. (2).The proof is a simple combination of the proof techniques in SGDM and Adam by observing that the stochastic adaptive heavy-ball algorithm has the following equivalent update rule $(\boldsymbol{u}(1) = \boldsymbol{w}(1))$

$$\boldsymbol{u}(t+1) = -\frac{\eta}{1-\beta}\frac{\nabla\mathcal{L}_{\boldsymbol{B}(t)}(\boldsymbol{w}(t))}{\sqrt{\varepsilon\mathbb{1}_d + \hat{\nu}(t)}} + \boldsymbol{u}(t),$$

$$\boldsymbol{w}(t+1) = \beta\boldsymbol{w}(t) + (1-\beta)\boldsymbol{u}(t+1).$$

However, Adam is different from the stochastic adaptive heavy-ball algorithm as it combine the conditioner and momentum as a whole, which makes our constructed Lyapunov function can not be applied. We will put further investigation on stochastic Adam in future work.

---

[3]Compared to (Tao et al., 2021), we put the $\varepsilon\mathbb{1}_d$ inside the square-root and use $\hat{\nu}(t)$ instead of $\nu(t)$ to demonstrate the difference between Adaptive Heavy Ball and Adam, while these changes will not influence the convergent direction.

## 7 DISCUSSIONS

**Extension to the Multi-Class Classification Problem.** As mentioned in the introduction, despite all the previous analyses are aimed at the binary classification problem, they can be naturally extended to the analyses multi-class classification problem. Specifically, in the linear multi-class classification problem, for any $(\boldsymbol{x}, \boldsymbol{y}) \in \mathbb{R}^{d_X} \times \{1, \cdots, C\}$ in the sample set $\boldsymbol{S}$, the (individual) logistic loss with parameter $\boldsymbol{W} \in \mathbb{R}^{C \times d_X}$ is denoted as

$$\ell(\boldsymbol{y}, \boldsymbol{W}\boldsymbol{x}) = \log \frac{e^{\boldsymbol{W}_{\boldsymbol{y},\boldsymbol{x}}}}{\sum_{i=1}^{C} e^{\boldsymbol{W}_{i,\boldsymbol{x}}}}.$$

Correspondingly, dataset $\boldsymbol{S}$ is separable if there exists a parameter $\boldsymbol{W}$, such that $\forall (\boldsymbol{x}, \boldsymbol{y}) \in \boldsymbol{S}$, we have $\boldsymbol{W}_{\boldsymbol{y},\boldsymbol{x}} > \boldsymbol{W}_{i,\boldsymbol{x}}, \forall i \neq \boldsymbol{y}$. The multi-class $L^2$ max-margin problem is then defined as

$$\min \|\boldsymbol{W}\|_F, \ subject \ to: \ \boldsymbol{W}_{\boldsymbol{y},\boldsymbol{x}} \geq \boldsymbol{W}_{i,\boldsymbol{x}} + 1, \forall (\boldsymbol{x}, \boldsymbol{y}) \in \boldsymbol{S}, i \neq \boldsymbol{y},$$

where $\| \cdot \|_F$ denotes the Frobenius norm. Denote $\hat{\boldsymbol{W}}$ as the $L^2$ max-margin solution, we have (mini-batch) SGDM and Adam (w/s) still converges to the direction of $\hat{\boldsymbol{W}}$, the proof of which is deferred to the Appendix D.2.

**Theorem 5.** *For linear multi-class classification problem using logistic loss and almost every separable data, with a small enough learning rate, and $1 > \beta_2 > \beta_1^4 \geq 0$ (for Adam (w/s)), (mini-batch) SGDM and Adam (w/s) converge to the multi-class $L^2$ max-margin solution (a.s. for SGDM).*

**Consistency with the Existing Experimental Results.** Our results stand with existing experimental observations. Specifically, Soudry et al. (2018) conduct experiments using GD and GDM on the same linear separable data (c.f., Figure 1 in (Soudry et al., 2018)), and it is observed that the training behaviors of GD and GDM are quite similar in terms of the direction $\boldsymbol{w}(t)$, the training loss, and the margin, which supports our Theorem 2. Nacson et al. (2019) extend the experiment to the stochastic setting (c.f. Figure 2 in (Nacson et al., 2019)), and observe the same similarity between SGD and SGDM, which agrees with our Theorem 3. Wang et al. (2021) conduct the experiments of SGD, SGDM, Adam (without momentum) and Adam on MNIST using homogeneous neural networks (c.f. Appendix F.1.2 in (Wang et al., 2021)), and observe such similarity for deep neural networks. Our theorems apply to linear models, which is a special case of homogeneous neural network, and meet their observation.

**Gap Between The Linear Model and Deep Neural Networks.** While our results only hold for the linear classification problem, extending the results to the deep neural networks is possible. Specifically, Lyu & Li (2019) construct a Lyapunov function bounding the rate between loss and parameter norm of GD on the homogeneous deep neural networks, and prove the parameter direction converges to some KKT point of the $L^2$ max-margin problem, which extends the result of GD on linear model. Wang et al. (2021) further show the proof techniques in (Lyu & Li, 2019) can be generalized, and successfully extend the Lyapunov function to AdaGrad, RMSProp and Adam (without momentum) on deep homogeneous neural networks. It will be interesting to see whether such a Lyapunov function can be constructed for GDM and Adam in homogeneous neural networks.

## 8 CONCLUSION

In this paper, we study the implicit bias of momentum-based optimizers in linear classification with exponential-tailed loss. Our results indicates that for SGD and Adam, adding momentum will not influence the implicit bias, and the direction of parameter converges to the $L^2$ max-margin solution. Our theoretical results stands with existing experimental observations.

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

# Supplementary materials for
# "Does momentum Change the Implicit Bias on separable data?"

## A    PREPARATIONS

This section collect definitions and lemmas which will be used throughout the proofs.

### A.1    CHARACTERIZATION OF THE MAX-MARGIN SOLUTION

This section collects several commonly-used characterization of the max-margin solution from Nacson et al. (2019) and Soudry et al. (2018).

To start with, we define support vectors and support set, which are two common terms in margin analysis.

**Definition 2** (Support vectors and support set). *For any $i \in [N]$, $\tilde{x}_i$ is called a support vector of the dataset $S$, if*

$$\langle \tilde{x}_i, \hat{w} \rangle = 1.$$

*Correspondingly, $\tilde{x}_i$ is called a non-support vector if $\langle \tilde{x}_i, \hat{w} \rangle > 1$. The support set of $S$ is then defined as*

$$S_s = \{(x_i, y_i) : \langle y_i x_i, \hat{w} \rangle = 1\}.$$

The following lemma delivers $\hat{w}$ as an linear combination of support vectors.

**Lemma 8** (Lemma 12, Soudry et al. (2018)). *For almost every datasets $S$, there exists a unique vector $v = (v_1, \cdots, v_N)$, such that $\hat{w}$ can be represented as*

$$\hat{w} = \sum_{i=1}^{N} v_i \tilde{x}_i, \tag{11}$$

*where $v$ satisfies $v_i = 0$ if $\tilde{x}_i \notin \mathcal{T}(S_s)$, and $v_i > 0$ if $\tilde{x}_i \in \mathcal{T}(S_s)$. Furthermore, the size of $\tilde{S}_s$ is at most $d$.*

By Lemma 8, we further have the following corollary:

**Corollary 2.** *For almost every datasets $S$, the unique $v$ given by Lemma 8 further satisfies that for any positive constant $C_2$, there exists a non-zero vector $\tilde{w}$, such that, $\tilde{x}_i \in \mathcal{T}(S_s)$, we have*

$$C_2 e^{-\langle \tilde{x}_i, \tilde{w} \rangle} = v_i. \tag{12}$$

*Proof.* For almost every datasets $S$, any subsets with size $d$ of $S$ is linearly independent. Since $\tilde{S}_s$ has size no larger than $d$ (by Lemma 8), and Eq. (12) is equivalent to linear equations, the proof is completed. □

For the stochastic case, we will also need the following lemma when we calculate the form of parameter at time $t$.

**Lemma 9** (Lemma 5, Nacson et al. (2019)). *Let $B(s)$ be the random subset used in SGD (i.e., the one used in SGDM). Almost surely, there exists a vector $\check{w}$*

$$\frac{N}{b} \sum_{s=1}^{t-1} \frac{1}{s} \sum_{i: \tilde{x}_i \in \mathcal{T}(B(s) \cap S_s)} v_i \tilde{x}_i = \log\left(\frac{bt}{N}\right) \hat{w} + n(t),$$

*where $n(t)$ satisfies $\|n(t)\| = o(t^{-0.5+\varepsilon})$ for any $\varepsilon > 0$, and $\|n(t+1) - n(t)\| = O(t^{-1})$. If the $B(s)$ is the random subset used in SGDM (i.e., the one used in mini-batch SGDM), then the a.s. condition can be removed.*

A.2    PREPARATIONS OF THE OPTIMIZATION ANALYSIS

This section collects technical lemmas which will be used in latter proofs. We begin with a lemma bounding the smooth constants if the loss is bounded.

**Lemma 10.** *If loss $\ell$ satisfies (D) in Assumption 3, then for any $w_0$, if $\mathcal{L}(w) \leq \mathcal{L}(w_0)$, then we have $\mathcal{L}$ is $H_{s_0}$ smooth at point $w$, where $s_0 = \ell^{-1}(N\mathcal{L}(w_0))$. Furthermore, $\mathcal{L}$ is globally $H_{s_0}$ smooth over the set $\{w : \mathcal{L}(w) \leq \mathcal{L}(w_0)\}$.*

*Proof.* Since $\ell$ is positive, we have $\forall i \in [N]$,

$$\frac{\tilde{\ell}(w, z_i)}{N} < \frac{\sum_{j=1}^N \tilde{\ell}(w, z_j)}{N} = \mathcal{L}(w) \leq \mathcal{L}(w_0),$$

which leads to $\tilde{\ell}(w, z_i) < N\mathcal{L}(w_0)$, and $\ell$ is $H_{s_0}$ smooth at $\langle w, \tilde{x}_i \rangle$.

Furthermore, since $\nabla_w \tilde{\ell}(w, z_i) = \nabla_w \ell(\langle w, \tilde{x}_i \rangle) = \ell'(\langle w, \tilde{x}_i \rangle)\tilde{x}_i$, for any two parameters $w_1$ and $w_2$ close enough to $w$,

$$\|\nabla_w \tilde{\ell}(w_1, z_i) - \nabla_w \tilde{\ell}(w_2, z_i)\| = \|(\ell'(\langle w_1, \tilde{x}_i \rangle) - \ell'(\langle w_2, \tilde{x}_i \rangle))\tilde{x}_i\|$$
$$\leq |\ell'(\langle w_1, \tilde{x}_i \rangle) - \ell'(\langle w_2, \tilde{x}_i \rangle)| \leq H_{s_0}|\langle w_1 - w_2, \tilde{x}_i \rangle| \leq H_{s_0}\|w_1 - w_2\|,$$

and thus,

$$\|\nabla_w \mathcal{L}(w_1) - \nabla_w \mathcal{L}(w_2)\| \leq \frac{1}{N} \sum_{i=1}^N \|\nabla_w \tilde{\ell}(w_1, z_i) - \nabla_w \tilde{\ell}(w_2, z_i)\| \leq H_{s_0}\|w_1 - w_2\|,$$

which completes the proof that $\mathcal{L}$ is locally $H_{s_0}$ smooth at $w$.

Now if $w_1$ and $w_2$ both belong to $\{w : \mathcal{L}(w) \leq \mathcal{L}(w_0)\}$, we have for any $\tilde{x}_i \in \mathcal{T}(S)$, $\langle w_1, \tilde{x}_i \rangle > \ell^{-1}(N\mathcal{L}(w_0))$, and $\langle w_2, \tilde{x}_i \rangle > \ell^{-1}(N\mathcal{L}(w_0))$. Following the same routine as the locally smooth proof, we complete the second argument.

The proof is completed. $\qquad\square$

Based on Assumption 2, we also have the following lemma characterizing the relationship between loss $\ell$ and its derivative $\ell'$ when $x$ is large enough.

**Lemma 11.** *Let loss $\ell$ satisfy Assumption 2. Then, there exists an large enough $x_0$ and a positive real $K$, such that, $\forall x > x_0$, we have*

$$-\frac{1}{K}\ell'(x) \leq \ell(x) \leq -K\ell'(x).$$

*Proof.* By Assumption 2, there exists a large enough $x_0$, such that $\forall x > x_0$, we have

$$\frac{1}{2}e^{-x} \leq -\ell'(x) \leq 2e^{-x}. \tag{13}$$

On the other hand, as $\lim_{t \to \infty} \ell(x) = 0$, we have

$$\ell(x) = \int_{s=x}^\infty -\ell'(s)\mathrm{d}s,$$

which by Eq. (13) leads to

$$\frac{1}{2}e^{-x} = \frac{1}{2}\int_x^\infty e^{-s}\mathrm{d}s \leq \ell(x) \leq 2\int_x^\infty e^{-s}\mathrm{d}s = 2e^{-x}.$$

Therefore, setting $K = 4$ completes the proof. $\qquad\square$

The following lemma bridges the second moment of $\nabla \mathcal{L}_{B(t)}$ with its squared first moment.

**Lemma 12.** *Let the dataset $S$ satisfies the separable assumption 1. Let $B$ be a random subset of $S$ with size $b$ sampled independently and uniformly without replacement. Then, at any point $w$, we have*

$$\|\nabla\mathcal{L}(w)\|^2 \le \mathbb{E}_B\left[\|\nabla\mathcal{L}_B(w)\|^2\right] \le \frac{N}{\gamma^2 b}\|\nabla\mathcal{L}(w)\|^2.$$

*Proof.* To start with, notice that

$$\|\nabla\mathcal{L}(w)\| = \|\mathbb{E}_B\mathcal{L}_B(w)\| \le \mathbb{E}_B\|\mathcal{L}_B(w)\|.$$

Therefore, the first inequality can be directly obtained by Cauchy-Schwartz's inequality. To prove the second inequality, we first calculate the explicit form of $\nabla\mathcal{L}_B(w)$.

$$\|\nabla\mathcal{L}_B(w)\|^2 = \frac{1}{b^2}\left\|\nabla\sum_{z\in B}\tilde{\ell}(w,z)\right\|^2$$

$$= \frac{1}{b^2}\left\|\sum_{\tilde{x}\in\mathcal{T}(B)}\ell'(\langle w,\tilde{x}\rangle)\tilde{x}\right\|^2 = \frac{1}{b^2}\sum_{\tilde{x},\tilde{x}'\in\mathcal{T}(B)}\ell'(\langle w,\tilde{x}\rangle)\ell'(\langle w,\tilde{x}'\rangle)\langle\tilde{x},\tilde{x}'\rangle$$

$$\le \frac{1}{b^2}\sum_{\tilde{x},\tilde{x}'\in\mathcal{T}(B)}\ell'(\langle w,\tilde{x}\rangle)\ell'(\langle w,\tilde{x}'\rangle).$$

Therefore,

$$\mathbb{E}_B\|\nabla\mathcal{L}_B(w)\|^2$$

$$\le \mathbb{E}_B\frac{1}{b^2}\sum_{\tilde{x},\tilde{x}'\in\mathcal{T}(B)}\ell'(\langle w,\tilde{x}\rangle)\ell'(\langle w,\tilde{x}'\rangle) \tag{14}$$

$$= \mathbb{E}_B\frac{1}{b^2}\sum_{\tilde{x},\tilde{x}'\in\mathcal{T}(S)}\ell'(\langle w,\tilde{x}\rangle)\ell'(\langle w,\tilde{x}'\rangle)\mathbb{1}_{\tilde{x},\tilde{x}'\in\mathcal{T}(B)}$$

$$= \frac{1}{b^2}\sum_{\tilde{x},\tilde{x}'\in\mathcal{T}(S)}\ell'(\langle w,\tilde{x}\rangle)\ell'(\langle w,\tilde{x}'\rangle)\mathbb{E}_B\mathbb{1}_{\tilde{x},\tilde{x}'\in\mathcal{T}(B)}$$

$$= \frac{1}{Nb}\sum_{\tilde{x}\in\mathcal{T}(S)}\ell'(\langle w,\tilde{x}\rangle)^2 + \frac{b-1}{bN(N-1)}\sum_{\tilde{x},\tilde{x}'\in\mathcal{T}(S),\tilde{x}\ne\tilde{x}'}\ell'(\langle w,\tilde{x}\rangle)\ell'(\langle w,\tilde{x}'\rangle)$$

$$\le \frac{1}{Nb}\left(\sum_{\tilde{x}\in\mathcal{T}(S)}\ell'(\langle w,\tilde{x}\rangle)\right)^2. \tag{15}$$

On the other hand,

$$\|\nabla\mathcal{L}(w)\| = \frac{1}{N}\left\|\sum_{\tilde{x}\in\mathcal{T}(S)}\ell'(\langle w,\tilde{x}\rangle)\tilde{x}\right\|$$

$$\ge \frac{1}{N}\left\langle\sum_{\tilde{x}\in\mathcal{T}(S)}\ell'(\langle w,\tilde{x}\rangle)\tilde{x}, -\frac{\hat{w}}{\|\hat{w}\|}\right\rangle \overset{(\star)}{\ge} \frac{\gamma}{N}\sum_{\tilde{x}\in\mathcal{T}(S)}\ell'(\langle w,\tilde{x}\rangle)$$

where Eq. $(\star)$ is due to $\forall z \in \langle\tilde{x}, -\hat{w}\rangle \ge 1$ and $\ell' < 0$.

Therefore,

$$\|\nabla\mathcal{L}(w)\|^2 \ge \frac{\gamma^2}{N^2}\left(\sum_{\tilde{x}\in\mathcal{T}(S)}\ell'(\langle w,\tilde{x}\rangle)\right)^2. \tag{16}$$

The proof is completed by putting Eqs. (15) and (16) together. $\qquad\square$

In the following lemma, we show the updates of GDM, Adam, and SGDM are all non-zero.

**Lemma 13.** *Regardless of GDM, Adam, or SGDM, the updates of all steps are non-zero, i.e.,*
$$\|\boldsymbol{w}(t+1) - \boldsymbol{w}(t)\| > 0, \forall t > 1.$$

*Proof.* We start with the alternative forms of the update rule of GDM, Adam, and SGDM using the gradients along the trajectory respectively. For GDM, by Eq. (5), the update rule can be written as

$$\boldsymbol{w}(t+1) - \boldsymbol{w}(t) = -\eta \left( \sum_{s=1}^{t} \beta^{t-s} \nabla \mathcal{L}(\boldsymbol{w}(s)) \right). \tag{17}$$

Similarly, the update rule of SGDM can be written as

$$\boldsymbol{w}(t+1) - \boldsymbol{w}(t) = -\eta \left( \sum_{s=1}^{t} \beta^{t-s} \nabla \mathcal{L}_{\boldsymbol{B}(s)}(\boldsymbol{w}(s)) \right), \tag{18}$$

while the update rule of Adam can be given as

$$\boldsymbol{w}(t+1) - \boldsymbol{w}(t) = -\eta \frac{\sum_{s=1}^{t} \frac{1-\beta_1}{1-\beta_1^s} \beta_1^{t-s} \nabla \mathcal{L}(\boldsymbol{w}(s))}{\sqrt{\varepsilon \mathbf{1}_d + \sum_{s=1}^{t} \frac{1-\beta_2}{1-\beta_2^s} \beta_2^{t-s} (\nabla \mathcal{L}(\boldsymbol{w}(s)))^2}}. \tag{19}$$

On the other hand, by the definition of empirical risk $\mathcal{L}$, the gradient of $\mathcal{L}$ at point $\boldsymbol{w}$ can be given as

$$\nabla \mathcal{L}(\boldsymbol{w}) = \frac{\sum_{i=1}^{N} \ell'(\langle \boldsymbol{w}, \tilde{\boldsymbol{x}}_i \rangle) \tilde{\boldsymbol{x}}_i}{N}. \tag{20}$$

By Eq. (20) and Eq. (17), we further have for GDM,

$$\boldsymbol{w}(t+1) - \boldsymbol{w}(t) = -\eta \left( \sum_{s=1}^{t} \beta^{t-s} \frac{\sum_{i=1}^{N} \ell'(\langle \boldsymbol{w}(s), \tilde{\boldsymbol{x}}_i \rangle) \tilde{\boldsymbol{x}}_i}{N} \right). \tag{21}$$

By Assumption 1, there exists a non-zero parameter $\hat{\boldsymbol{w}}$, such that, $\langle \hat{\boldsymbol{w}}, \tilde{\boldsymbol{x}}_i \rangle > 0, \forall i$. Therefore, by executing inner product between Eq. (21) and $\hat{\boldsymbol{w}}$, we have

$$\|\boldsymbol{w}(t+1) - \boldsymbol{w}(t)\| \|\hat{\boldsymbol{w}}\| \geq \langle \boldsymbol{w}(t+1) - \boldsymbol{w}(t), \hat{\boldsymbol{w}} \rangle = -\eta \left( \sum_{s=1}^{t} \beta^{t-s} \frac{\sum_{i=1}^{N} \ell'(\langle \boldsymbol{w}(s), \tilde{\boldsymbol{x}}_i \rangle) \langle \tilde{\boldsymbol{x}}_i, \hat{\boldsymbol{w}} \rangle}{N} \right) \overset{(*)}{>} 0,$$

where Eq. $(*)$ is due to $\ell' < 0$. This complete the proof for GDM.

Similarly, for SGDM, we have

$$\|\boldsymbol{w}(t+1) - \boldsymbol{w}(t)\| \|\hat{\boldsymbol{w}}\| \geq -\eta \left( \sum_{s=1}^{t} \beta^{t-s} \frac{\sum_{(\boldsymbol{x},\boldsymbol{y}) \in \boldsymbol{B}} \ell'(\langle \boldsymbol{w}(s), \boldsymbol{y}\boldsymbol{x} \rangle) \langle \boldsymbol{y}\boldsymbol{x}, \hat{\boldsymbol{w}} \rangle}{b} \right) > 0,$$

which completes the proof of SGDM.

For Adam, we have

$$\|\boldsymbol{w}(t+1) - \boldsymbol{w}(t)\| \left\| \hat{\boldsymbol{w}} \odot \sqrt{\varepsilon \mathbf{1}_d + \sum_{s=1}^{t} \frac{1-\beta_2}{1-\beta_2^s} \beta_2^{t-s} (\nabla \mathcal{L}(\boldsymbol{w}(s)))^2} \right\|$$

$$\geq - \left\langle \hat{\boldsymbol{w}} \odot \sqrt{\varepsilon \mathbf{1}_d + \sum_{s=1}^{t} \frac{1-\beta_2}{1-\beta_2^s} \beta_2^{t-s} (\nabla \mathcal{L}(\boldsymbol{w}(s)))^2}, \eta \frac{\sum_{s=1}^{t} \frac{1-\beta_1}{1-\beta_1^s} \beta_1^{t-s} \nabla \mathcal{L}(\boldsymbol{w}(s))}{\sqrt{\varepsilon \mathbf{1}_d + \sum_{s=1}^{t} \frac{1-\beta_2}{1-\beta_2^s} \beta_2^{t-s} (\nabla \mathcal{L}(\boldsymbol{w}(s)))^2}} \right\rangle$$

$$= \left\langle \hat{\boldsymbol{w}}, \eta \sum_{s=1}^{t} \frac{1-\beta_1}{1-\beta_1^s} \beta_1^{t-s} \nabla \mathcal{L}(\boldsymbol{w}(s)) \right\rangle$$

$$= -\eta \left( \sum_{s=1}^{t} \frac{1-\beta_1}{1-\beta_1^s} \beta_1^{t-s} \frac{\sum_{i=1}^{N} \ell'(\langle \boldsymbol{w}(s), \tilde{\boldsymbol{x}}_i \rangle) \langle \tilde{\boldsymbol{x}}_i, \hat{\boldsymbol{w}} \rangle}{N} \right) > 0,$$

which completes the proof of Adam.

The proof is completed. $\qquad \square$

# B   IMPLICIT BIAS OF GD/SGD WITH MOMENTUM

This section collects the proof of the implicit bias of gradient descent with momentum and stochastic gradient descent with momentum.

## B.1   IMPLICIT BIAS OF GD WITH MOMENTUM

This section collects the proof of Theorem 2.

### B.1.1   PROOF OF THE SUM OF SQUARED GRADIENTS CONVERGES

To begin with, we will prove the sum of squared norm of gradients along the trajectory is finite for gradient descent with momentum. To see this, we first define the continuous-time update rule as

$$\boldsymbol{w}(t+\alpha) - \boldsymbol{w}(t) = \alpha(\boldsymbol{w}(t+1) - \boldsymbol{w}(t)), \forall t \in \mathbb{Z}^+, \forall \alpha \in [0,1].$$

We then prove a generalized case of Lemma 1 for any $\boldsymbol{w}(t+\alpha)$.

**Lemma 14** (Lemma 1, extended). *Let all conditions in Theorem 2 hold. We then have*

$$\mathcal{L}(\boldsymbol{w}(t)) + \frac{\beta}{2\eta}\|\boldsymbol{w}(t) - \boldsymbol{w}(t-1)\|^2 \geq \mathcal{L}(\boldsymbol{w}(t+\alpha)) + \frac{\beta}{2\eta}\alpha^2\|\boldsymbol{w}(t+1) - \boldsymbol{w}(t)\|^2$$
$$+ \frac{(1-\beta)(1-C_1)\alpha^2}{\eta}\|\boldsymbol{w}(t+1) - \boldsymbol{w}(t)\|^2,$$

*where $C_1$ is a positive real such that $\eta = 2\frac{1-\beta}{H_{s_0}}C_1$.*

*Proof of Lemma 14.* For brevity, we denote $s_0 \overset{\triangle}{=} \ell^{-1}(N\mathcal{L}(\boldsymbol{w}_1))$. We prove this lemma by reduction to absurdity.

Concretely, let $t^*$ be the smallest positive integer time such that there exists an $\alpha \in [0,1]$, such that Eq. (7) doesn't hold. Let $\alpha^* = \inf\{\alpha \in [0,1] : Eq.\ (7)\ doesn't\ hold\ for\ (t^*, \alpha)\}$. By continuity, Eq. (7) holds for $(t^*, \alpha^*)$.

We further divide the proof into two cases depending on the value of $\alpha^*$.

**Case 1:** $\alpha^* = 0$: For any $t^* > t \geq 1$, we have Eq. (7) holds for $(t, 1)$. Specifically, we have

$$\mathcal{L}(\boldsymbol{w}(t)) + \frac{\beta}{2\eta}\|\boldsymbol{w}(t) - \boldsymbol{w}(t-1)\|^2 \geq \mathcal{L}(\boldsymbol{w}(t+1)) + \frac{\beta}{2\eta}\|\boldsymbol{w}(t+1) - \boldsymbol{w}(t)\|^2,$$

which further leads to

$$\mathcal{L}(\boldsymbol{w}(1)) = \mathcal{L}(\boldsymbol{w}(1)) + \frac{\beta}{2\eta}\|\boldsymbol{w}(1) - \boldsymbol{w}(0)\|^2 \geq \mathcal{L}(\boldsymbol{w}(t^*)) + \frac{\beta}{2\eta}\|\boldsymbol{w}(t^*) - \boldsymbol{w}(t^* - 1)\|^2.$$

Since $\frac{\beta}{2\eta}\|\boldsymbol{w}(t^*) - \boldsymbol{w}(t^* - 1)\|^2$ is non-negative, we have

$$\mathcal{L}(\boldsymbol{w}(1)) \geq \mathcal{L}(\boldsymbol{w}(t^*)).$$

By Lemma 10, we have $\mathcal{L}$ is $H_{s_0}$ smooth at $\boldsymbol{w}(t^*)$. Therefore, by Taylor's expansion for $\mathcal{L}$ at point $\boldsymbol{w}(t^*)$, we have for small enough $\alpha > 0$

$$\mathcal{L}(\boldsymbol{w}(t^* + \alpha))$$
$$\leq \mathcal{L}(\boldsymbol{w}(t^*)) + \langle \nabla\mathcal{L}(\boldsymbol{w}(t^*)), \boldsymbol{w}(t^* + \alpha) - \boldsymbol{w}(t^*)\rangle + \frac{H_{s_0}}{2}\|\boldsymbol{w}(t^* + \alpha) - \boldsymbol{w}(t^*)\|^2$$
$$= \mathcal{L}(\boldsymbol{w}(t^*)) + \alpha\langle \nabla\mathcal{L}(\boldsymbol{w}(t^*)), \boldsymbol{w}(t^* + 1) - \boldsymbol{w}(t^*)\rangle + \frac{H_{s_0}\alpha^2}{2}\|\boldsymbol{w}(t^* + 1) - \boldsymbol{w}(t^*)\|^2$$
$$\overset{(*)}{=} \mathcal{L}(\boldsymbol{w}(t^*)) + \alpha\left\langle \frac{1}{\eta}(\beta(\boldsymbol{w}(t^*) - \boldsymbol{w}(t^* - 1)) - (\boldsymbol{w}(t^* + 1) - \boldsymbol{w}(t^*))), \boldsymbol{w}(t^* + 1) - \boldsymbol{w}(t^*)\right\rangle$$
$$+ \frac{H_{s_0}\alpha^2}{2}\|\boldsymbol{w}(t^* + 1) - \boldsymbol{w}(t^*)\|^2$$

$$=\mathcal{L}(\boldsymbol{w}(t^*)) + \frac{\alpha\beta}{\eta}\langle(\boldsymbol{w}(t^*) - \boldsymbol{w}(t^*-1), \boldsymbol{w}(t^*+1) - \boldsymbol{w}(t^*)\rangle + \left(\frac{H_{s_0}\alpha^2}{2} - \frac{\alpha}{\eta}\right)\|\boldsymbol{w}(t^*+1) - \boldsymbol{w}(t^*)\|^2$$

$$\overset{(**)}{\leq}\mathcal{L}(\boldsymbol{w}(t^*)) + \frac{\alpha\beta}{2\eta}\|(\boldsymbol{w}(t^*) - \boldsymbol{w}(t^*-1)\|^2 + \frac{\alpha\beta}{2\eta}\|(\boldsymbol{w}(t^*+1) - \boldsymbol{w}(t^*)\|^2$$

$$+\left(\frac{H_{s_0}\alpha^2}{2} - \frac{\alpha}{\eta}\right)\|\boldsymbol{w}(t^*+1) - \boldsymbol{w}(t^*)\|^2$$

$$=\mathcal{L}(\boldsymbol{w}(t^*)) + \frac{\alpha\beta}{2\eta}\|(\boldsymbol{w}(t^*) - \boldsymbol{w}(t^*-1)\|^2 + \left(\frac{\alpha\beta}{2\eta} - \frac{\alpha}{\eta} + \frac{H_{s_0}\alpha^2}{2}\right)\|(\boldsymbol{w}(t^*+1) - \boldsymbol{w}(t^*)\|^2$$

$$=\mathcal{L}(\boldsymbol{w}(t^*)) + \frac{\beta}{2\eta}\|(\boldsymbol{w}(t^*) - \boldsymbol{w}(t^*-1)\|^2 - \frac{(1-\alpha)\beta}{2\eta}\|(\boldsymbol{w}(t^*) - \boldsymbol{w}(t^*-1)\|^2$$

$$+\left(\frac{\alpha\beta}{2\eta} - \frac{\alpha}{\eta} + \frac{H_{s_0}\alpha^2}{2}\right)\|(\boldsymbol{w}(t^*+1) - \boldsymbol{w}(t^*)\|^2$$

$$\overset{(\diamond)}{\leq}\mathcal{L}(\boldsymbol{w}(t^*)) + \frac{\beta}{2\eta}\|\boldsymbol{w}(t^*) - \boldsymbol{w}(t^*-1)\|^2 - \frac{\beta}{2\eta}\alpha^2\|\boldsymbol{w}(t^*+1) - \boldsymbol{w}(t^*)\|^2$$

$$-\frac{(1-\beta)(1-C_1)\alpha^2}{\eta}\|\boldsymbol{w}(t^*+1) - \boldsymbol{w}(t^*)\|^2, \tag{22}$$

where Eq. $(*)$ is due to a simple rearrangement of the update rule of gradient descent with momentum (Eq. (5)), i.e.,

$$\nabla\mathcal{L}(\boldsymbol{w}(t)) = \frac{1}{\eta}(\beta(\boldsymbol{w}(t) - \boldsymbol{w}(t-1)) - (\boldsymbol{w}(t+1) - \boldsymbol{w}(t))), \forall t \geq 1, \tag{23}$$

Inequality $(**)$ is due to Cauchy Schwarz's inequality and arithmetic-geometric average inequality, and Inequality $(\diamond)$ is due to

$$-\frac{(1-\alpha)\beta}{2\eta}\|(\boldsymbol{w}(t^*) - \boldsymbol{w}(t^*-1)\|^2 + \left(\frac{\alpha\beta}{2\eta} - \frac{\alpha}{\eta} + \frac{H_{s_0}\alpha^2}{2}\right)\|(\boldsymbol{w}(t^*+1) - \boldsymbol{w}(t^*)\|^2$$

$$=-\frac{(1-\alpha)\beta}{2\eta}\|(\boldsymbol{w}(t^*) - \boldsymbol{w}(t^*-1)\|^2 + \mathcal{O}(\alpha)$$

$$\leq\mathcal{O}(\alpha^2)$$

$$=-\frac{\beta}{2\eta}\alpha^2\|\boldsymbol{w}(t^*+1) - \boldsymbol{w}(t^*)\|^2 - \frac{(1-\beta)(1-C_1)\alpha^2}{\eta}\|\boldsymbol{w}(t^*+1) - \boldsymbol{w}(t^*)\|^2.$$

Here the inequality is due to that $-\frac{(1-\alpha)\beta}{2\eta}\|(\boldsymbol{w}(t^*)-\boldsymbol{w}(t^*-1)\|^2$ tend to $-\frac{\beta}{2\eta}\|(\boldsymbol{w}(t^*)-\boldsymbol{w}(t^*-1)\|^2$ as $\alpha$ tend to zero, which is a negative constant by Lemma 13.

Eq. (22) indicates Eq. (7) holds at $(t^*, \alpha)$ for $\alpha > 0$ is small enough, which contradicts to $\alpha^* = 0$.

**Case 2:** $\alpha^* \neq 0$: Same as **Case 1**, we have for any $1 \leq t < t^*$,

$$\mathcal{L}(\boldsymbol{w}(t)) + \frac{\beta}{2\eta}\|\boldsymbol{w}(t) - \boldsymbol{w}(t-1)\|^2 \geq \mathcal{L}(\boldsymbol{w}(t+1)) + \frac{\beta}{2\eta}\|\boldsymbol{w}(t+1) - \boldsymbol{w}(t)\|^2,$$

which further leads to

$$\mathcal{L}(\boldsymbol{w}(1)) \geq \mathcal{L}(\boldsymbol{w}(t^*)) + \frac{\beta}{2\eta}\|\boldsymbol{w}(t^*) - \boldsymbol{w}(t^*-1)\|^2. \tag{24}$$

On the other hand, by the definition of $\alpha^*$, we have for any $0 \leq \alpha < \alpha^*$, we have Eq. (7) holds for $(t^*, \alpha)$, which by continuity further leads to Eq. (7) holds for $(t^*, \alpha^*)$. Therefore, $\alpha^* < 1$, otherwise, Eq. (7) holds for $(t^*, \alpha), \forall \alpha \in [0, 1]$ which contradicts the definition of $t^*$.

Combining Eq. (7) with $(t^*, \alpha)$ and Eq. (24), we further have

$$\mathcal{L}(\boldsymbol{w}(1)) \geq \mathcal{L}(\boldsymbol{w}(t^*+\alpha)) + \frac{\beta}{2\eta}\alpha^2\|\boldsymbol{w}(t+1) - \boldsymbol{w}(t)\|^2 + \frac{1-C_1}{2C_1}H_{s_0}\alpha^2\|\boldsymbol{w}(t+1) - \boldsymbol{w}(t)\|^2,$$

Consequently, for any $\alpha \in [0, \alpha^*]$

$$\mathcal{L}(\boldsymbol{w}(1)) \geq \mathcal{L}(\boldsymbol{w}(t^* + \alpha)),$$

and by Lemma 10, we then have $\mathcal{L}$ is $H_{s_0}$ smooth at $\boldsymbol{w}(t^* + \alpha)$, which further by Taylor's expansion leads to

$$\mathcal{L}(\boldsymbol{w}(t^* + \alpha^*))$$

$$\leq \mathcal{L}(\boldsymbol{w}(t^*)) + \langle \nabla \mathcal{L}(\boldsymbol{w}(t^*)), \boldsymbol{w}(t^* + \alpha^*) - \boldsymbol{w}(t^*) \rangle + \frac{H_{s_0}}{2} \|\boldsymbol{w}(t^* + \alpha^*) - \boldsymbol{w}(t^*)\|^2$$

$$\overset{(\circ)}{\leq} \mathcal{L}(\boldsymbol{w}(t^*)) + \frac{\alpha^* \beta}{2\eta} \|\boldsymbol{w}(t^*) - \boldsymbol{w}(t^* - 1)\|^2 + \frac{\alpha^* \beta}{2\eta} \|\boldsymbol{w}(t^* + 1) - \boldsymbol{w}(t^*)\|^2$$

$$+ \left( \frac{H_{s_0}(\alpha^*)^2}{2} - \frac{\alpha^*}{\eta} \right) \|\boldsymbol{w}(t^* + 1) - \boldsymbol{w}(t^*)\|^2$$

$$= \mathcal{L}(\boldsymbol{w}(t^*)) + \frac{\alpha^* \beta}{2\eta} \|\boldsymbol{w}(t^*) - \boldsymbol{w}(t^* - 1)\|^2 + \left( \frac{H_{s_0}(\alpha^*)^2}{2} - \frac{\alpha^*(2 - \beta)}{2\eta} \right) \|\boldsymbol{w}(t^* + 1) - \boldsymbol{w}(t^*)\|^2$$

$$\overset{(\bullet)}{=} \mathcal{L}(\boldsymbol{w}(t^*)) + \frac{\alpha^* \beta}{2\eta} \|\boldsymbol{w}(t^*) - \boldsymbol{w}(t^* - 1)\|^2 + \left( \frac{(1 - \beta)C_1(\alpha^*)^2}{\eta} - \frac{\alpha^*(2 - \beta)}{2\eta} \right) \|\boldsymbol{w}(t^* + 1) - \boldsymbol{w}(t^*)\|^2$$

$$\overset{(*)}{<} \mathcal{L}(\boldsymbol{w}(t^*)) + \frac{\beta}{2\eta} \|\boldsymbol{w}(t^*) - \boldsymbol{w}(t^* - 1)\|^2 - \frac{(\alpha^*)^2 \beta}{2\eta} \|\boldsymbol{w}(t^* + 1) - \boldsymbol{w}(t^*)\|^2$$

$$- \frac{(1 - \beta)(1 - C_1)(\alpha^*)^2}{\eta} \|\boldsymbol{w}(t^* + 1) - \boldsymbol{w}(t^*)\|^2$$

where Eq. ($\circ$) follows the same routine as **Case 1**, Eq. ($\bullet$) is due to the definition of $\eta$ and $C_1$, and Eq. ($*$) is due to $\alpha^* < 1$, and $\|\boldsymbol{w}(t^* + 1) - \boldsymbol{w}(t^*)\|^2 > 0$ (given by Lemma 13).

By the continuity of $\mathcal{L}$, for any small enough $\delta > 0$, Eq. (7) holds for $(t^*, \alpha^* + \delta)$, which contradicts to the definition of $\alpha^*$.

The proof is completed. $\square$

By Lemma 1, one can easily obtain the sum of the squared norms of the updates across the trajectory converges.

**Corollary 3.** *Let all conditions in Theorem 2 hold. We have*

$$\sum_{t=1}^{\infty} \|\boldsymbol{w}(t + 1) - \boldsymbol{w}(t)\|^2 < \infty. \tag{25}$$

*Consequentially, we have*

$$\|\boldsymbol{w}(t)\| = \mathcal{O}(\sqrt{t}).$$

*Proof.* By Lemma 1, we have

$$\mathcal{L}(\boldsymbol{w}(t)) + \frac{\beta}{2\eta} \|\boldsymbol{w}(t) - \boldsymbol{w}(t - 1)\|^2 - \left( \mathcal{L}(\boldsymbol{w}(t + 1)) + \frac{\beta}{2\eta} \|\boldsymbol{w}(t + 1) - \boldsymbol{w}(t)\|^2 \right)$$

$$\geq \frac{(1 - C_1)(1 - \beta)}{\eta} \|\boldsymbol{w}(t + 1) - \boldsymbol{w}(t)\|^2,$$

which by summing over $t$ further leads to

$$\mathcal{L}(\boldsymbol{w}(1)) \geq \mathcal{L}(\boldsymbol{w}(1)) - \left( \mathcal{L}(\boldsymbol{w}(t + 1)) + \frac{\beta}{2\eta} \|\boldsymbol{w}(t + 1) - \boldsymbol{w}(t)\|^2 \right) \geq \frac{(1 - C_1)(1 - \beta)}{\eta} \sum_{s=1}^{t} \|\boldsymbol{w}(s+1) - \boldsymbol{w}(s)\|^2.$$

Taking $t \to \infty$ leads to

$$\sum_{s=1}^{\infty} \|\boldsymbol{w}(s + 1) - \boldsymbol{w}(s)\|^2 < \infty.$$

By triangle inequality, we further have

$$\|\boldsymbol{w}(t)\| \leq \sum_{s=1}^{t} \|\boldsymbol{w}(s+1) - \boldsymbol{w}(s)\| + \|\boldsymbol{w}(1)\|$$

$$\overset{(\star)}{\leq} \sqrt{t \left( \sum_{s=1}^{t} \|\boldsymbol{w}(s+1) - \boldsymbol{w}(s)\|^2 \right)} + \|\boldsymbol{w}(1)\| = \mathcal{O}(\sqrt{t}),$$

where Eq. $(\star)$ is due to Cauchy-Schwartz's inequality.

The proof is completed. $\qquad\square$

By the negative derivative of the loss and the separable data, we can finally prove Corollary 1.

*Proof of Corollary 1.* By Eq. (21), we have

$$\|\boldsymbol{w}(t+1) - \boldsymbol{w}(t)\|^2 = \eta^2 \left\| \sum_{s=1}^{t} \beta^{t-s} \frac{\sum_{i=1}^{N} \ell'(\langle \boldsymbol{w}(s), \tilde{\boldsymbol{x}}_i\rangle)\tilde{\boldsymbol{x}}_i}{N} \right\|^2$$

$$= \eta^2 \left\| \sum_{s=1}^{t} \beta^{t-s} \frac{\sum_{i=1}^{N} \ell'(\langle \boldsymbol{w}(s), \tilde{\boldsymbol{x}}_i\rangle)\tilde{\boldsymbol{x}}_i}{N} \right\|^2 \frac{\|\hat{\boldsymbol{w}}\|^2}{\|\hat{\boldsymbol{w}}\|^2}$$

$$\overset{(*)}{\geq} \eta^2 \gamma^2 \left\langle \hat{\boldsymbol{w}}, \sum_{s=1}^{t} \beta^{t-s} \frac{\sum_{i=1}^{N} \ell'(\langle \boldsymbol{w}(s), \tilde{\boldsymbol{x}}_i\rangle)\tilde{\boldsymbol{x}}_i}{N} \right\rangle^2$$

$$\overset{(**)}{\geq} \eta^2 \gamma^2 \left( \sum_{s=1}^{t} \beta^{t-s} \frac{\sum_{i=1}^{N} \ell'(\langle \boldsymbol{w}(s), \tilde{\boldsymbol{x}}_i\rangle)}{N} \right)^2$$

$$\geq \eta^2 \gamma^2 \left( \frac{\sum_{i=1}^{N} \ell'(\langle \boldsymbol{w}(t), \tilde{\boldsymbol{x}}_i\rangle)}{N} \right)^2$$

$$\overset{(\bullet)}{\geq} \eta^2 \gamma^2 \left( \frac{\sum_{i=1}^{N} \ell'(\langle \boldsymbol{w}(t), \tilde{\boldsymbol{x}}_i\rangle)\|\tilde{\boldsymbol{x}}_i\|}{N} \right)^2$$

$$\geq \eta^2 \gamma^2 \left\| \frac{\sum_{i=1}^{N} \ell'(\langle \boldsymbol{w}(t), \tilde{\boldsymbol{x}}_i\rangle)\tilde{\boldsymbol{x}}_i}{N} \right\|^2$$

$$= \eta^2 \gamma^2 \|\nabla \mathcal{L}(\boldsymbol{w}(t))\|^2, \tag{26}$$

where Inequality $(*)$ is due to Cauchy-Schwartz's inequality, Inequality $(**)$ is due to $\ell'(s) < 0$, $\forall s \in \mathbb{R}$ and $\langle \hat{\boldsymbol{w}}, \tilde{\boldsymbol{x}}_i\rangle \geq \gamma$, $\forall i \in [N]$, and Inequality $(\bullet)$ is due to $\|\tilde{\boldsymbol{x}}_i\| \leq 1$. By combining Eq. (25) and Eq. (26), we complete the proof. $\qquad\square$

By the exponential-tailed assumption of the loss (Assumption 2), we further have the following corollary.

**Corollary 4.** *Let all conditions in Theorem 2 hold. Then, $\lim_{t\to\infty} \|\nabla \mathcal{L}(\boldsymbol{w}(t))\| = 0$, and*

$$\lim_{t\to\infty} \langle \boldsymbol{w}(t), \tilde{\boldsymbol{x}}_i\rangle = \infty, \forall i.$$

*Consequently, there exists an large enough time $t_0$, such that, $\forall t > t_0$, $\forall i$, we have $\langle \boldsymbol{w}(t), \tilde{\boldsymbol{x}}_i\rangle > 0$, and*

$$-\ell'(\langle \boldsymbol{w}(t), \tilde{\boldsymbol{x}}_i\rangle) \leq (1 + e^{-\mu_+ \langle \boldsymbol{w}(t), \tilde{\boldsymbol{x}}_i\rangle})e^{-\langle \boldsymbol{w}(t), \tilde{\boldsymbol{x}}_i\rangle},$$
$$-\ell'(\langle \boldsymbol{w}(t), \tilde{\boldsymbol{x}}_i\rangle) \geq (1 - e^{-\mu_- \langle \boldsymbol{w}(t), \tilde{\boldsymbol{x}}_i\rangle})e^{-\langle \boldsymbol{w}(t), \tilde{\boldsymbol{x}}_i\rangle}.$$

### B.1.2 BOUNDING THE ORTHOGONAL PART

To prove Theorem 2, we only need to show $\boldsymbol{w}(t) - \log(t)\hat{\boldsymbol{w}}$ $(t \geq 1)$ has bounded norm for any iteration $t > 0$. Letting $C_2 = \frac{\eta}{(1-\beta)N}$ in Corollary 2, we obtain an constant vector $\tilde{\boldsymbol{w}}$ satisfying Eq. (12). Define

$$\boldsymbol{r}(t) \overset{\triangle}{=} \boldsymbol{w}(t) - \log(t)\hat{\boldsymbol{w}} - \tilde{\boldsymbol{w}}. \tag{27}$$

As $\tilde{\boldsymbol{w}}$ is a constant vector, that $\boldsymbol{w}(t) - \log(t)\hat{\boldsymbol{w}}$ $(t \geq 1)$ has bounded norm is equivalent to $\boldsymbol{r}(t)$ has bounded norm. Lemma 2 then propose an equivalent proposition of $\|\boldsymbol{r}(t)\|$ is bounded, and further prove this proposition is fulfilled. As the proof is rather complex, we separate it into two sub-lemmas. We first prove $\|\boldsymbol{r}(t)\|$ is bounded if and only if function $g(t)$ is upper bounded.

**Lemma 15** (First argument in Lemma 2). *Let all conditions in Theorem 2 hold. Then, $\|\boldsymbol{r}(t)\|$ is bounded if and only if function $g(t)$ is upper bounded.*

*Proof.* We start the proof by showing that $A_1(t) \overset{\triangle}{=} \sum_{\tau=2}^{t} \langle \boldsymbol{r}(\tau) - \boldsymbol{r}(\tau-1), \boldsymbol{w}(\tau) - \boldsymbol{w}(\tau-1) \rangle$ has bounded absolute value.

By the definition of $\boldsymbol{r}(t)$, we have

$$\boldsymbol{r}(t) - \boldsymbol{r}(t-1) = \boldsymbol{w}(t) - \boldsymbol{w}(t-1) - \log\left(\frac{t}{t-1}\right)\hat{\boldsymbol{w}},$$

which further indicates

$$A_1(t) = \sum_{\tau=2}^{t} \left\langle \boldsymbol{w}(\tau) - \boldsymbol{w}(\tau-1) - \log\left(\frac{\tau}{\tau-1}\right)\hat{\boldsymbol{w}}, \boldsymbol{w}(\tau) - \boldsymbol{w}(\tau-1) \right\rangle.$$

Therefore, the absolute value of $A_1(t)$ can be bounded as

$$|A_1(t)| = \left| \sum_{\tau=2}^{t} \left\langle \boldsymbol{w}(\tau) - \boldsymbol{w}(\tau-1) - \log\left(\frac{\tau}{\tau-1}\right)\hat{\boldsymbol{w}}, \boldsymbol{w}(\tau) - \boldsymbol{w}(\tau-1) \right\rangle \right|$$

$$\leq \sum_{\tau=2}^{t} \left| \left\langle \boldsymbol{w}(\tau) - \boldsymbol{w}(\tau-1) - \log\left(\frac{\tau}{\tau-1}\right)\hat{\boldsymbol{w}}, \boldsymbol{w}(\tau) - \boldsymbol{w}(\tau-1) \right\rangle \right|$$

$$\leq \sum_{\tau=2}^{t} \|\boldsymbol{w}(\tau) - \boldsymbol{w}(\tau-1)\|^2 + \sum_{\tau=2}^{t} \left| \left\langle \log\left(\frac{\tau}{\tau-1}\right)\hat{\boldsymbol{w}}, \boldsymbol{w}(\tau) - \boldsymbol{w}(\tau-1) \right\rangle \right|$$

$$\leq \sum_{\tau=2}^{t} \|\boldsymbol{w}(\tau) - \boldsymbol{w}(\tau-1)\|^2 + \sum_{\tau=2}^{t} \left\| \log\left(\frac{\tau}{\tau-1}\right)\hat{\boldsymbol{w}} \right\| \|\boldsymbol{w}(\tau) - \boldsymbol{w}(\tau-1)\|$$

$$\overset{(\star)}{\leq} \frac{3}{2} \sum_{\tau=2}^{t} \|\boldsymbol{w}(\tau) - \boldsymbol{w}(\tau-1)\|^2 + \frac{1}{2} \sum_{\tau=2}^{t} \left\| \log\left(\frac{\tau}{\tau-1}\right)\hat{\boldsymbol{w}} \right\|^2$$

$$\overset{(\circ)}{<} \infty,$$

where Inequality $(\star)$ is due to the Inequality of arithmetic and geometric means, and Inequality $(\circ)$ is due to Corollary 3 and $\log\frac{\tau}{\tau-1} = \mathcal{O}(\frac{1}{\tau})$.

Therefore, $g(t)$ is upper bounded is then equivalent to $\frac{1}{2}\|\boldsymbol{r}(t)\|^2 + \frac{\beta}{1-\beta}\langle \boldsymbol{r}(t), \boldsymbol{w}(t) - \boldsymbol{w}(t-1) \rangle$ is upper bounded. Now if $\frac{1}{2}\|\boldsymbol{r}(t)\|^2 + \frac{\beta}{1-\beta}\langle \boldsymbol{r}(t), \boldsymbol{w}(t) - \boldsymbol{w}(t-1) \rangle$ is upper bounded, we will prove $\|\boldsymbol{r}(t)\|$ is bounded by reduction to absurdity.

Suppose that $\|\boldsymbol{r}(t)\|$ has unbounded norm. By Corollary 3, we have $\lim_{t\to\infty} \|\boldsymbol{w}(t) - \boldsymbol{w}(t-1)\| = 0$, and there exists a large enough time $T$, such that $\|\boldsymbol{w}(t) - \boldsymbol{w}(t-1)\| < 1$ for any $t \geq T$. On the other hand, since $\boldsymbol{r}(t)$ is unbounded from above, there exists an increasing time sequence $k_i > T$, $i \in \mathbb{Z}^+$, such that

$$\lim_{i\to\infty} \|\boldsymbol{r}(k_i)\| = \infty.$$

Therefore, we have

$$\lim_{i \to \infty} \frac{1}{2} \|\boldsymbol{r}(k_i)\|^2 + \frac{\beta}{1-\beta} \langle \boldsymbol{r}(k_i), \boldsymbol{w}(k_i) - \boldsymbol{w}(k_i - 1) \rangle$$

$$\geq \lim_{i \to \infty} \frac{1}{2} \|\boldsymbol{r}(k_i)\|^2 - \frac{\beta}{1-\beta} \|\boldsymbol{r}(k_i)\| \|\boldsymbol{w}(k_i) - \boldsymbol{w}(k_i - 1)\|$$

$$\geq \lim_{i \to \infty} \frac{1}{2} \|\boldsymbol{r}(k_i)\|^2 - \frac{\beta}{1-\beta} \|\boldsymbol{r}(k_i)\| = \infty,$$

which leads to contradictory, and completes the proof of necessity.

On the other hand, if $\|\boldsymbol{r}(t)\|$ is upper bounded, since $\|\boldsymbol{w}(t) - \boldsymbol{w}(t-1)\|$ is also upper bounded, we have $\frac{1}{2}\|\boldsymbol{r}(t)\|^2 + \frac{\beta}{1-\beta}\langle \boldsymbol{r}(t), \boldsymbol{w}(t) - \boldsymbol{w}(t-1) \rangle$ is upper bounded, which completes the proof of sufficiency.

The proof is completed. □

Therefore, the last piece of this puzzle is to prove $g(t)$ is upper bounded $\forall t > 0$.

**Lemma 16** (Second argument in Lemma 2). *Let all conditions in Theorem 2 hold. Then, we have $g(t)$ is upper bounded.*

*Proof.* We start the proof by calculating $g(t+1) - g(t)$. For any $t \geq 2$, we have

$$g(t+1) - g(t) = \frac{1}{2}\|\boldsymbol{r}(t+1) - \boldsymbol{r}(t)\|^2 + \langle \boldsymbol{r}(t), \boldsymbol{r}(t+1) - \boldsymbol{r}(t) \rangle + \frac{\beta}{1-\beta}\langle \boldsymbol{r}(t+1), \boldsymbol{w}(t+1) - \boldsymbol{w}(t) \rangle$$

$$- \frac{\beta}{1-\beta}\langle \boldsymbol{r}(t), \boldsymbol{w}(t) - \boldsymbol{w}(t-1) \rangle - \frac{\beta}{1-\beta}\langle \boldsymbol{r}(t+1) - \boldsymbol{r}(t), \boldsymbol{w}(t+1) - \boldsymbol{w}(t) \rangle$$

$$= \frac{1}{2}\|\boldsymbol{r}(t+1) - \boldsymbol{r}(t)\|^2 + \langle \boldsymbol{r}(t), \boldsymbol{r}(t+1) - \boldsymbol{r}(t) \rangle + \frac{\beta}{1-\beta}\langle \boldsymbol{r}(t), \boldsymbol{w}(t+1) + \boldsymbol{w}(t-1) - 2\boldsymbol{w}(t) \rangle.$$

On the other hand, by simply rearranging the update rule Eq. (5), we have

$$\frac{\beta}{1-\beta}(\boldsymbol{w}(t+1) + \boldsymbol{w}(t-1) - 2\boldsymbol{w}(t)) = -\frac{\eta}{1-\beta}\nabla\mathcal{L}(\boldsymbol{w}(t)) - (\boldsymbol{w}(t+1) - \boldsymbol{w}(t)), \qquad (28)$$

which further indicates

$$g(t+1) - g(t)$$

$$= \frac{1}{2}\|\boldsymbol{r}(t+1) - \boldsymbol{r}(t)\|^2 + \langle \boldsymbol{r}(t), \boldsymbol{r}(t+1) - \boldsymbol{r}(t) \rangle + \left\langle \boldsymbol{r}(t), -\frac{\eta}{1-\beta}\nabla\mathcal{L}(\boldsymbol{w}(t)) - (\boldsymbol{w}(t+1) - \boldsymbol{w}(t)) \right\rangle$$

$$= \frac{1}{2}\|\boldsymbol{r}(t+1) - \boldsymbol{r}(t)\|^2 + \left\langle \boldsymbol{r}(t), -\log\left(\frac{t+1}{t}\right)\hat{\boldsymbol{w}} - \frac{\eta}{1-\beta}\nabla\mathcal{L}(\boldsymbol{w}(t)) \right\rangle.$$

Denote $A_2(t) = \|\boldsymbol{r}(t+1) - \boldsymbol{r}(t)\|^2$, and $A_3(t) = \left\langle \boldsymbol{r}(t), -\log\left(\frac{t+1}{t}\right)\hat{\boldsymbol{w}} - \frac{\eta}{1-\beta}\nabla\mathcal{L}(\boldsymbol{w}(t)) \right\rangle$. We then prove respectively $\sum_{t=1}^{\infty} A_2(t)$ and $\sum_{t=1}^{\infty} A_3(t)$ are upper bounded.

First of all, by definition of $\boldsymbol{r}(t)$ Eq.(27), we have

$$\sum_{t=1}^{\infty} A_2(t) = \sum_{t=1}^{\infty}\left(\|\boldsymbol{w}(t+1) - \boldsymbol{w}(t)\|^2 + \log\left(\frac{t+1}{t}\right)^2\|\hat{\boldsymbol{w}}\|^2 - 2\log\left(\frac{t+1}{t}\right)\langle \boldsymbol{w}(t+1) - \boldsymbol{w}(t), \hat{\boldsymbol{w}} \rangle\right)$$

$$\leq 2\sum_{t=1}^{\infty}\left(\|\boldsymbol{w}(t+1) - \boldsymbol{w}(t)\|^2 + \log\left(\frac{t+1}{t}\right)^2\|\hat{\boldsymbol{w}}\|^2\right) \overset{(\bullet)}{<} \infty, \qquad (29)$$

where Eq. ($\bullet$) is due to Lemma 3 and $\log\left(\frac{t+1}{t}\right) = \mathcal{O}(\frac{1}{t})$.

Then we only need to prove $\sum_{t=1}^{\infty} A_3(t) < \infty$.

To begin with, by adding one additional term $\frac{1}{t}\hat{\boldsymbol{w}}$ into $A_3$, we have

$$A_3(t) = \left\langle \boldsymbol{r}(t), \frac{1}{t}\hat{\boldsymbol{w}} - \log\left(\frac{t+1}{t}\right)\hat{\boldsymbol{w}} \right\rangle + \left\langle \boldsymbol{r}(t), -\frac{1}{t}\hat{\boldsymbol{w}} - \frac{\eta}{1-\beta}\nabla\mathcal{L}(\boldsymbol{w}(t)) \right\rangle.$$

On the one hand, by Corollary 3, $\|\boldsymbol{w}(t)\| = \mathcal{O}(\sqrt{t})$, which further leads to

$$\|\boldsymbol{r}(t)\| = \|\boldsymbol{w}(t)\| + \log(t)\|\hat{\boldsymbol{w}}\| + \|\hat{\boldsymbol{w}}\| = \mathcal{O}(\sqrt{t})$$

By $\frac{1}{t} - \log\frac{t+1}{t} = \mathcal{O}\left(\frac{1}{t^2}\right)$, we have

$$\left\langle \boldsymbol{r}(t), \frac{1}{t}\hat{\boldsymbol{w}} - \log\left(\frac{t+1}{t}\right)\hat{\boldsymbol{w}} \right\rangle = \mathcal{O}\left(\frac{1}{t^{\frac{3}{2}}}\right). \tag{30}$$

On the other hand, by direct calculation of the gradient, we have

$$\left\langle \boldsymbol{r}(t), -\frac{1}{t}\hat{\boldsymbol{w}} - \frac{\eta}{1-\beta}\nabla\mathcal{L}(\boldsymbol{w}(t)) \right\rangle$$

$$= \left\langle \boldsymbol{r}(t), -\frac{1}{t}\hat{\boldsymbol{w}} - \frac{\eta}{(1-\beta)N}\sum_{i=1}^{N}\ell'(\langle\boldsymbol{w}(t),\tilde{\boldsymbol{x}}_i\rangle)\tilde{\boldsymbol{x}}_i \right\rangle$$

$$\overset{(\star)}{=} \frac{1}{N}\left\langle \boldsymbol{r}(t), -\frac{1}{t}\frac{\eta}{1-\beta}\sum_{\tilde{\boldsymbol{x}}_i\in\mathcal{T}(\boldsymbol{S}_s)}e^{-\langle\tilde{\boldsymbol{w}},\tilde{\boldsymbol{x}}_i\rangle}\tilde{\boldsymbol{x}}_i - \frac{\eta}{1-\beta}\sum_{i=1}^{N}\ell'(\langle\boldsymbol{w},\tilde{\boldsymbol{x}}_i\rangle)\tilde{\boldsymbol{x}}_i \right\rangle$$

$$= \frac{1}{N}\left\langle \boldsymbol{r}(t), -\frac{\eta}{1-\beta}\sum_{\tilde{\boldsymbol{x}}_i\in\mathcal{T}(\boldsymbol{S}_s)}\left(\frac{1}{t}e^{-\langle\tilde{\boldsymbol{w}},\tilde{\boldsymbol{x}}_i\rangle} + \ell'(\langle\boldsymbol{w}(t),\tilde{\boldsymbol{x}}_i\rangle)\right)\tilde{\boldsymbol{x}}_i \right\rangle - \frac{1}{N}\left\langle \boldsymbol{r}(t), \frac{\eta}{1-\beta}\sum_{\tilde{\boldsymbol{x}}_i\notin\mathcal{T}(\boldsymbol{S}_s)}\ell'(\langle\boldsymbol{w}(t),\tilde{\boldsymbol{x}}_i\rangle)\tilde{\boldsymbol{x}}_i \right\rangle,$$

where Eq. $(\star)$ is due to the definition of $\tilde{\boldsymbol{w}}$ (Eq. (12) with $C_2 = \frac{\eta}{1-\beta}$).

Denote

$$A_4(t) = -\left\langle \boldsymbol{r}(t), \frac{\eta}{1-\beta}\sum_{\tilde{\boldsymbol{x}}_i\notin\mathcal{T}(\boldsymbol{S}_s)}\ell'(\langle\boldsymbol{w}(t),\tilde{\boldsymbol{x}}_i\rangle)\tilde{\boldsymbol{x}}_i \right\rangle,$$

and

$$A_5(t) = \left\langle \boldsymbol{r}(t), -\frac{\eta}{1-\beta}\sum_{\tilde{\boldsymbol{x}}_i\in\mathcal{T}(\boldsymbol{S}_s)}\left(\frac{1}{t}e^{-\langle\tilde{\boldsymbol{w}},\tilde{\boldsymbol{x}}_i\rangle} + \ell'(\langle\boldsymbol{w}(t),\tilde{\boldsymbol{x}}_i\rangle)\right)\tilde{\boldsymbol{x}}_i \right\rangle.$$

We then analysis these two terms respectively. As for $A_4(t)$, due to $\ell' < 0$, we have

$$A_4(t) \le -\frac{\eta}{1-\beta}\left\langle \boldsymbol{r}(t), \sum_{\tilde{\boldsymbol{x}}_i\notin\mathcal{T}(\boldsymbol{S}_s),\langle\boldsymbol{r}(t),\tilde{\boldsymbol{x}}_i\rangle>0}\ell'(\langle\boldsymbol{w}(t),\tilde{\boldsymbol{x}}_i\rangle)\tilde{\boldsymbol{x}}_i \right\rangle.$$

By Corollary 4, we further have $\forall t > t_0$

$$-\ell'(\langle\boldsymbol{w}(t),\tilde{\boldsymbol{x}}_i\rangle) \le (1 + e^{-\mu_+\langle\boldsymbol{w}(t),\tilde{\boldsymbol{x}}_i\rangle})e^{-\langle\boldsymbol{w}(t),\tilde{\boldsymbol{x}}_i\rangle} \le 2e^{-\langle\boldsymbol{w}(t),\tilde{\boldsymbol{x}}_i\rangle},$$

which further indicates

$$A_4(t) \le -\frac{\eta}{1-\beta}\sum_{\tilde{\boldsymbol{x}}_i\notin\mathcal{T}(\boldsymbol{S}_s),\langle\boldsymbol{r}(t),\tilde{\boldsymbol{x}}_i\rangle>0}\ell'(\langle\boldsymbol{w}(t),\tilde{\boldsymbol{x}}_i\rangle)\langle\boldsymbol{r}(t),\tilde{\boldsymbol{x}}_i\rangle$$

$$\le \frac{\eta}{1-\beta}\sum_{\tilde{\boldsymbol{x}}_i\notin\mathcal{T}(\boldsymbol{S}_s),\langle\boldsymbol{r}(t),\tilde{\boldsymbol{x}}_i\rangle>0}2e^{-\langle\boldsymbol{w}(t),\tilde{\boldsymbol{x}}_i\rangle}\langle\boldsymbol{r}(t),\tilde{\boldsymbol{x}}_i\rangle$$

$$= \frac{\eta}{1-\beta}\sum_{\tilde{\boldsymbol{x}}_i\notin\mathcal{T}(\boldsymbol{S}_s),\langle\boldsymbol{r}(t),\tilde{\boldsymbol{x}}_i\rangle>0}2e^{-\langle\boldsymbol{r}(t)+\log t\hat{\boldsymbol{w}}+\tilde{\boldsymbol{w}},\tilde{\boldsymbol{x}}_i\rangle}\langle\boldsymbol{r}(t),\tilde{\boldsymbol{x}}_i\rangle$$

$$\le \frac{\eta}{1-\beta}\left(\max_i e^{\langle-\tilde{\boldsymbol{w}},\tilde{\boldsymbol{x}}_i\rangle}\right)\sum_{\tilde{\boldsymbol{x}}_i\notin\mathcal{T}(\boldsymbol{S}_s),\langle\boldsymbol{r}(t),\tilde{\boldsymbol{x}}_i\rangle>0}2e^{-\langle\boldsymbol{r}(t)+\log t\hat{\boldsymbol{w}},\tilde{\boldsymbol{x}}_i\rangle}\langle\boldsymbol{r}(t),\tilde{\boldsymbol{x}}_i\rangle$$

$$\overset{(\circ)}{\le} \frac{\eta}{1-\beta}\frac{\left(\max_i e^{\langle-\tilde{\boldsymbol{w}},\tilde{\boldsymbol{x}}_i\rangle}\right)}{t^\theta}\sum_{\tilde{\boldsymbol{x}}_i\notin\mathcal{T}(\boldsymbol{S}_s),\langle\boldsymbol{r}(t),\tilde{\boldsymbol{x}}_i\rangle>0}2e^{-\langle\boldsymbol{r}(t),\tilde{\boldsymbol{x}}_i\rangle}\langle\boldsymbol{r}(t),\tilde{\boldsymbol{x}}_i\rangle$$

$$\overset{(\diamond)}{\le} \frac{\eta}{1-\beta}\frac{\left(\max_i e^{\langle-\tilde{\boldsymbol{w}},\tilde{\boldsymbol{x}}_i\rangle}\right)}{t^\theta}2N,$$

where $\theta$ in Eq. ($\circ$) is defined as

$$\theta = \min_{\tilde{\boldsymbol{x}}_i \notin \mathcal{T}(\boldsymbol{S}_s)} \langle \tilde{\boldsymbol{x}}_i, \hat{\boldsymbol{w}} \rangle > 1. \tag{31}$$

As $\sum_{t=1}^{\infty} \frac{1}{t^\theta} < \infty$, we have

$$\sum_{t=1}^{\infty} A_4(t) < \infty^4. \tag{32}$$

For each term $\left\langle \boldsymbol{r}(t), -\frac{\eta}{1-\beta} \left( \frac{1}{t} e^{-\langle \tilde{\boldsymbol{w}}, \tilde{\boldsymbol{x}}_i \rangle} + \ell'(\langle \boldsymbol{w}, \tilde{\boldsymbol{x}}_i \rangle) \right) \tilde{\boldsymbol{x}}_i \right\rangle$ ($\tilde{\boldsymbol{x}}_i \notin \mathcal{T}(\boldsymbol{S}_s)$) in $A_5(t)$, we divide the analysis into two parts depending on the sign of $\langle \boldsymbol{r}(t), \boldsymbol{x}_i \rangle$.

**Case 1:** $\langle \boldsymbol{r}(t), \tilde{\boldsymbol{x}}_i \rangle \geq 0$. By Corollary 4, we have

$$\left\langle \boldsymbol{r}(t), -\frac{\eta}{1-\beta} \left( \frac{1}{t} e^{-\langle \tilde{\boldsymbol{w}}, \tilde{\boldsymbol{x}}_i \rangle} + \ell'(\langle \boldsymbol{w}, \tilde{\boldsymbol{x}}_i \rangle) \right) \tilde{\boldsymbol{x}}_i \right\rangle$$

$$= -\frac{\eta}{1-\beta} \left( \frac{1}{t} e^{-\langle \tilde{\boldsymbol{w}}, \tilde{\boldsymbol{x}}_i \rangle} + \ell'(\langle \boldsymbol{w}, \tilde{\boldsymbol{x}}_i \rangle) \right) \langle \boldsymbol{r}(t), \tilde{\boldsymbol{x}}_i \rangle$$

$$\leq \frac{\eta}{1-\beta} \left( -\frac{1}{t} e^{-\langle \tilde{\boldsymbol{w}}, \tilde{\boldsymbol{x}}_i \rangle} + (1 + e^{-\mu_+ \langle \boldsymbol{w}(t), \tilde{\boldsymbol{x}}_i \rangle}) e^{-\langle \boldsymbol{w}(t), \tilde{\boldsymbol{x}}_i \rangle} \right) \langle \boldsymbol{r}(t), \tilde{\boldsymbol{x}}_i \rangle$$

$$\stackrel{(\diamond)}{=} \frac{\eta}{1-\beta} \left( -\frac{1}{t} e^{-\langle \tilde{\boldsymbol{w}}, \tilde{\boldsymbol{x}}_i \rangle} + (1 + e^{-\mu_+ \langle \boldsymbol{r}(t) + \log t \hat{\boldsymbol{w}} + \tilde{\boldsymbol{w}}, \tilde{\boldsymbol{x}}_i \rangle}) e^{-\langle \boldsymbol{r}(t) + \log t \hat{\boldsymbol{w}} + \tilde{\boldsymbol{w}}, \tilde{\boldsymbol{x}}_i \rangle} \right) \langle \boldsymbol{r}(t), \tilde{\boldsymbol{x}}_i \rangle,$$

where Eq. ($\diamond$) is due to the definition of $\boldsymbol{r}(t)$ (Eq. (27)).

Since $\langle \boldsymbol{r}(t), \tilde{\boldsymbol{x}}_i \rangle \geq 0$, we further have

$$\left\langle \boldsymbol{r}(t), -\frac{\eta}{1-\beta} \left( \frac{1}{t} e^{-\langle \tilde{\boldsymbol{w}}, \tilde{\boldsymbol{x}}_i \rangle} + \ell'(\langle \boldsymbol{w}, \tilde{\boldsymbol{x}}_i \rangle) \right) \tilde{\boldsymbol{x}}_i \right\rangle$$

$$\leq \frac{\eta}{1-\beta} \left( -\frac{1}{t} e^{-\langle \tilde{\boldsymbol{w}}, \tilde{\boldsymbol{x}}_i \rangle} + (1 + e^{-\mu_+ \langle \log t \hat{\boldsymbol{w}} + \tilde{\boldsymbol{w}}, \tilde{\boldsymbol{x}}_i \rangle}) e^{-\langle \boldsymbol{r}(t) + \log t \hat{\boldsymbol{w}} + \tilde{\boldsymbol{w}}, \tilde{\boldsymbol{x}}_i \rangle} \right) \langle \boldsymbol{r}(t), \tilde{\boldsymbol{x}}_i \rangle$$

$$\stackrel{(\square)}{=} \frac{\eta}{1-\beta} \left( -\frac{1}{t} e^{-\langle \tilde{\boldsymbol{w}}, \tilde{\boldsymbol{x}}_i \rangle} + \frac{1}{t}(1 + t^{-\mu_+} e^{-\mu_+ \langle \tilde{\boldsymbol{w}}, \tilde{\boldsymbol{x}}_i \rangle}) e^{-\langle \boldsymbol{r}(t) + \tilde{\boldsymbol{w}}, \tilde{\boldsymbol{x}}_i \rangle} \right) \langle \boldsymbol{r}(t), \tilde{\boldsymbol{x}}_i \rangle$$

$$= \frac{\eta}{1-\beta} \frac{1}{t} e^{-\langle \tilde{\boldsymbol{w}}, \tilde{\boldsymbol{x}}_i \rangle} \left( -1 + (1 + t^{-\mu_+} e^{-\mu_+ \langle \tilde{\boldsymbol{w}}, \tilde{\boldsymbol{x}}_i \rangle}) e^{-\langle \boldsymbol{r}(t), \tilde{\boldsymbol{x}}_i \rangle} \right) \langle \boldsymbol{r}(t), \tilde{\boldsymbol{x}}_i \rangle,$$

where Eq. ($\square$) is due to $\langle \hat{\boldsymbol{w}}, \tilde{\boldsymbol{x}}_i \rangle = 1, \forall \tilde{\boldsymbol{x}}_i \in \mathcal{T}(\boldsymbol{S}_s)$.

Specifically,

$$-1 + (1 + t^{-\mu_+} e^{-\mu_+ \langle \tilde{\boldsymbol{w}}, \tilde{\boldsymbol{x}}_i \rangle}) e^{-\langle \boldsymbol{r}(t), \tilde{\boldsymbol{x}}_i \rangle}$$

$$= -1 + e^{-\langle \boldsymbol{r}(t), \tilde{\boldsymbol{x}}_i \rangle} + t^{-\mu_+} e^{-\mu_+ \langle \tilde{\boldsymbol{w}}, \tilde{\boldsymbol{x}}_i \rangle} e^{-\langle \boldsymbol{r}(t), \tilde{\boldsymbol{x}}_i \rangle}$$

$$\leq t^{-\mu_+} e^{-\mu_+ \langle \tilde{\boldsymbol{w}}, \tilde{\boldsymbol{x}}_i \rangle} e^{-\langle \boldsymbol{r}(t), \tilde{\boldsymbol{x}}_i \rangle}.$$

Therefore,

$$\frac{\eta}{1-\beta} \frac{1}{t} e^{-\langle \tilde{\boldsymbol{w}}, \tilde{\boldsymbol{x}}_i \rangle} \left( -1 + (1 + t^{-\mu_+} e^{-\mu_+ \langle \tilde{\boldsymbol{w}}, \tilde{\boldsymbol{x}}_i \rangle}) e^{-\langle \boldsymbol{r}(t), \tilde{\boldsymbol{x}}_i \rangle} \right) \langle \boldsymbol{r}(t), \tilde{\boldsymbol{x}}_i \rangle$$

$$\leq \frac{\eta}{1-\beta} \frac{1}{t} e^{-\langle \tilde{\boldsymbol{w}}, \tilde{\boldsymbol{x}}_i \rangle} \left( t^{-\mu_+} e^{-\mu_+ \langle \tilde{\boldsymbol{w}}, \tilde{\boldsymbol{x}}_i \rangle} e^{-\langle \boldsymbol{r}(t), \tilde{\boldsymbol{x}}_i \rangle} \right) \langle \boldsymbol{r}(t), \tilde{\boldsymbol{x}}_i \rangle$$

$$\leq \frac{\eta}{(1-\beta)e} \frac{1}{t^{1+\mu_+}} e^{-(1+\mu_+) \langle \tilde{\boldsymbol{w}}, \tilde{\boldsymbol{x}}_i \rangle} = \mathcal{O}\left( \frac{1}{t^{1+\mu_+}} \right).$$

---

[4]In this paper, for a real series $\{r_i\}_{i=1}^{\infty}$, we use $\sum_{i=1}^{\infty} r_i < \infty$ representing $\sum_{i=1}^{T} r_i$ is uniformly upper bounded for any $T$.

**Case 2:** $\langle r(t), \tilde{x}_i \rangle < 0$. Similar to **Case 1.**, in this case we have

$$\left\langle r(t), -\frac{\eta}{1-\beta}\left(\frac{1}{t}e^{-\langle \tilde{w}, \tilde{x}_i \rangle} + \ell'(\langle w, \tilde{x}_i \rangle)\right)\tilde{x}_i \right\rangle$$

$$\leq \frac{\eta}{1-\beta}\left(-\frac{1}{t}e^{-\langle \tilde{w}, \tilde{x}_i \rangle} + \left(1 - e^{-\mu_-\langle w(t), \tilde{x}_i \rangle}\right)e^{-\langle w(t), \tilde{x}_i \rangle}\right)\langle r(t), \tilde{x}_i \rangle$$

$$= \frac{\eta}{1-\beta}\left(-\frac{1}{t}e^{-\langle \tilde{w}, \tilde{x}_i \rangle} + \left(1 - e^{-\mu_-\langle r(t)+\log t\hat{w}+\tilde{w}, \tilde{x}_i \rangle}\right)e^{-\langle r(t)+\log t\hat{w}+\tilde{w}, \tilde{x}_i \rangle}\right)\langle r(t), \tilde{x}_i \rangle$$

$$= \frac{\eta}{1-\beta}\frac{1}{t}e^{-\langle \tilde{w}, \tilde{x}_i \rangle}\left(-1 + \left(1 - e^{-\mu_-\langle r(t)+\log t\hat{w}+\tilde{w}, \tilde{x}_i \rangle}\right)e^{-\langle r(t), \tilde{x}_i \rangle}\right)\langle r(t), \tilde{x}_i \rangle.$$

Specifically, if $\langle r(t), \tilde{x}_i \rangle \geq -t^{-0.5\mu_-}$,

$$\left|\frac{\eta}{1-\beta}\frac{1}{t}e^{-\langle \tilde{w}, \tilde{x}_i \rangle}\left(-1 + (1 - e^{-\mu_-\langle r(t)+\log t\hat{w}+\tilde{w}, \tilde{x}_i \rangle})e^{-\langle r(t), \tilde{x}_i \rangle}\right)\langle r(t), \tilde{x}_i \rangle\right|$$

$$= \left|\frac{\eta}{1-\beta}\frac{1}{t}e^{-\langle \tilde{w}, \tilde{x}_i \rangle}\left(-1 + (1 - t^{-\mu_-}e^{-\mu_-\langle r(t)+\tilde{w}, \tilde{x}_i \rangle})e^{-\langle r(t), \tilde{x}_i \rangle}\right)\langle r(t), \tilde{x}_i \rangle\right|$$

$$\leq \frac{\eta}{1-\beta}\frac{1}{t^{1+0.5\mu_-}}e^{-\langle \tilde{w}, \tilde{x}_i \rangle}\left|-1 + \left(1 - t^{-\mu_-}e^{-\mu_-\langle r(t)+\tilde{w}, \tilde{x}_i \rangle}\right)e^{-\langle r(t), \tilde{x}_i \rangle}\right|$$

$$\overset{(\dagger)}{=} \mathcal{O}\left(\frac{1}{t^{1+0.5\mu_-}}\right),$$

where Eq. (†) is due to if $\langle r(t), \tilde{x}_i \rangle \geq -t^{-0.5\mu_-}$,

$$\lim_{t\to\infty}\left|-1 + \left(1 - t^{-\mu_-}e^{-\mu_-\langle r(t)+\tilde{w}, \tilde{x}_i \rangle}\right)e^{-\langle r(t), \tilde{x}_i \rangle}\right| = 0.$$

If $-2 \leq \langle r(t), \tilde{x}_i \rangle < -t^{-0.5\mu_-}$, we have

$$\frac{\eta}{1-\beta}\frac{1}{t}e^{-\langle \tilde{w}, \tilde{x}_i \rangle}\left(-1 + \left(1 - e^{-\mu_-\langle r(t)+\log t\hat{w}+\tilde{w}, \tilde{x}_i \rangle}\right)e^{-\langle r(t), \tilde{x}_i \rangle}\right)\langle r(t), \tilde{x}_i \rangle$$

$$= \frac{\eta}{1-\beta}\frac{1}{t}e^{-\langle \tilde{w}, \tilde{x}_i \rangle}\left(-1 + \left(1 - \frac{1}{t^{\mu_-}}e^{-\mu_-\langle r(t)+\tilde{w}, \tilde{x}_i \rangle}\right)e^{-\langle r(t), \tilde{x}_i \rangle}\right)\langle r(t), \tilde{x}_i \rangle$$

$$\leq \frac{\eta}{1-\beta}\frac{1}{t}e^{-\langle \tilde{w}, \tilde{x}_i \rangle}\left(-1 + \left(1 - \frac{e^{2\mu_-}}{t^{\mu_-}}e^{-\mu_-\langle \tilde{w}, \tilde{x}_i \rangle}\right)e^{-\langle r(t), \tilde{x}_i \rangle}\right)\langle r(t), \tilde{x}_i \rangle.$$

Therefore, when $t$ is large enough, $1 - \frac{e^{2\mu_-}}{t^{\mu_-}}e^{-\mu_-\langle \tilde{w}, \tilde{x}_i \rangle} > 0$, which by $e^{-\langle r(t), \tilde{x}_i \rangle} \geq 1 - \langle r(t), \tilde{x}_i \rangle$ leads to

$$\frac{\eta}{1-\beta}\frac{1}{t}e^{-\langle \tilde{w}, \tilde{x}_i \rangle}\left(-1 + \left(1 - \frac{e^{2\mu_-}}{t^{\mu_-}}e^{-\mu_-\langle \tilde{w}, \tilde{x}_i \rangle}\right)e^{-\langle r(t), \tilde{x}_i \rangle}\right)\langle r(t), \tilde{x}_i \rangle$$

$$\leq \frac{\eta}{1-\beta}\frac{1}{t}e^{-\langle \tilde{w}, \tilde{x}_i \rangle}\left(-1 + \left(1 - \frac{e^{2\mu_-}}{t^{\mu_-}}e^{-\mu_-\langle \tilde{w}, \tilde{x}_i \rangle}\right)(1 - \langle r(t), \tilde{x}_i \rangle)\right)\langle r(t), \tilde{x}_i \rangle$$

$$\leq \frac{\eta}{1-\beta}\frac{1}{t}e^{-\langle \tilde{w}, \tilde{x}_i \rangle}\left(-1 + \left(1 - \frac{e^{2\mu_-}}{t^{\mu_-}}e^{-\mu_-\langle \tilde{w}, \tilde{x}_i \rangle}\right)\left(1 + \frac{1}{t^{0.5\mu_-}}\right)\right)\langle r(t), \tilde{x}_i \rangle$$

$$= \frac{\eta}{1-\beta}\frac{1}{t}e^{-\langle \tilde{w}, \tilde{x}_i \rangle}\left(\frac{1}{t^{0.5\mu_-}} + o\left(\frac{1}{t^{0.5\mu_-}}\right)\right)\langle r(t), \tilde{x}_i \rangle < 0.$$

If $-2 > \langle r(t), \tilde{x}_i \rangle$,

$$\frac{\eta}{1-\beta}\frac{1}{t}e^{-\langle \tilde{w}, \tilde{x}_i \rangle}\left(-1 + \left(1 - e^{-\mu_-\langle r(t)+\log t\hat{w}+\tilde{w}, \tilde{x}_i \rangle}\right)e^{-\langle r(t), \tilde{x}_i \rangle}\right)\langle r(t), \tilde{x}_i \rangle$$

$$= \frac{\eta}{1-\beta}\frac{1}{t}e^{-\langle \tilde{w}, \tilde{x}_i \rangle}\left(-1 + \left(1 - e^{-\mu_-\langle w(t), \tilde{x}_i \rangle}\right)e^{-\langle r(t), \tilde{x}_i \rangle}\right)\langle r(t), \tilde{x}_i \rangle.$$

For large enough $t$, $1 - e^{-\mu_- \langle \boldsymbol{w}(t), \tilde{\boldsymbol{x}}_i \rangle} > \frac{1}{2}$, and

$$
\frac{\eta}{1 - \beta} \frac{1}{t} e^{-\langle \tilde{\boldsymbol{w}}, \tilde{\boldsymbol{x}}_i \rangle} \left( -1 + \left( 1 - e^{-\mu_- \langle \boldsymbol{w}(t), \tilde{\boldsymbol{x}}_i \rangle} \right) e^{-\langle \boldsymbol{r}(t), \tilde{\boldsymbol{x}}_i \rangle} \right) \langle \boldsymbol{r}(t), \tilde{\boldsymbol{x}}_i \rangle
$$

$$
\leq \frac{\eta}{1 - \beta} \frac{1}{t} e^{-\langle \tilde{\boldsymbol{w}}, \tilde{\boldsymbol{x}}_i \rangle} \left( -1 + \left( 1 - e^{-\mu_- \langle \boldsymbol{w}(t), \tilde{\boldsymbol{x}}_i \rangle} \right) e^2 \right) \langle \boldsymbol{r}(t), \tilde{\boldsymbol{x}}_i \rangle
$$

$$
\leq \frac{\eta}{1 - \beta} \frac{1}{t} e^{-\langle \tilde{\boldsymbol{w}}, \tilde{\boldsymbol{x}}_i \rangle} \left( -1 + \frac{e^2}{2} \right) \langle \boldsymbol{r}(t), \tilde{\boldsymbol{x}}_i \rangle < 0.
$$

Therefore, in **Case 2.**, for large enough $t$, we have

$$
\left\langle \boldsymbol{r}(t), -\frac{\eta}{1 - \beta} \left( \frac{1}{t} e^{-\langle \tilde{\boldsymbol{w}}, \tilde{\boldsymbol{x}}_i \rangle} + \ell'(\langle \boldsymbol{w}, \tilde{\boldsymbol{x}}_i \rangle) \right) \tilde{\boldsymbol{x}}_i \right\rangle \leq \mathcal{O} \left( \frac{1}{t^{1 + 0.5 \mu_-}} \right).
$$

Combining **Case 1.** and **Case 2.**, we conclude that

$$
A_5(t) \leq \mathcal{O} \left( \frac{1}{t^{1 + 0.5 \mu_+}} \right),
$$

which further yields

$$
\sum_{t=1}^{\infty} A_5(t) < \infty. \tag{33}
$$

Combining Eq. (32) and Eq. (33), we conclude that $\sum_{t=1}^{\infty} A_3(t) < \infty$, which together with Eq. (29) yields $\sum_{t=2}^{\infty} g(t+1) - g(t) < \infty$, and completes the proof. $\qquad \square$

We are now ready to prove Theorem 2.

*Proof of Theorem 2.* By Lemma 16, we have $g(t)$ is upper bounded. Therefore, by Lemma 2, we have $\|\boldsymbol{r}(t)\|$ is bounded, which further indicates $\|\boldsymbol{w}(t) - \log(t)\hat{\boldsymbol{w}}\|$ is bounded.

Therefore, the direction of $\boldsymbol{w}(t)$ can be calculated as

$$
\frac{\boldsymbol{w}(t)}{\|\boldsymbol{w}(t)\|} = \frac{\log(t)\hat{\boldsymbol{w}}}{\|\boldsymbol{w}(t)\|} + \frac{\boldsymbol{w}(t) - \log(t)\hat{\boldsymbol{w}}}{\|\boldsymbol{w}(t)\|} = \frac{\log(t)\hat{\boldsymbol{w}}}{\|\log(t)\hat{\boldsymbol{w}} + \boldsymbol{w}(t) - \log(t)\hat{\boldsymbol{w}}\|} + \frac{\boldsymbol{w}(t) - \log(t)\hat{\boldsymbol{w}}}{\|\boldsymbol{w}(t)\|}
$$

$$
= \frac{\hat{\boldsymbol{w}}}{\left\| \hat{\boldsymbol{w}} + \frac{\boldsymbol{w}(t) - \log(t)\hat{\boldsymbol{w}}}{\log t} \right\|} + \frac{\boldsymbol{w}(t) - \log(t)\hat{\boldsymbol{w}}}{\|\boldsymbol{w}(t)\|} \to \frac{\hat{\boldsymbol{w}}}{\|\hat{\boldsymbol{w}}\|} \ (as \ t \to \infty).
$$

The proof is completed.

$\qquad \square$

## B.2 IMPLICIT BIAS OF SGD WITH MOMENTUM

This section collects the proof of Theorem 3. Following the same framework as Appendix B.1, we will first prove that the sum of the squared gradient norms along the trajectory is finite. One may expect $\mathcal{L}(\boldsymbol{w}(t)) + \frac{\beta}{2\eta} \|\boldsymbol{w}(t) - \boldsymbol{w}(t-1)\|^2$ is a Lyapunov function of SGDM. However, due to the randomness of the update rule of SGDM, $\mathcal{L}(\boldsymbol{w}(t)) + \frac{\beta}{2\eta} \|\boldsymbol{w}(t) - \boldsymbol{w}(t-1)\|^2$ may no longer decrease (we will show this in the end of Appendix B.2, please see Appendix B.2.3 for explanation).

### B.2.1 PROOF OF THE SUM OF GRADIENTS ALONG THE TRAJECTORY IS FINITE

We first provide a proof of Lemma 3.

*Proof of Lemma 3.* We denote the parameter of the $t$-th step in Eq. (9) as $\tilde{\boldsymbol{w}}$, while that in Eq. (1) as $\boldsymbol{w}(t)$. With $\boldsymbol{w}(1) = \tilde{\boldsymbol{w}}(1)$, we will prove $\tilde{\boldsymbol{w}}(t) = \boldsymbol{w}(t)$, $\forall t > 1$ iteratively. Specifically, suppose for any $1 \le k \le t$ $\boldsymbol{w}(k) = \tilde{\boldsymbol{w}}(k)$. Then, we have

$$
\begin{aligned}
\boldsymbol{u}(t+1) &= -\tilde{\eta}\nabla\mathcal{L}_{\boldsymbol{B}(t)}(\tilde{\boldsymbol{w}}(t)) + \boldsymbol{u}(t) = -\tilde{\eta}\nabla\mathcal{L}_{\boldsymbol{B}(t)}(\boldsymbol{w}(t)) + \boldsymbol{u}(t) \\
&= -\tilde{\eta}\nabla\mathcal{L}_{\boldsymbol{B}(t)}(\boldsymbol{w}(t)) - \tilde{\eta}\nabla\mathcal{L}_{\boldsymbol{B}(t-1)}(\boldsymbol{w}(t-1)) + \boldsymbol{u}(t-1) \\
&= \cdots = -\sum_{k=1}^{t}\tilde{\eta}\nabla\mathcal{L}_{\boldsymbol{B}(k)}(\boldsymbol{w}(k)) + \boldsymbol{u}(1) = -\sum_{k=1}^{t}\tilde{\eta}\nabla\mathcal{L}_{\boldsymbol{B}(k)}(\boldsymbol{w}(k)) + \boldsymbol{w}(1).
\end{aligned}
$$

Therefore,

$$
\tilde{\boldsymbol{w}}(t+1) = \beta\tilde{\boldsymbol{w}}(t) + (1-\beta)\boldsymbol{u}(t+1) = \beta\tilde{\boldsymbol{w}}(t) + (1-\beta)\left(-\sum_{k=1}^{t}\tilde{\eta}\nabla\mathcal{L}_{\boldsymbol{B}(k)}(\tilde{\boldsymbol{w}}(k)) + \tilde{\boldsymbol{w}}(1)\right),
$$

which by iteration further indicates

$$
\tilde{\boldsymbol{w}}(t+1) = \tilde{\boldsymbol{w}}(1) - \tilde{\eta}\left(\sum_{k=1}^{t}(1-\beta^{t+1-k})\nabla\mathcal{L}_{\boldsymbol{B}(k)}(\tilde{\boldsymbol{w}}(k))\right) = \boldsymbol{w}(1) - \tilde{\eta}\left(\sum_{k=1}^{t}(1-\beta^{t+1-k})\nabla\mathcal{L}_{\boldsymbol{B}(k)}(\boldsymbol{w}(k))\right),
$$

$$
\boldsymbol{w}(t) = \boldsymbol{w}(1) - \tilde{\eta}\left(\sum_{k=1}^{t-1}(1-\beta^{t-k})\nabla\mathcal{L}_{\boldsymbol{B}(k)}(\boldsymbol{w}(k))\right),
$$

and

$$
\boldsymbol{w}(t-1) = \boldsymbol{w}(1) - \tilde{\eta}\left(\sum_{k=1}^{t-2}(1-\beta^{t-1-k})\nabla\mathcal{L}_{\boldsymbol{B}(k)}(\boldsymbol{w}(k))\right).
$$

On the other hand, by Eq. (1), we have

$$
\begin{aligned}
&\boldsymbol{w}(t+1) \\
=&\boldsymbol{w}(t) - \eta\nabla\mathcal{L}_{\boldsymbol{B}(t)}(\boldsymbol{w}(t)) + \beta(\boldsymbol{w}(t) - \boldsymbol{w}(t-1)) \\
=&\boldsymbol{w}(1) - \tilde{\eta}\left(\sum_{k=1}^{t-1}(1-\beta^{t-k})\nabla\mathcal{L}_{\boldsymbol{B}(k)}(\boldsymbol{w}(k))\right) - \eta\nabla\mathcal{L}_{\boldsymbol{B}(t)}(\boldsymbol{w}(t)) + \beta\tilde{\eta}\left(\sum_{k=1}^{t-1}(\beta^{t-1-k} - \beta^{t-k})\nabla\mathcal{L}_{\boldsymbol{B}(k)}(\boldsymbol{w}(k))\right) \\
=&\boldsymbol{w}(1) - \tilde{\eta}\left(\sum_{k=1}^{t}(1-\beta^{t+1-k})\nabla\mathcal{L}_{\boldsymbol{B}(k)}(\boldsymbol{w}(k))\right) = \tilde{\boldsymbol{w}}(t+1).
\end{aligned}
$$

The proof is completed. $\qquad\square$

Using the alternative form given by Lemma 3, we then prove Lemma 4, which indicates $\mathcal{L}(\boldsymbol{u}(t))$ is a proper choice of Lyapunov function.

*Proof of Lemma 4.* We start the proof by applying the Taylor's expansion of $\mathcal{L}$ at the point $\boldsymbol{u}(t)$ to the point $\boldsymbol{u}(t+1)$. Concretely, by Assumption 3. (S), we have

$$
\mathcal{L}(\boldsymbol{u}(t+1)) \le \mathcal{L}(\boldsymbol{u}(t)) + \langle\boldsymbol{u}(t+1) - \boldsymbol{u}(t), \nabla\mathcal{L}(\boldsymbol{u}(t))\rangle + \frac{H}{2}\|\boldsymbol{u}(t+1) - \boldsymbol{u}(t)\|^2,
$$

which by Eq. (9) leads to

$$
\mathcal{L}(\boldsymbol{u}(t+1)) \le \mathcal{L}(\boldsymbol{u}(t)) - \tilde{\eta}\langle\nabla\mathcal{L}_{\boldsymbol{B}(t)}(\boldsymbol{w}(t)), \nabla\mathcal{L}(\boldsymbol{u}(t))\rangle + \frac{H\tilde{\eta}^2}{2}\|\nabla\mathcal{L}_{\boldsymbol{B}(t)}(\boldsymbol{w}(t))\|^2. \qquad (34)
$$

Taking the expectation of Eq. (34) with respect to $\boldsymbol{w}(t+1)$ conditioning on $\{\boldsymbol{w}(s)\}_{s=1}^{t}$, we have

$$\mathbb{E}_{\boldsymbol{w}(t+1)}[\mathcal{L}(\boldsymbol{u}(t+1))|\{\boldsymbol{w}(s)\}_{s=1}^{t}]$$

$$\overset{(\star)}{=}\mathbb{E}_{\boldsymbol{B}(t)}[\mathcal{L}(\boldsymbol{u}(t+1))|\{\boldsymbol{w}(s)\}_{s=1}^{t}]$$

$$\leq\mathbb{E}_{\boldsymbol{B}(t)}\left[\mathcal{L}(\boldsymbol{u}(t))-\tilde{\eta}\langle\nabla\mathcal{L}_{\boldsymbol{B}(t)}(\boldsymbol{w}(t)),\nabla\mathcal{L}(\boldsymbol{u}(t))\rangle+\frac{H\tilde{\eta}^2}{2}\|\nabla\mathcal{L}_{\boldsymbol{B}(t)}(\boldsymbol{w}(t))\|^2\Big|\{\boldsymbol{w}(s)\}_{s=1}^{t}\right]$$

$$\overset{(\circ)}{=}\mathcal{L}(\boldsymbol{u}(t))-\tilde{\eta}\langle\nabla\mathcal{L}(\boldsymbol{w}(t)),\nabla\mathcal{L}(\boldsymbol{u}(t))\rangle+\frac{H\tilde{\eta}^2}{2}\mathbb{E}_{\boldsymbol{B}(t)}\left[\|\nabla\mathcal{L}_{\boldsymbol{B}(t)}(\boldsymbol{w}(t))\|^2\right]$$

$$\overset{(\bullet)}{\leq}\mathcal{L}(\boldsymbol{u}(t))-\tilde{\eta}\langle\nabla\mathcal{L}(\boldsymbol{w}(t)),\nabla\mathcal{L}(\boldsymbol{u}(t))\rangle+\frac{H\tilde{\eta}^2 N}{2b\gamma^2}\|\nabla\mathcal{L}(\boldsymbol{w}(t))\|^2, \tag{35}$$

where Eq. $(\star)$ is due to that $\boldsymbol{w}(t+1)$ is uniquely determined by $\boldsymbol{B}(t)$ given $\{\boldsymbol{w}(s)\}_{s=1}^{t}$, Eq. $(\circ)$ is due to $\boldsymbol{u}(t)$ is uniquely determined by $\{\boldsymbol{w}(s)\}_{s=1}^{t}$, and Inequality. $(\bullet)$ is due to Lemma 12.

Therefore, we have

$$\mathbb{E}_{\boldsymbol{w}(t+1)}[\mathcal{L}(\boldsymbol{u}(t+1))|\{\boldsymbol{w}(s)\}_{s=1}^{t}]$$

$$\leq\mathcal{L}(\boldsymbol{u}(t))-\tilde{\eta}\langle\nabla\mathcal{L}(\boldsymbol{w}(t)),\nabla\mathcal{L}(\boldsymbol{u}(t))\rangle+\frac{H\tilde{\eta}^2 N}{2b\gamma^2}\|\nabla\mathcal{L}(\boldsymbol{w}(t))\|^2$$

$$=\mathcal{L}(\boldsymbol{u}(t))-\tilde{\eta}\langle\nabla\mathcal{L}(\boldsymbol{w}(t)),\nabla\mathcal{L}(\boldsymbol{w}(t))\rangle+\tilde{\eta}\langle\nabla\mathcal{L}(\boldsymbol{w}(t)),\nabla\mathcal{L}(\boldsymbol{w}(t))-\nabla\mathcal{L}(\boldsymbol{u}(t))\rangle+\frac{H\tilde{\eta}^2 N}{2b\gamma^2}\|\nabla\mathcal{L}(\boldsymbol{w}(t))\|^2$$

$$=\mathcal{L}(\boldsymbol{u}(t))-\tilde{\eta}\left(1-\frac{H\tilde{\eta}N}{2b\gamma^2}\right)\|\nabla\mathcal{L}(\boldsymbol{w}(t))\|^2+\langle\tilde{\eta}\nabla\mathcal{L}(\boldsymbol{w}(t)),\nabla\mathcal{L}(\boldsymbol{w}(t))-\nabla\mathcal{L}(\boldsymbol{u}(t))\rangle$$

$$\leq\mathcal{L}(\boldsymbol{u}(t))-\tilde{\eta}\left(1-\frac{H\tilde{\eta}N}{2b\gamma^2}\right)\|\nabla\mathcal{L}(\boldsymbol{w}(t))\|^2+\frac{1}{2}\|\tilde{\eta}\nabla\mathcal{L}(\boldsymbol{w}(t))\|^2+\frac{1}{2}\|\nabla\mathcal{L}(\boldsymbol{w}(t))-\nabla\mathcal{L}(\boldsymbol{u}(t))\|^2$$

$$=\mathcal{L}(\boldsymbol{u}(t))-\tilde{\eta}\left(1-\left(\frac{1}{2}+\frac{HN}{2b\gamma^2}\right)\tilde{\eta}\right)\|\nabla\mathcal{L}(\boldsymbol{w}(t))\|^2+\frac{1}{2}\|\nabla\mathcal{L}(\boldsymbol{w}(t))-\nabla\mathcal{L}(\boldsymbol{u}(t))\|^2.$$

By Assumption 3. (S), $\ell$ is $H$-smooth, which further leads to

$$\|\nabla\mathcal{L}(\boldsymbol{w}(t))-\nabla\mathcal{L}(\boldsymbol{u}(t))\|^2=\left\|\sum_{\tilde{\boldsymbol{x}}\in\tilde{S}}\left(\ell'(\langle\tilde{\boldsymbol{x}},\boldsymbol{w}(t)\rangle)-\ell'(\langle\tilde{\boldsymbol{x}},\boldsymbol{u}(t)\rangle)\right)\tilde{\boldsymbol{x}}\right\|^2$$

$$\leq\left(\sum_{\tilde{\boldsymbol{x}}\in\tilde{S}}|\ell'(\langle\tilde{\boldsymbol{x}},\boldsymbol{w}(t)\rangle)-\ell'(\langle\tilde{\boldsymbol{x}},\boldsymbol{u}(t)\rangle)|\right)^2\leq\left(H\sum_{\tilde{\boldsymbol{x}}\in\tilde{S}}|\langle\tilde{\boldsymbol{x}},\boldsymbol{w}(t)\rangle-\langle\tilde{\boldsymbol{x}},\boldsymbol{u}(t)\rangle|\right)^2$$

$$\leq H^2 N^2\|\boldsymbol{w}(t)-\boldsymbol{u}(t)\|^2\overset{(\square)}{\leq}\frac{H^2 N^2\beta^2}{(1-\beta)^2}\|\boldsymbol{w}(t)-\boldsymbol{w}(t-1)\|^2$$

$$=\frac{H^2 N^2\beta^2}{(1-\beta)^2}\left\|\sum_{s=1}^{t-1}\eta\beta^{t-1-s}\nabla\mathcal{L}_{\boldsymbol{B}(s)}(\boldsymbol{w}(s))\right\|^2=H^2 N^2\tilde{\eta}^2\beta^2\left\|\sum_{s=1}^{t-1}\beta^{t-1-s}\nabla\mathcal{L}_{\boldsymbol{B}(s)}(\boldsymbol{w}(s))\right\|^2$$

$$\overset{(\circ)}{\leq}\frac{H^2 N^2\tilde{\eta}^2\beta^2}{\gamma^2}\left(\sum_{s=1}^{t-1}\beta^{t-1-s}\|\nabla\mathcal{L}_{\boldsymbol{B}(s)}(\boldsymbol{w}(s))\|\right)^2$$

$$\overset{(\clubsuit)}{\leq}\frac{H^2 N^2\tilde{\eta}^2\beta^2}{\gamma^2}\left(\sum_{s=1}^{t-1}\beta^{t-1-s}\|\nabla\mathcal{L}_{\boldsymbol{B}(s)}(\boldsymbol{w}(s))\|^2\right)\left(\sum_{s=1}^{t-1}\beta^{t-1-s}\right)$$

$$\leq\frac{H^2 N^2\tilde{\eta}^2\beta^2}{\gamma^2(1-\beta)}\left(\sum_{s=1}^{t-1}\beta^{t-1-s}\|\nabla\mathcal{L}_{\boldsymbol{B}(s)}(\boldsymbol{w}(s))\|^2\right), \tag{36}$$

where Inequality $(\square)$ is due to $\beta(\boldsymbol{w}(t)-\boldsymbol{w}(t-1))=(1-\beta)(\boldsymbol{u}(t)-\boldsymbol{w}(t))$ by Eq. (9), Inequality $(\diamond)$ is due to triangular inequality, and Inequality $(\clubsuit)$ is due to Cauchy-Schwartz Inequality.

Combining Eqs. (35) and (36), we have

$$\mathbb{E}_{\boldsymbol{w}(t+1)}[\mathcal{L}(\boldsymbol{u}(t+1))|\{\boldsymbol{w}(s)\}_{s=1}^{t}]$$
$$\leq \mathcal{L}(\boldsymbol{u}(t)) - \tilde{\eta}\left(1 - \left(\frac{1}{2} + \frac{HN}{2b\gamma^2}\right)\tilde{\eta}\right)\|\nabla\mathcal{L}(\boldsymbol{w}(t))\|^2 + \frac{1}{2}\frac{H^2N^2\tilde{\eta}^2\beta^2}{\gamma^2(1-\beta)}\left(\sum_{s=1}^{t-1}\beta^{t-1-s}\left\|\nabla\mathcal{L}_{\boldsymbol{B}(s)}(\boldsymbol{w}(s))\right\|^2\right),$$

which by taking expectation to $\{\boldsymbol{w}(s)\}_{s=1}^{t}$ leads to

$$\mathbb{E}_{\mathcal{F}_{t+1}}[\mathcal{L}(\boldsymbol{u}(t+1))] = \mathbb{E}_{\{\boldsymbol{w}(s)\}_{s=1}^{t+1}}[\mathcal{L}(\boldsymbol{u}(t+1))]$$
$$\leq \mathbb{E}_{\mathcal{F}_t}\mathcal{L}(\boldsymbol{u}(t)) - \mathbb{E}_{\mathcal{F}_t}\tilde{\eta}\left(1 - \left(\frac{1}{2} + \frac{HN}{2b\gamma^2}\right)\tilde{\eta}\right)\|\nabla\mathcal{L}(\boldsymbol{w}(t))\|^2 + \mathbb{E}_{\mathcal{F}_t}\frac{1}{2}\frac{H^2N^2\tilde{\eta}^2\beta^2}{\gamma^2(1-\beta)}\left(\sum_{s=1}^{t-1}\beta^{t-1-s}\left\|\nabla\mathcal{L}_{\boldsymbol{B}(s)}(\boldsymbol{w}(s))\right\|^2\right)$$
$$= \mathbb{E}_{\mathcal{F}_t}\mathcal{L}(\boldsymbol{u}(t)) - \tilde{\eta}\left(1 - \left(\frac{1}{2} + \frac{HN}{2b\gamma^2}\right)\tilde{\eta}\right)\mathbb{E}_{\mathcal{F}_t}\|\nabla\mathcal{L}(\boldsymbol{w}(t))\|^2 + \frac{1}{2}\frac{H^2N^2\tilde{\eta}^2\beta^2}{\gamma^2(1-\beta)}\left(\sum_{s=1}^{t-1}\beta^{t-1-s}\mathbb{E}_{\mathcal{F}_{s+1}}\left\|\nabla\mathcal{L}_{\boldsymbol{B}(s)}(\boldsymbol{w}(s))\right\|^2\right).$$

By Lemma 12 and $\eta < \min\{\frac{1-\beta}{1+\frac{HN}{b\gamma^2}}, \frac{(1-\beta)^3\gamma^4 b}{2H^2N^3\beta^2}\}$, we further have

$$\mathbb{E}_{\mathcal{F}_{t+1}}[\mathcal{L}(\boldsymbol{u}(t+1))] = \mathbb{E}_{\{\boldsymbol{w}(s)\}_{s=1}^{t+1}}[\mathcal{L}(\boldsymbol{u}(t+1))]$$
$$\leq \mathbb{E}_{\mathcal{F}_t}\mathcal{L}(\boldsymbol{u}(t)) - \tilde{\eta}\left(1 - \left(\frac{1}{2} + \frac{HN}{2b\gamma^2}\right)\tilde{\eta}\right)\mathbb{E}_{\mathcal{F}_t}\|\nabla\mathcal{L}(\boldsymbol{w}(t))\|^2$$
$$+ \frac{1}{2}\frac{H^2N^3\tilde{\eta}^2\beta^2}{\gamma^4 b(1-\beta)}\left(\sum_{s=1}^{t-1}\beta^{t-1-s}\mathbb{E}_{\mathcal{F}_s}\|\nabla\mathcal{L}(\boldsymbol{w}(s))\|^2\right)$$
$$\leq \mathbb{E}_{\mathcal{F}_t}\mathcal{L}(\boldsymbol{u}(t)) - \frac{\tilde{\eta}}{2}\mathbb{E}_{\mathcal{F}_t}\|\nabla\mathcal{L}(\boldsymbol{w}(t))\|^2 + \frac{1-\beta}{4}\tilde{\eta}\left(\sum_{s=1}^{t-1}\beta^{t-1-s}\mathbb{E}_{\mathcal{F}_s}\|\nabla\mathcal{L}(\boldsymbol{w}(s))\|^2\right). \quad (37)$$

Summing Eq. (37) with $t$ from 1 to $T$ leads to

$$\mathbb{E}[\mathcal{L}(\boldsymbol{u}(T+1))] = \mathbb{E}_{\mathcal{F}_{T+1}}[\mathcal{L}(\boldsymbol{u}(T+1))]$$
$$\leq \mathcal{L}(\boldsymbol{u}(1)) - \sum_{t=1}^{T}\frac{\tilde{\eta}}{2}\mathbb{E}_{\mathcal{F}_t}\|\nabla\mathcal{L}(\boldsymbol{w}(t))\|^2 + \sum_{t=1}^{T}\frac{1-\beta}{4}\tilde{\eta}\left(\sum_{s=1}^{t-1}\beta^{t-1-s}\mathbb{E}_{\mathcal{F}_s}\|\nabla\mathcal{L}(\boldsymbol{w}(s))\|^2\right)$$
$$= \mathcal{L}(\boldsymbol{u}(1)) - \sum_{t=1}^{T}\frac{\tilde{\eta}}{2}\mathbb{E}_{\mathcal{F}_t}\|\nabla\mathcal{L}(\boldsymbol{w}(t))\|^2 + \sum_{s=1}^{T-1}\frac{1-\beta}{4}\tilde{\eta}\left(\sum_{t=s+1}^{T}\beta^{t-1-s}\mathbb{E}_{\mathcal{F}_s}\|\nabla\mathcal{L}(\boldsymbol{w}(s))\|^2\right)$$
$$\leq \mathcal{L}(\boldsymbol{u}(1)) - \sum_{t=1}^{T}\frac{\tilde{\eta}}{2}\mathbb{E}_{\mathcal{F}_t}\|\nabla\mathcal{L}(\boldsymbol{w}(t))\|^2 + \sum_{s=1}^{T-1}\frac{1}{4}\tilde{\eta}\left(\mathbb{E}_{\mathcal{F}_s}\|\nabla\mathcal{L}(\boldsymbol{w}(s))\|^2\right)$$
$$\leq \mathcal{L}(\boldsymbol{u}(1)) - \sum_{t=1}^{T}\frac{\tilde{\eta}}{4}\mathbb{E}_{\mathcal{F}_t}\|\nabla\mathcal{L}(\boldsymbol{w}(t))\|^2$$
$$= \mathcal{L}(\boldsymbol{u}(1)) - \sum_{t=1}^{T}\frac{\tilde{\eta}}{4}\mathbb{E}\|\nabla\mathcal{L}(\boldsymbol{w}(t))\|^2.$$

The proof is completed. $\qquad \square$

As $\mathcal{L}(\boldsymbol{u}(1))$ is upper bounded, we have the following corollary given by Lemma 4.

**Corollary 5.** *Let all conditions in Theorem 3 hold. Then, we have*

$$\sum_{t=1}^{\infty}\mathbb{E}\|\nabla\mathcal{L}(\boldsymbol{w}(t))\|^2 < \infty. \quad (38)$$

*Consequently,*

$$\sum_{t=1}^{\infty} \|\nabla \mathcal{L}(\boldsymbol{w}(t))\|^2 < \infty$$

*and*

$$\langle \boldsymbol{w}(t), \tilde{\boldsymbol{x}} \rangle \to \infty, \forall \tilde{\boldsymbol{x}} \in \tilde{\boldsymbol{S}}$$

*hold almost surely.*

*Proof.* By Lemma 4, we have for any $T > 1$,

$$\sum_{t=1}^{T} \frac{\tilde{\eta}}{4} \mathbb{E} \|\nabla \mathcal{L}(\boldsymbol{w}(t))\|^2 \leq \mathcal{L}(\boldsymbol{u}(1)) - \mathbb{E}[\mathcal{L}(\boldsymbol{u}(T+1))] \leq \mathcal{L}(\boldsymbol{u}(1)) < \infty,$$

which completes the proof of Eq. (38). The rest of claims follows immediately by Fubini's Theorem and Assumption 2.

The proof is completed. $\qquad\square$

### B.2.2 Bounding the orthogonal part

Similar to the case of GDM, we define $\tilde{\boldsymbol{w}}$ as the solution of Eq. (12) with $C_2 = \frac{\eta}{(1-\beta)N}$. We also let $\boldsymbol{n}(t)$ be given by Lemma 9, and define $\boldsymbol{r}(t)$ in this case as

$$\boldsymbol{r}(t) \triangleq \boldsymbol{w}(t) - \log(t)\hat{\boldsymbol{w}} - \tilde{\boldsymbol{w}} - \boldsymbol{n}(t). \tag{39}$$

As $\tilde{\boldsymbol{w}}$ is a constant vector, and $\|\boldsymbol{n}(t)\| \to 0$ as $t \to \infty$, we have $\boldsymbol{w}(t) - \log(t)\hat{\boldsymbol{w}}$ has bounded norm if and only if $\|\boldsymbol{r}(t)\|$ is upper bounded. Similar to the GDM case, we have the following equivalent condition of that $\|\boldsymbol{r}(t)\|$ is bounded.

**Lemma 17.** *Let all conditions in Theorem 3 hold. Then, $\|\boldsymbol{r}(t)\|$ is bounded almost surely if and only if function $g(t)$ is upper bounded almost surely, where $g : \mathbb{Z}^+ \to \mathbb{R}$ is defined as*

$$g(t) \triangleq \frac{1}{2} \|\boldsymbol{r}(t)\|^2 + \frac{\beta}{1-\beta} \langle \boldsymbol{r}(t), \boldsymbol{w}(t) - \boldsymbol{w}(t-1) \rangle - \frac{\beta}{1-\beta} \sum_{\tau=2}^{t} \langle \boldsymbol{r}(\tau) - \boldsymbol{r}(\tau-1), \boldsymbol{w}(\tau) - \boldsymbol{w}(\tau-1) \rangle. \tag{40}$$

*Proof.* To begin with, we prove that almost surely $|\sum_{\tau=2}^{t} \langle \boldsymbol{r}(\tau) - \boldsymbol{r}(\tau-1), \boldsymbol{w}(\tau) - \boldsymbol{w}(\tau-1) \rangle|$ is upper bounded for any $t$. By Corollary 5, we have almost surly

$$\sum_{t=1}^{\infty} \|\nabla \mathcal{L}(\boldsymbol{w}(t))\|^2 < \infty.$$

On the other hand, for any $\boldsymbol{w}$, we have

$$\|\nabla \mathcal{L}_{\boldsymbol{B}(t)}(\boldsymbol{w})\| = \frac{1}{b} \left\| \sum_{\tilde{\boldsymbol{x}} \in \mathcal{T}(\boldsymbol{B}(t))} \ell'(\langle \boldsymbol{w}, \tilde{\boldsymbol{x}} \rangle)\tilde{\boldsymbol{x}} \right\|$$

$$\leq -\frac{1}{b} \sum_{\tilde{\boldsymbol{x}} \in \mathcal{T}(\boldsymbol{B}(t))} \ell'(\langle \boldsymbol{w}, \tilde{\boldsymbol{x}} \rangle) < -\frac{1}{b} \sum_{\tilde{\boldsymbol{x}} \in \mathcal{T}(\boldsymbol{S})} \ell'(\langle \boldsymbol{w}, \tilde{\boldsymbol{x}} \rangle)$$

$$\leq -\frac{1}{b} \sum_{\tilde{\boldsymbol{x}} \in \mathcal{T}(\boldsymbol{S})} \ell'(\langle \boldsymbol{w}, \tilde{\boldsymbol{x}} \rangle)\langle \hat{\boldsymbol{w}}, \tilde{\boldsymbol{x}} \rangle \leq \frac{N}{b} \left\| \frac{1}{N} \sum_{\tilde{\boldsymbol{x}} \in \mathcal{T}(\boldsymbol{S})} \ell'(\langle \boldsymbol{w}, \tilde{\boldsymbol{x}} \rangle)\tilde{\boldsymbol{x}} \right\| \|\hat{\boldsymbol{w}}\|$$

$$= \frac{N}{b\gamma} \|\nabla \mathcal{L}(\boldsymbol{w})\| .$$

Therefore, we have almost surely,

$$\sum_{t=1}^{\infty} \left\| \nabla \mathcal{L}_{\boldsymbol{B}(t)}(\boldsymbol{w}(t)) \right\|^2 < \infty,$$

which further leads to almost surely

$$
\sum_{t=1}^{\infty} \|\boldsymbol{w}(t+1) - \boldsymbol{w}(t)\|^2 \leq \eta^2 \sum_{t=1}^{\infty} \left\| \sum_{s=1}^{t} \beta^{t-s} \nabla \mathcal{L}_{\boldsymbol{B}(s)}(\boldsymbol{w}(s)) \right\|^2
$$

$$
\leq \eta^2 \sum_{t=1}^{\infty} \left( \sum_{s=1}^{t} \beta^{t-s} \left\| \nabla \mathcal{L}_{\boldsymbol{B}(s)}(\boldsymbol{w}(s)) \right\| \right)^2
$$

$$
\leq \eta^2 \sum_{t=1}^{\infty} \left( \sum_{s=1}^{t} \beta^{t-s} \left\| \nabla \mathcal{L}_{\boldsymbol{B}(s)}(\boldsymbol{w}(s)) \right\|^2 \right) \left( \sum_{s=1}^{t} \beta^{t-s} \right)
$$

$$
\leq \frac{\eta^2}{1-\beta} \sum_{s=1}^{\infty} \left\| \nabla \mathcal{L}_{\boldsymbol{B}(s)}(\boldsymbol{w}(s)) \right\|^2 \sum_{t=s}^{\infty} \beta^{t-s}
$$

$$
= \frac{\eta^2}{1-\beta} \sum_{s=1}^{\infty} \left\| \nabla \mathcal{L}_{\boldsymbol{B}(s)}(\boldsymbol{w}(s)) \right\|^2 < \infty.
$$

By the definition of $\boldsymbol{r}(t)$ (Eq. (39)), we further have

$$
\left| \sum_{\tau=2}^{t} \langle \boldsymbol{r}(\tau) - \boldsymbol{r}(\tau-1), \boldsymbol{w}(\tau) - \boldsymbol{w}(\tau-1) \rangle \right|
$$

$$
\leq \sum_{\tau=2}^{t} |\langle \boldsymbol{r}(\tau) - \boldsymbol{r}(\tau-1), \boldsymbol{w}(\tau) - \boldsymbol{w}(\tau-1) \rangle|
$$

$$
= \sum_{\tau=2}^{t} \left| \left\langle \boldsymbol{w}(\tau) - \boldsymbol{w}(\tau-1) - \log\left(\frac{\tau+1}{\tau}\right) - (\boldsymbol{n}(\tau) - \boldsymbol{n}(\tau-1)), \boldsymbol{w}(\tau) - \boldsymbol{w}(\tau-1) \right\rangle \right|
$$

$$
\leq \sum_{\tau=2}^{t} \|\boldsymbol{w}(\tau) - \boldsymbol{w}(\tau-1)\|^2 + \sum_{\tau=2}^{t} \left| \left\langle -\log\left(\frac{\tau+1}{\tau}\right) - (\boldsymbol{n}(\tau) - \boldsymbol{n}(\tau-1)), \boldsymbol{w}(\tau) - \boldsymbol{w}(\tau-1) \right\rangle \right|
$$

$$
\leq \frac{3}{2} \sum_{\tau=2}^{t} \|\boldsymbol{w}(\tau) - \boldsymbol{w}(\tau-1)\|^2 + \frac{1}{2} \sum_{\tau=2}^{t} \left\| -\log\left(\frac{\tau+1}{\tau}\right) - (\boldsymbol{n}(\tau) - \boldsymbol{n}(\tau-1)) \right\|^2
$$

$$
\overset{(\star)}{\leq} \frac{3}{2} \sum_{\tau=2}^{t} \|\boldsymbol{w}(\tau) - \boldsymbol{w}(\tau-1)\|^2 + \frac{1}{2} \sum_{\tau=2}^{t} \mathcal{O}\left(\frac{1}{\tau}\right)^2 < \infty,
$$

where Inequality $(\star)$ is due to $\|\boldsymbol{n}(\tau) - \boldsymbol{n}(\tau-1)\| = \mathcal{O}(\frac{1}{\tau})$ and $\log\frac{\tau+1}{\tau} = \mathcal{O}(\frac{1}{\tau})$.

Therefore, $g(t)$ is upper bounded almost surely is equivalent to $\frac{1}{2}\|\boldsymbol{r}(t)\|^2 + \frac{\beta}{1-\beta}\langle \boldsymbol{r}(t), \boldsymbol{w}(t) - \boldsymbol{w}(t-1)\rangle$ is upper bounded, which can be shown to be equivalent with $\|\boldsymbol{r}(t)\|$ is bounded following the same routine as Lemma 2.

The proof is completed. $\qquad\square$

As the case of GDM, we only need to prove $g(t)$ is upper bounded to complete the proof of Theorem 3.

**Lemma 18.** *Let all conditions in Theorem 3 hold. We have $g(t)$ is upper bounded.*

*Proof.* Following the same routine as Lemma 2, we have

$$
g(t+1) - g(t)
$$

$$
= \frac{1}{2}\|\boldsymbol{r}(t+1) - \boldsymbol{r}(t)\|^2 + \langle \boldsymbol{r}(t), \boldsymbol{r}(t+1) - \boldsymbol{r}(t)\rangle + \left\langle \boldsymbol{r}(t), -\frac{\eta}{1-\beta}\nabla\mathcal{L}_{\boldsymbol{B}(t)}(\boldsymbol{w}(t)) - (\boldsymbol{w}(t+1) - \boldsymbol{w}(t)) \right\rangle,
$$

where $\sum_{t=1}^{\infty}\|\boldsymbol{r}(t+1) - \boldsymbol{r}(t)\|^2$ is upper bounded.

On the other hand, by the definition of $\boldsymbol{r}(t)$ (Eq. (39)), we have

$$\boldsymbol{r}(t+1) - \boldsymbol{r}(t)$$
$$= \boldsymbol{w}(t+1) - \boldsymbol{w}(t) - \log\left(\frac{t+1}{t}\right)\hat{\boldsymbol{w}} - \boldsymbol{n}(t+1) + \boldsymbol{n}(t),$$

while by Lemma 9,

$$\frac{N}{b}\frac{1}{t}\sum_{i:\tilde{\boldsymbol{x}}_i\in\mathcal{T}(\boldsymbol{B}(t)\cap\boldsymbol{S}_s)}\boldsymbol{v}_i\tilde{\boldsymbol{x}}_i = \log\left(\frac{t+1}{t}\right)\hat{\boldsymbol{w}} + \boldsymbol{n}(t+1) - \boldsymbol{n}(t).$$

Combining the above two equations, we further have

$$\boldsymbol{r}(t+1) - \boldsymbol{r}(t) = \boldsymbol{w}(t+1) - \boldsymbol{w}(t) - \frac{N}{bt}\sum_{i:\tilde{\boldsymbol{x}}_i\in\mathcal{T}(\boldsymbol{B}(t)\cap\boldsymbol{S}_s)}\boldsymbol{v}_i\tilde{\boldsymbol{x}}_i,$$

which further indicates

$$g(t+1) - g(t) = \frac{1}{2}\|\boldsymbol{r}(t+1) - \boldsymbol{r}(t)\|^2 + \left\langle \boldsymbol{r}(t), -\frac{\eta}{1-\beta}\nabla\mathcal{L}_{\boldsymbol{B}(t)}(\boldsymbol{w}(t)) - \frac{N}{bt}\sum_{i:\tilde{\boldsymbol{x}}_i\in\mathcal{T}(\boldsymbol{B}(t)\cap\boldsymbol{S}_s)}\boldsymbol{v}_i\tilde{\boldsymbol{x}}_i \right\rangle.$$

Therefore, we only need to prove $\sum_{t=1}^{\infty}\langle\boldsymbol{r}(t), -\frac{\eta}{1-\beta}\nabla\mathcal{L}(\boldsymbol{w}(t)) - \frac{N}{bt}\sum_{i:\tilde{\boldsymbol{x}}_i\in\mathcal{T}(\boldsymbol{B}(t)\cap\boldsymbol{S}_s)}\boldsymbol{v}_i\tilde{\boldsymbol{x}}_i\rangle < \infty$. By directly applying the form of $\nabla\mathcal{L}(\boldsymbol{w}(t))$, we have

$$\left\langle \boldsymbol{r}(t), -\frac{\eta}{1-\beta}\nabla\mathcal{L}_{\boldsymbol{B}(t)}(\boldsymbol{w}(t)) - \frac{N}{bt}\sum_{i:\tilde{\boldsymbol{x}}_i\in\mathcal{T}(\boldsymbol{B}(s)\cap\boldsymbol{S}_s)}\boldsymbol{v}_i\tilde{\boldsymbol{x}}_i \right\rangle$$

$$= \left\langle \boldsymbol{r}(t), -\frac{\eta}{(1-\beta)b}\sum_{i:\tilde{\boldsymbol{x}}_i\in\mathcal{T}(\boldsymbol{B}(s))}\ell'(\langle\boldsymbol{w}(t),\tilde{\boldsymbol{x}}_i\rangle)\tilde{\boldsymbol{x}}_i - \frac{N}{bt}\sum_{i:\tilde{\boldsymbol{x}}_i\in\mathcal{T}(\boldsymbol{B}(s)\cap\boldsymbol{S}_s)}\boldsymbol{v}_i\tilde{\boldsymbol{x}}_i \right\rangle$$

$$= \frac{\eta}{(1-\beta)b}\left\langle \boldsymbol{r}(t), -\sum_{i:\tilde{\boldsymbol{x}}_i\in\mathcal{T}(\boldsymbol{B}(s))}\ell'(\langle\boldsymbol{w}(t),\tilde{\boldsymbol{x}}_i\rangle)\tilde{\boldsymbol{x}}_i - \frac{N(1-\beta)}{\eta}\frac{1}{t}\sum_{i:\tilde{\boldsymbol{x}}_i\in\mathcal{T}(\boldsymbol{B}(s)\cap\boldsymbol{S}_s)}\boldsymbol{v}_i\tilde{\boldsymbol{x}}_i \right\rangle$$

$$= \frac{\eta}{(1-\beta)b}\left\langle \boldsymbol{r}(t), -\sum_{i:\tilde{\boldsymbol{x}}_i\in\mathcal{T}(\boldsymbol{B}(s))}\ell'(\langle\boldsymbol{w}(t),\tilde{\boldsymbol{x}}_i\rangle)\tilde{\boldsymbol{x}}_i - \frac{1}{t}\sum_{i:\tilde{\boldsymbol{x}}_i\in\mathcal{T}(\boldsymbol{B}(s)\cap\boldsymbol{S}_s)}e^{-\langle\tilde{\boldsymbol{w}},\tilde{\boldsymbol{x}}_i\rangle}\tilde{\boldsymbol{x}}_i \right\rangle$$

$$= \frac{\eta}{(1-\beta)b}\sum_{i:\tilde{\boldsymbol{x}}_i\in\mathcal{T}(\boldsymbol{B}(s)\cap\boldsymbol{S}_s)}\left(-\ell'(\langle\boldsymbol{w}(t),\tilde{\boldsymbol{x}}_i\rangle) - \frac{1}{t}e^{-\langle\tilde{\boldsymbol{w}},\tilde{\boldsymbol{x}}_i\rangle}\right)\langle\boldsymbol{r}(t),\tilde{\boldsymbol{x}}_i\rangle$$

$$+ \frac{\eta}{(1-\beta)b}\sum_{i:\tilde{\boldsymbol{x}}_i\in\mathcal{T}(\boldsymbol{B}(s)\cap\boldsymbol{S}_s^c)}\langle\boldsymbol{r}(t), -\ell'(\langle\boldsymbol{w}(t),\tilde{\boldsymbol{x}}_i\rangle)\tilde{\boldsymbol{x}}_i\rangle.$$

Let $A_6(t) = \sum_{i:\tilde{\boldsymbol{x}}_i\in\mathcal{T}(\boldsymbol{B}(s)\cap\boldsymbol{S}_s)}\left(-\ell'(\langle\boldsymbol{w}(t),\tilde{\boldsymbol{x}}_i\rangle) - \frac{1}{t}e^{-\langle\tilde{\boldsymbol{w}}(t),\tilde{\boldsymbol{x}}_i\rangle}\right)\langle\boldsymbol{r}(t),\tilde{\boldsymbol{x}}_i\rangle$, and $A_7(t) = \sum_{i:\tilde{\boldsymbol{x}}_i\in\mathcal{T}(\boldsymbol{B}(s)\cap\boldsymbol{S}_s^c)}\langle\boldsymbol{r}(t), -\ell'(\langle\boldsymbol{w}(t),\tilde{\boldsymbol{x}}_i\rangle)\tilde{\boldsymbol{x}}_i\rangle$. We will investigate these two terms respectively.

As $\langle\boldsymbol{w}(t),\tilde{\boldsymbol{x}}\rangle \to \infty$, $\forall\tilde{\boldsymbol{x}}\in\mathcal{T}(\boldsymbol{S})$, a.s., we have a.s., there exists a large enough time $t_0$, s.t., $\forall t \geq t_0$, $\forall\tilde{\boldsymbol{x}}\in\mathcal{T}(\boldsymbol{S})$,

$$-\ell'(\langle\boldsymbol{w}(t),\tilde{\boldsymbol{x}}\rangle) \leq (1 + e^{-\mu_+\langle\boldsymbol{w}(t),\tilde{\boldsymbol{x}}\rangle})e^{-\langle\boldsymbol{w}(t),\tilde{\boldsymbol{x}}\rangle},$$
$$-\ell'(\langle\boldsymbol{w}(t),\tilde{\boldsymbol{x}}_i\rangle) \geq (1 - e^{-\mu_-\langle\boldsymbol{w}(t),\tilde{\boldsymbol{x}}\rangle})e^{-\langle\boldsymbol{w}(t),\tilde{\boldsymbol{x}}\rangle},$$
$$\langle\tilde{\boldsymbol{x}},\boldsymbol{w}(t)\rangle > 0.$$

Therefore,

$$
\begin{aligned}
A_7(t) &\leq \sum_{i:\tilde{\boldsymbol{x}}_i \in \mathcal{T}(\boldsymbol{B}(s) \cap \boldsymbol{S}_s^c)} -\ell'(\langle \boldsymbol{w}(t), \tilde{\boldsymbol{x}}_i \rangle) \langle \boldsymbol{r}(t), \tilde{\boldsymbol{x}}_i \rangle \mathbb{1}_{\langle \boldsymbol{r}(t), \tilde{\boldsymbol{x}}_i \rangle \geq 0} \\
&\leq \sum_{i:\tilde{\boldsymbol{x}}_i \in \mathcal{T}(\boldsymbol{B}(s) \cap \boldsymbol{S}_s^c)} (1 + e^{-\mu_+ \langle \boldsymbol{w}(t), \tilde{\boldsymbol{x}}_i \rangle}) e^{-\langle \boldsymbol{w}(t), \tilde{\boldsymbol{x}}_i \rangle} \langle \boldsymbol{r}(t), \tilde{\boldsymbol{x}}_i \rangle \mathbb{1}_{\langle \boldsymbol{r}(t), \tilde{\boldsymbol{x}}_i \rangle \geq 0} \\
&\leq 2 \sum_{i:\tilde{\boldsymbol{x}}_i \in \mathcal{T}(\boldsymbol{B}(s) \cap \boldsymbol{S}_s^c)} e^{-\langle \boldsymbol{r}(t) + \log(t)\hat{\boldsymbol{w}} + \tilde{\boldsymbol{w}} + \boldsymbol{n}(t), \tilde{\boldsymbol{x}}_i \rangle} \langle \boldsymbol{r}(t), \tilde{\boldsymbol{x}}_i \rangle \mathbb{1}_{\langle \boldsymbol{r}(t), \tilde{\boldsymbol{x}}_i \rangle \geq 0} \\
&\overset{(\star)}{\leq} 2 \sum_{i:\tilde{\boldsymbol{x}}_i \in \mathcal{T}(\boldsymbol{B}(s) \cap \boldsymbol{S}_s^c)} \frac{1}{t^\theta} e^{-\langle \tilde{\boldsymbol{w}} + \boldsymbol{n}(t), \tilde{\boldsymbol{x}}_i \rangle} e^{-\langle \boldsymbol{r}(t), \tilde{\boldsymbol{x}}_i \rangle} \langle \boldsymbol{r}(t), \tilde{\boldsymbol{x}}_i \rangle \mathbb{1}_{\langle \boldsymbol{r}(t), \tilde{\boldsymbol{x}}_i \rangle \geq 0} \\
&\overset{(\dagger)}{\leq} \frac{2}{e} \frac{1}{t^\theta} \sum_{i:\tilde{\boldsymbol{x}}_i \in \mathcal{T}(\boldsymbol{B}(s) \cap \boldsymbol{S}_s^c)} e^{-\langle \tilde{\boldsymbol{w}} + \boldsymbol{n}(t), \tilde{\boldsymbol{x}}_i \rangle} \mathbb{1}_{\langle \boldsymbol{r}(t), \tilde{\boldsymbol{x}}_i \rangle \geq 0} \\
&\overset{(\circ)}{=} \mathcal{O}\left( \frac{1}{t^\theta} \right),
\end{aligned}
$$

where Inequality. $(\star)$ is due the definition of $\theta$ (Eq. (31)), Inequality. $(\dagger)$ is due to $e^{-\langle \boldsymbol{r}(t), \tilde{\boldsymbol{x}}_i \rangle} \langle \boldsymbol{r}(t), \tilde{\boldsymbol{x}}_i \rangle \leq e^{-1}$, and Eq. $(\circ)$ is due to $\lim_{t \to \infty} e^{-\langle \tilde{\boldsymbol{w}} + \boldsymbol{n}(t), \tilde{\boldsymbol{x}}_i \rangle} = e^{-\langle \tilde{\boldsymbol{w}}, \tilde{\boldsymbol{x}}_i \rangle}$. Thus,

$$
\sum_{t=1}^{\infty} A_7(t) < \infty.
$$

On the other hand, $A_6(t)$ can be rewritten as

$$
\begin{aligned}
A_6(t) = &\sum_{i:\tilde{\boldsymbol{x}}_i \in \mathcal{T}(\boldsymbol{B}(s) \cap \boldsymbol{S}_s)} \left( -\ell'(\langle \boldsymbol{w}(t), \tilde{\boldsymbol{x}}_i \rangle) - \frac{1}{t} e^{-\langle \tilde{\boldsymbol{w}}(t), \tilde{\boldsymbol{x}}_i \rangle} \right) \langle \boldsymbol{r}(t), \tilde{\boldsymbol{x}}_i \rangle \mathbb{1}_{\langle \boldsymbol{r}(t), \tilde{\boldsymbol{x}}_i \rangle \geq 0} \\
&+ \sum_{i:\tilde{\boldsymbol{x}}_i \in \mathcal{T}(\boldsymbol{B}(s) \cap \boldsymbol{S}_s)} \left( -\ell'(\langle \boldsymbol{w}(t), \tilde{\boldsymbol{x}}_i \rangle) - \frac{1}{t} e^{-\langle \tilde{\boldsymbol{w}}(t), \tilde{\boldsymbol{x}}_i \rangle} \right) \langle \boldsymbol{r}(t), \tilde{\boldsymbol{x}}_i \rangle \mathbb{1}_{\langle \boldsymbol{r}(t), \tilde{\boldsymbol{x}}_i \rangle < 0}.
\end{aligned}
$$

If $\langle \boldsymbol{r}(t), \tilde{\boldsymbol{x}}_i \rangle \geq 0$, we have for $\varepsilon < 0.5$,

$$
\begin{aligned}
&\left( -\ell'(\langle \boldsymbol{w}(t), \tilde{\boldsymbol{x}}_i \rangle) - \frac{1}{t} e^{-\langle \tilde{\boldsymbol{w}}(t), \tilde{\boldsymbol{x}}_i \rangle} \right) \langle \boldsymbol{r}(t), \tilde{\boldsymbol{x}}_i \rangle \\
&\leq \left( \left( 1 + e^{-\mu_+ \langle \boldsymbol{w}(t), \tilde{\boldsymbol{x}}_i \rangle} \right) e^{-\langle \boldsymbol{w}(t), \tilde{\boldsymbol{x}}_i \rangle} - \frac{1}{t} e^{-\langle \tilde{\boldsymbol{w}}(t), \tilde{\boldsymbol{x}}_i \rangle} \right) \langle \boldsymbol{r}(t), \tilde{\boldsymbol{x}}_i \rangle \\
&= \left( \left( 1 + e^{-\mu_+ \langle \boldsymbol{r}(t) + \log(t)\hat{\boldsymbol{w}} + \tilde{\boldsymbol{w}} + \boldsymbol{n}(t), \tilde{\boldsymbol{x}}_i \rangle} \right) e^{-\langle \boldsymbol{r}(t) + \log(t)\hat{\boldsymbol{w}} + \tilde{\boldsymbol{w}} + \boldsymbol{n}(t), \tilde{\boldsymbol{x}}_i \rangle} - \frac{1}{t} e^{-\langle \tilde{\boldsymbol{w}}(t), \tilde{\boldsymbol{x}}_i \rangle} \right) \langle \boldsymbol{r}(t), \tilde{\boldsymbol{x}}_i \rangle \\
&= \left( \left( 1 + e^{-\mu_+ \langle \boldsymbol{r}(t) + \log(t)\hat{\boldsymbol{w}} + \tilde{\boldsymbol{w}} + \boldsymbol{n}(t), \tilde{\boldsymbol{x}}_i \rangle} \right) e^{-\langle \boldsymbol{r}(t) + \boldsymbol{n}(t), \tilde{\boldsymbol{x}}_i \rangle} - 1 \right) \frac{1}{t} \langle \boldsymbol{r}(t), \tilde{\boldsymbol{x}}_i \rangle e^{-\langle \tilde{\boldsymbol{w}}(t), \tilde{\boldsymbol{x}}_i \rangle} \\
&\leq \left( \left( 1 + \frac{1}{t^{\mu_+}} e^{-\mu_+ \langle \tilde{\boldsymbol{w}} + \boldsymbol{n}(t), \tilde{\boldsymbol{x}}_i \rangle} \right) e^{-\langle \boldsymbol{r}(t) + \boldsymbol{n}(t), \tilde{\boldsymbol{x}}_i \rangle} - 1 \right) \frac{1}{t} \langle \boldsymbol{r}(t), \tilde{\boldsymbol{x}}_i \rangle e^{-\langle \tilde{\boldsymbol{w}}(t), \tilde{\boldsymbol{x}}_i \rangle} \\
&\overset{(\bullet)}{=} \left( \left( 1 + \mathcal{O}\left( \frac{1}{t^{\mu_+}} \right) \right) \left( 1 + \mathcal{O}\left( \frac{1}{t^{0.5 - \varepsilon}} \right) \right) e^{-\langle \boldsymbol{r}(t), \tilde{\boldsymbol{x}}_i \rangle} - 1 \right) \frac{1}{t} \langle \boldsymbol{r}(t), \tilde{\boldsymbol{x}}_i \rangle e^{-\langle \tilde{\boldsymbol{w}}(t), \tilde{\boldsymbol{x}}_i \rangle} \\
&= \left( e^{-\langle \boldsymbol{r}(t), \tilde{\boldsymbol{x}}_i \rangle} - 1 \right) \frac{1}{t} \langle \boldsymbol{r}(t), \tilde{\boldsymbol{x}}_i \rangle e^{-\langle \tilde{\boldsymbol{w}}(t), \tilde{\boldsymbol{x}}_i \rangle} + \frac{1}{t} \mathcal{O}\left( \frac{1}{t^{\min\{\mu_+, 0.5 - \varepsilon\}}} \right) e^{-\langle \boldsymbol{r}(t), \tilde{\boldsymbol{x}}_i \rangle} \langle \boldsymbol{r}(t), \tilde{\boldsymbol{x}}_i \rangle e^{-\langle \tilde{\boldsymbol{w}}(t), \tilde{\boldsymbol{x}}_i \rangle} \\
&\leq \frac{1}{t} \mathcal{O}\left( \frac{1}{t^{\min\{\mu_+, 0.5 - \varepsilon\}}} \right) e^{-\langle \boldsymbol{r}(t), \tilde{\boldsymbol{x}}_i \rangle} \langle \boldsymbol{r}(t), \tilde{\boldsymbol{x}}_i \rangle e^{-\langle \tilde{\boldsymbol{w}}(t), \tilde{\boldsymbol{x}}_i \rangle} \\
&\overset{(\diamond)}{=} \mathcal{O}\left( \frac{1}{t^{\min\{1 + \mu_+, 1.5 - \varepsilon\}}} \right),
\end{aligned}
$$

where Eq. $(\bullet)$ is due to $\boldsymbol{n}(t) = \mathcal{O}(\frac{1}{t^{0.5 - \varepsilon}})$, and Eq. $(\diamond)$ is due to $e^{-\langle \boldsymbol{r}(t), \tilde{\boldsymbol{x}}_i \rangle} \langle \boldsymbol{r}(t), \tilde{\boldsymbol{x}}_i \rangle \leq \frac{1}{e}$.

On the other hand, if $\langle \boldsymbol{r}(t), \tilde{\boldsymbol{x}}_i \rangle < 0$, we have

$$
\left( -\ell'(\langle \boldsymbol{w}(t), \tilde{\boldsymbol{x}}_i \rangle) - \frac{1}{t} e^{-\langle \tilde{\boldsymbol{w}}(t), \tilde{\boldsymbol{x}}_i \rangle} \right) \langle \boldsymbol{r}(t), \tilde{\boldsymbol{x}}_i \rangle
$$

$$
\leq \left( \left( 1 - e^{-\mu_- \langle \boldsymbol{w}(t), \tilde{\boldsymbol{x}}_i \rangle} \right) e^{-\langle \boldsymbol{w}(t), \tilde{\boldsymbol{x}}_i \rangle} - \frac{1}{t} e^{-\langle \tilde{\boldsymbol{w}}(t), \tilde{\boldsymbol{x}}_i \rangle} \right) \langle \boldsymbol{r}(t), \tilde{\boldsymbol{x}}_i \rangle
$$

$$
= \frac{1}{t} e^{-\langle \tilde{\boldsymbol{w}}, \tilde{\boldsymbol{x}}_i \rangle} \left( -1 + \left( 1 - e^{-\mu_- \langle \boldsymbol{w}(t), \tilde{\boldsymbol{x}}_i \rangle} \right) e^{-\langle \boldsymbol{r}(t) + \boldsymbol{n}(t), \tilde{\boldsymbol{x}}_i \rangle} \right) \langle \boldsymbol{r}(t), \tilde{\boldsymbol{x}}_i \rangle
$$

Specifically, if $\langle \boldsymbol{r}(t), \tilde{\boldsymbol{x}}_i \rangle \geq -t^{-0.5 \min\{\mu_-, 0.5\}}$,

$$
\left| \frac{1}{t} e^{-\langle \tilde{\boldsymbol{w}}, \tilde{\boldsymbol{x}}_i \rangle} \left( -1 + \left( 1 - e^{-\mu_- \langle \boldsymbol{w}(t), \tilde{\boldsymbol{x}}_i \rangle} \right) e^{-\langle \boldsymbol{r}(t) + \boldsymbol{n}(t), \tilde{\boldsymbol{x}}_i \rangle} \right) \langle \boldsymbol{r}(t), \tilde{\boldsymbol{x}}_i \rangle \right|
$$

$$
\leq \frac{1}{t^{1 + 0.5 \min\{\mu_-, 0.5\}}} e^{-\langle \tilde{\boldsymbol{w}}, \tilde{\boldsymbol{x}}_i \rangle} \left| -1 + \left( 1 - e^{-\mu_- \langle \boldsymbol{w}(t), \tilde{\boldsymbol{x}}_i \rangle} \right) e^{-\langle \boldsymbol{r}(t) + \boldsymbol{n}(t), \tilde{\boldsymbol{x}}_i \rangle} \right|
$$

$$
\overset{(\Box)}{=} \mathcal{O} \left( \frac{1}{t^{1 + 0.5 \min\{\mu_-, 0.5\}}} \right),
$$

where Eq. ($\Box$) is due to that as $\langle \boldsymbol{w}(t), \tilde{\boldsymbol{x}}_i \rangle \to \infty$ and $t^{-0.5 \min\{\mu_-, 0.5\}} \to 0$ as $t \to \infty$, there exists a large enough time $T$, s.t., $\forall t > T$, under the circumstance $0 > \langle \boldsymbol{r}(t), \tilde{\boldsymbol{x}}_i \rangle \geq -t^{-0.5 \min\{\mu_-, 0.5\}}$, $e^{-\langle \boldsymbol{r}(t) + \boldsymbol{n}(t), \tilde{\boldsymbol{x}}_i \rangle} < 1$ and $e^{-\mu_- \langle \boldsymbol{w}(t), \tilde{\boldsymbol{x}}_i \rangle} < 1$.

If $-2 \leq \langle \boldsymbol{r}(t), \tilde{\boldsymbol{x}}_i \rangle < -t^{-0.5 \min\{\mu_-, 0.5\}}$, then, for large enough $t$, $|\langle \tilde{\boldsymbol{x}}_i, \boldsymbol{n}(t) \rangle| < 2$, $1 - \frac{e^{\mu_- (-\langle \tilde{\boldsymbol{w}}, \tilde{\boldsymbol{x}}_i \rangle + 4)}}{t^{\mu_-}} > 0$, and

$$
\frac{1}{t} e^{-\langle \tilde{\boldsymbol{w}}, \tilde{\boldsymbol{x}}_i \rangle} \left( -1 + \left( 1 - e^{-\mu_- \langle \boldsymbol{w}(t), \tilde{\boldsymbol{x}}_i \rangle} \right) e^{-\langle \boldsymbol{r}(t) + \boldsymbol{n}(t), \tilde{\boldsymbol{x}}_i \rangle} \right) \langle \boldsymbol{r}(t), \tilde{\boldsymbol{x}}_i \rangle
$$

$$
= \frac{1}{t} e^{-\langle \tilde{\boldsymbol{w}}, \tilde{\boldsymbol{x}}_i \rangle} \left( -1 + \left( 1 - e^{-\mu_- \langle \boldsymbol{r}(t) + \log(t) \hat{\boldsymbol{w}} + \tilde{\boldsymbol{w}} + \boldsymbol{n}(t), \tilde{\boldsymbol{x}}_i \rangle} \right) e^{-\langle \boldsymbol{r}(t) + \boldsymbol{n}(t), \tilde{\boldsymbol{x}}_i \rangle} \right) \langle \boldsymbol{r}(t), \tilde{\boldsymbol{x}}_i \rangle
$$

$$
= \frac{1}{t} e^{-\langle \tilde{\boldsymbol{w}}, \tilde{\boldsymbol{x}}_i \rangle} \left( -1 + \left( 1 - \frac{e^{-\mu_- \langle \tilde{\boldsymbol{w}}, \tilde{\boldsymbol{x}}_i \rangle}}{t^{\mu_-}} e^{-\mu_- \langle \boldsymbol{r}(t) + \boldsymbol{n}(t), \tilde{\boldsymbol{x}}_i \rangle} \right) e^{-\langle \boldsymbol{r}(t) + \boldsymbol{n}(t), \tilde{\boldsymbol{x}}_i \rangle} \right) \langle \boldsymbol{r}(t), \tilde{\boldsymbol{x}}_i \rangle
$$

$$
\leq \frac{1}{t} e^{-\langle \tilde{\boldsymbol{w}}, \tilde{\boldsymbol{x}}_i \rangle} \left( -1 + \left( 1 - \frac{e^{\mu_- (-\langle \tilde{\boldsymbol{w}}, \tilde{\boldsymbol{x}}_i \rangle + 4)}}{t^{\mu_-}} \right) e^{-\mu_- \langle \boldsymbol{r}(t) + \boldsymbol{n}(t), \tilde{\boldsymbol{x}}_i \rangle} \right) \langle \boldsymbol{r}(t), \tilde{\boldsymbol{x}}_i \rangle
$$

$$
\leq \frac{1}{t} e^{-\langle \tilde{\boldsymbol{w}}, \tilde{\boldsymbol{x}}_i \rangle} \left( -1 + \left( 1 - \frac{e^{\mu_- (-\langle \tilde{\boldsymbol{w}}, \tilde{\boldsymbol{x}}_i \rangle + 4)}}{t^{\mu_-}} \right) (1 - \mu_- \langle \boldsymbol{r}(t) + \boldsymbol{n}(t), \tilde{\boldsymbol{x}}_i \rangle) \right) \langle \boldsymbol{r}(t), \tilde{\boldsymbol{x}}_i \rangle
$$

$$
\leq \frac{1}{t} e^{-\langle \tilde{\boldsymbol{w}}, \tilde{\boldsymbol{x}}_i \rangle} \left( -1 + \left( 1 - \frac{e^{\mu_- (-\langle \tilde{\boldsymbol{w}}, \tilde{\boldsymbol{x}}_i \rangle + 4)}}{t^{\mu_-}} \right) \left( 1 + \mu_- t^{-0.5 \min\{\mu_-, 0.5\}} - \mu_- \langle \boldsymbol{n}(t), \tilde{\boldsymbol{x}}_i \rangle \right) \right) \langle \boldsymbol{r}(t), \tilde{\boldsymbol{x}}_i \rangle
$$

$$
= \frac{1}{t} e^{-\langle \tilde{\boldsymbol{w}}, \tilde{\boldsymbol{x}}_i \rangle} \left( -1 + \left( 1 - \frac{e^{\mu_- (-\langle \tilde{\boldsymbol{w}}, \tilde{\boldsymbol{x}}_i \rangle + 4)}}{t^{\mu_-}} \right) \left( 1 + \mu_- t^{-0.5 \min\{\mu_-, 0.5\}} + \boldsymbol{o} \left( t^{-0.5 \min\{\mu_-, 0.5\}} \right) \right) \right) \langle \boldsymbol{r}(t), \tilde{\boldsymbol{x}}_i \rangle
$$

$$
= \frac{1}{t} e^{-\langle \tilde{\boldsymbol{w}}, \tilde{\boldsymbol{x}}_i \rangle} \left( -1 + 1 + \mu_- t^{-0.5 \min\{\mu_-, 0.5\}} + \boldsymbol{o} \left( t^{-0.5 \min\{\mu_-, 0.5\}} \right) \right) \langle \boldsymbol{r}(t), \tilde{\boldsymbol{x}}_i \rangle < 0.
$$

If $-2 > \langle \boldsymbol{r}(t), \tilde{\boldsymbol{x}}_i \rangle$, then for large enough time $t$, $e^{-\langle \boldsymbol{r}(t) + \boldsymbol{n}(t), \tilde{\boldsymbol{x}}_i \rangle} \geq e^{\frac{3}{2}}$, $1 - e^{-\mu_- \langle \boldsymbol{w}(t), \tilde{\boldsymbol{x}}_i \rangle} \geq e^{-\frac{1}{2}}$, and

$$
\frac{1}{t} e^{-\langle \tilde{\boldsymbol{w}}, \tilde{\boldsymbol{x}}_i \rangle} \left( -1 + \left( 1 - e^{-\mu_- \langle \boldsymbol{w}(t), \tilde{\boldsymbol{x}}_i \rangle} \right) e^{-\langle \boldsymbol{r}(t) + \boldsymbol{n}(t), \tilde{\boldsymbol{x}}_i \rangle} \right) \langle \boldsymbol{r}(t), \tilde{\boldsymbol{x}}_i \rangle
$$

$$
\leq \frac{1}{t} e^{-\langle \tilde{\boldsymbol{w}}, \tilde{\boldsymbol{x}}_i \rangle} (-1 + e) \langle \boldsymbol{r}(t), \tilde{\boldsymbol{x}}_i \rangle < 0.
$$

Conclusively, if $\langle \boldsymbol{r}(t), \tilde{\boldsymbol{x}}_i \rangle < 0$, for large enough $t$, we have

$$
\left( -\ell'(\langle \boldsymbol{w}(t), \tilde{\boldsymbol{x}}_i \rangle) - \frac{1}{t} e^{-\langle \tilde{\boldsymbol{w}}(t), \tilde{\boldsymbol{x}}_i \rangle} \right) \langle \boldsymbol{r}(t), \tilde{\boldsymbol{x}}_i \rangle \leq \mathcal{O} \left( \frac{1}{t^{1 + 0.5 \min\{\mu_-, 0.5\}}} \right),
$$

which further indicates, for large enough $t$, we have

$$A_6(t) \leq \max \left\{ \mathcal{O}\left(\frac{1}{t^{1+0.5\min\{\mu_-, 0.5\}}}\right), \mathcal{O}\left(\frac{1}{t^{\min\{1+\mu_+, 1.5-\varepsilon\}}}\right) \right\},$$

which indicates

$$\sum_{t=1}^{\infty} A_6(t) < \infty.$$

Therefore,

$$\sum_{t=1}^{\infty} (g(t+1) - g(t))$$

$$= \sum_{t=1}^{\infty} \left( \frac{1}{2} \|\boldsymbol{r}(t+1) - \boldsymbol{r}(t)\|^2 + \left\langle \boldsymbol{r}(t), -\frac{\eta}{1-\beta} \nabla \mathcal{L}_{\boldsymbol{B}(t)}(\boldsymbol{w}(t)) - \frac{N}{bt} \sum_{i:\tilde{\boldsymbol{x}}_i \in \mathcal{T}(\boldsymbol{B}(t) \cap \boldsymbol{S}_s)} \boldsymbol{v}_i \tilde{\boldsymbol{x}}_i \right\rangle \right)$$

$$= \sum_{t=1}^{\infty} \left( \frac{1}{2} \|\boldsymbol{r}(t+1) - \boldsymbol{r}(t)\|^2 + \frac{\eta}{1-\beta} A_6(t) + \frac{\eta}{1-\beta} A_7(t) \right)$$

$$< \infty.$$

The proof is completed. □

### B.2.3 Explanation for Proper Lyapunov Function

Based on the success of applying Lyapunov function $\mathcal{L}(\boldsymbol{w}(t)) + \frac{\beta}{2\eta}\|\boldsymbol{w}(t) - \boldsymbol{w}(t-1)\|^2$ to analyze gradient descent with momentum, it is natural to try to extend this routine to analyze stochastic gradient descent with momentum. However, in this section, we will show such Lyapunov function is not proper to analyze SGDM as this will put constraints on the range of the momentum rate $\beta$. Specifically, at any step $t$, since the loss $\mathcal{L}$ is $H$ smooth at $\boldsymbol{w}(t)$, we can expand the loss $\mathcal{L}$ in the same way as the GDM case:

$$\mathcal{L}(\boldsymbol{w}(t+1)) \leq \mathcal{L}(\boldsymbol{w}(t)) + \langle \boldsymbol{w}(t+1) - \boldsymbol{w}(t), \nabla \mathcal{L}(\boldsymbol{w}(t)) \rangle + \frac{H}{2}\|\boldsymbol{w}(t+1) - \boldsymbol{w}(t)\|^2.$$

By taking expectation with respect to $\boldsymbol{w}(t+1)$ conditioning on $\{\boldsymbol{w}(s)\}_{s=1}^{t}$ for both sides, we further obtain

$$\mathbb{E}_{\boldsymbol{w}(t+1)}\left[\mathcal{L}(\boldsymbol{w}(t+1)) \big| \{\boldsymbol{w}(s)\}_{s=1}^{t}\right]$$

$$\leq \mathcal{L}(\boldsymbol{w}(t)) + \langle \mathbb{E}_{\boldsymbol{w}(t+1)}\left[\boldsymbol{w}(t+1) - \boldsymbol{w}(t) \big| \{\boldsymbol{w}(s)\}_{s=1}^{t}\right], \nabla \mathcal{L}(\boldsymbol{w}(t)) \rangle + \frac{H}{2} \mathbb{E}_{\boldsymbol{w}(t+1)}\left[\|\boldsymbol{w}(t+1) - \boldsymbol{w}(t)\|^2 \big| \{\boldsymbol{w}(s)\}_{s=1}^{t}\right]$$

$$\overset{(\star)}{=} \mathcal{L}(\boldsymbol{w}(t)) + \frac{1}{\eta} \langle \mathbb{E}_{\boldsymbol{w}(t+1)}\left[\boldsymbol{w}(t+1) - \boldsymbol{w}(t) \big| \{\boldsymbol{w}(s)\}_{s=1}^{t}\right], \beta\left(\boldsymbol{w}(t) - \boldsymbol{w}(t-1)\right) - \mathbb{E}_{\boldsymbol{w}(t+1)}\left[\boldsymbol{w}(t+1) - \boldsymbol{w}(t) \big| \{\boldsymbol{w}(s)\}_{s=1}^{t}\right] \rangle$$

$$+ \frac{H}{2} \mathbb{E}_{\boldsymbol{w}(t+1)}\left[\|\boldsymbol{w}(t+1) - \boldsymbol{w}(t)\|^2 \big| \{\boldsymbol{w}(s)\}_{s=1}^{t}\right]$$

$$= \mathcal{L}(\boldsymbol{w}(t)) + \frac{\beta}{\eta} \langle \left(\boldsymbol{w}(t) - \boldsymbol{w}(t-1)\right), \mathbb{E}_{\boldsymbol{w}(t+1)}\left[\boldsymbol{w}(t+1) - \boldsymbol{w}(t) \big| \{\boldsymbol{w}(s)\}_{s=1}^{t}\right] \rangle$$

$$+ \frac{H}{2} \mathbb{E}_{\boldsymbol{w}(t+1)}\left[\|\boldsymbol{w}(t+1) - \boldsymbol{w}(t)\|^2 \big| \{\boldsymbol{w}(s)\}_{s=1}^{t}\right] - \frac{1}{\eta} \left\|\mathbb{E}_{\boldsymbol{w}(t+1)}\left[\boldsymbol{w}(t+1) - \boldsymbol{w}(t) \big| \{\boldsymbol{w}(s)\}_{s=1}^{t}\right]\right\|^2$$

$$\leq \mathcal{L}(\boldsymbol{w}(t)) + \frac{\beta}{2\eta} \|\boldsymbol{w}(t) - \boldsymbol{w}(t-1)\|^2 + \frac{\beta}{2\eta} \left\|\mathbb{E}_{\boldsymbol{w}(t+1)}\left[\boldsymbol{w}(t+1) - \boldsymbol{w}(t) \big| \{\boldsymbol{w}(s)\}_{s=1}^{t}\right]\right\|^2$$

$$+ \frac{H}{2} \mathbb{E}_{\boldsymbol{w}(t+1)}\left[\|\boldsymbol{w}(t+1) - \boldsymbol{w}(t)\|^2 \big| \{\boldsymbol{w}(s)\}_{s=1}^{t}\right] - \frac{1}{\eta} \left\|\mathbb{E}_{\boldsymbol{w}(t+1)}\left[\boldsymbol{w}(t+1) - \boldsymbol{w}(t) \big| \{\boldsymbol{w}(s)\}_{s=1}^{t}\right]\right\|^2,$$

where Eq. $(\star)$ is becasue $\mathbb{E}_{\boldsymbol{w}(t+1)}\left[\boldsymbol{w}(t+1) - \boldsymbol{w}(t) \big| \{\boldsymbol{w}(s)\}_{s=1}^{t}\right] = -\eta \nabla \mathcal{L}(\boldsymbol{w}(t)) + \beta\left(\boldsymbol{w}(t) - \boldsymbol{w}(t-1)\right)$ due to the definition of SGDM (Eq. (1)). Rearranging the above inequal-

ity and taking expectations of both sides with respect to $\{w(s)\}_{s=1}^{t}$ leads to

$$
\mathbb{E}_{w(t+1)}\left[\mathcal{L}(w(t+1))\right] + \frac{2-\beta}{2\eta}\mathbb{E}_{\{w(s)\}_{s=1}^{t}}\left\|\mathbb{E}_{w(t+1)}\left[w(t+1)-w(t)\left|\{w(s)\}_{s=1}^{t}\right]\right\|^{2}\right.
$$

$$
-\frac{H}{2}\mathbb{E}_{\{w(s)\}_{s=1}^{t+1}}\left[\|w(t+1)-w(t)\|^{2}\right]
$$

$$
\leq \mathbb{E}_{w(t)}\mathcal{L}(w(t)) + \frac{\beta}{2\eta}\mathbb{E}_{\{w(s)\}_{s=1}^{t}}\|w(t)-w(t-1)\|^{2}. \tag{41}
$$

On the other hand, we wish to obtain some positive constant $\alpha$ from Eq. (41), such that (at least),

$$
\mathbb{E}_{w(t+1)}\left[\mathcal{L}(w(t+1))\right] + \alpha\mathbb{E}_{\{w(s)\}_{s=1}^{t}}\|w(t+1)-w(t)\|^{2}.
$$

$$
\leq \mathbb{E}_{w(t)}\mathcal{L}(w(t)) + \alpha\mathbb{E}_{\{w(s)\}_{s=1}^{t}}\|w(t)-w(t-1)\|^{2}, \tag{42}
$$

which requires to lower bound $\mathbb{E}_{\{w(s)\}_{s=1}^{t}}\left\|\mathbb{E}_{w(t+1)}\left[w(t+1)-w(t)\left|\{w(s)\}_{s=1}^{t}\right]\right\|^{2}\right.$ by $\mathbb{E}_{\{w(s)\}_{s=1}^{t+1}}\|w(t+1)-w(t)\|^{2}$. However, in general cases, $\mathbb{E}_{\{w(s)\}_{s=1}^{t}}\left\|\mathbb{E}_{w(t+1)}\left[w(t+1)-w(t)\left|\{w(s)\}_{s=1}^{t}\right]\right\|^{2}\right.$ is only upper bounded by $\mathbb{E}_{\{w(s)\}_{s=1}^{t+1}}\|w(t+1)-w(t)\|^{2}$ (Holder's Inequality), although in our case, $\left\|\mathbb{E}_{w(t+1)}\left[w(t+1)-w(t)\left|\{w(s)\}_{s=1}^{t}\right]\right\|^{2}\right.$ can be bounded as

$$
\left\|\mathbb{E}_{w(t+1)}\left[w(t+1)-w(t)\left|\{w(s)\}_{s=1}^{t}\right]\right\|^{2}\right.
$$

$$
= \left\|-\eta\nabla\mathcal{L}(w(t)) + \beta\left(w(t)-w(t-1)\right)\right\|^{2}
$$

$$
= \left\|-\eta\nabla\mathcal{L}(w(t))\right\|^{2} + \left\|\beta\left(w(t)-w(t-1)\right)\right\|^{2} + 2\beta\eta\langle w(t)-w(t-1), -\nabla\mathcal{L}(w(t))\rangle,
$$

while $\mathbb{E}_{w(t+1)}\left[\|w(t+1)-w(t)\|^{2}\left|\{w(s)\}_{s=1}^{t}\right]\right.$ can be calculated as

$$
\mathbb{E}_{w(t+1)}\left[\|w(t+1)-w(t)\|^{2}\left|\{w(s)\}_{s=1}^{t}\right]\right.
$$

$$
= \mathbb{E}_{B(t)}\left\|-\eta\nabla\mathcal{L}_{B(t)}(w(t)) + \beta(w(t)-w(t-1))\right\|^{2}
$$

$$
= \mathbb{E}_{B(t)}\left\|-\eta\nabla\mathcal{L}_{B(t)}(w(t))\right\|^{2} + \|\beta(w(t)-w(t-1))\|^{2} + 2\eta\beta\mathbb{E}_{B(t)}\langle-\nabla\mathcal{L}_{B(t)}(w(t)), w(t)-w(t-1)\rangle
$$

$$
= \mathbb{E}_{B(t)}\left\|-\eta\nabla\mathcal{L}_{B(t)}(w(t))\right\|^{2} + \|\beta(w(t)-w(t-1))\|^{2} + 2\eta\beta\langle-\nabla\mathcal{L}(w(t)), w(t)-w(t-1)\rangle
$$

$$
\leq \frac{N}{b\gamma^{2}}\left\|-\eta\nabla\mathcal{L}(w(t))\right\|^{2} + \|\beta(w(t)-w(t-1))\|^{2} + 2\eta\beta\langle-\nabla\mathcal{L}(w(t)), w(t)-w(t-1)\rangle
$$

$$
\leq \frac{N}{b\gamma^{2}}\left(\left\|-\eta\nabla\mathcal{L}(w(t))\right\|^{2} + \|\beta(w(t)-w(t-1))\|^{2} + 2\eta\beta\langle-\nabla\mathcal{L}(w(t)), w(t)-w(t-1)\rangle\right)
$$

$$
= \frac{N}{b\gamma^{2}}\left\|\mathbb{E}_{w(t+1)}\left[w(t+1)-w(t)\left|\{w(s)\}_{s=1}^{t}\right]\right\|^{2}. \tag{43}
$$

By Eqs. (41) and (43), we have that to ensure Eq. (42), it is required that

$$
\frac{2-\beta}{2\eta}\frac{b\gamma^{2}}{N} - \frac{H}{2} \leq \frac{\beta}{2\eta},
$$

which puts constraint on $\beta$ that

$$
\beta \leq \frac{2\frac{b}{N}\gamma^{2} - H\eta}{1 + \frac{b}{N}\gamma^{2}}.
$$

Specifically, $\beta < \frac{2\frac{b}{N}\gamma^{2}}{1+\frac{b}{N}\gamma^{2}}$, which approaches 0 when $\gamma$ approaches 0, and constrains $\beta$ in a small range.

## C   IMPLICIT BIAS OF ADAM

This section collects the proof of the convergent direction of Adam, i.e., Theorem 4. The methodology of this section bears great similarity with GDM, although the preconditioner of Adam requires specific treatment for analysis. The proof is still divided into two stages: (1). we first prove the sum of squared gradients along the trajectory is finite. Additionally, we prove the convergent rate of loss is $\mathcal{O}(\frac{1}{t})$; (2). we prove $\boldsymbol{w}(t) - \log(t)\hat{\boldsymbol{w}}$ has bounded norm. Before we present these two stages of proof, we will first give the required range of $\eta$ for which Theorem 3 holds.

### C.1   CHOICE OF LEARNING RATE

Let $H_{s_0}$ be the smooth parameter over $[s_0, \infty)$ given by Assumption 3. (D). Let $\beta_2 = (c\beta_1)^4$ $(c > 1)$. The "sufficiently small learning rate" in Theorem 3 means

$$\eta \leq \frac{\sqrt{\varepsilon}\inf_{t\geq 2}\left(\frac{1-\beta_1^t}{1-\beta_1} - \frac{1-\beta_1^{t-1}}{c(1-\beta_1)}\frac{1-(c\beta_1)^t}{1-(c\beta_1)^{t-1}}\right)}{H_{\ell^{-1}((1-c\beta_1)^{-1}N\mathcal{L}(\boldsymbol{w}(1)))}}.$$

To ensure $\eta$ is well-defined, we need to prove

$$\inf_{t\geq 2}\left(\frac{1-\beta_1^t}{1-\beta_1} - \frac{1-\beta_1^{t-1}}{c(1-\beta_1)}\frac{1-(c\beta_1)^t}{1-(c\beta_1)^{t-1}}\right) > 0,$$

and we introduce the following technical lemma:

**Lemma 19.** *Define $f_t(x) = \frac{1-x^t}{x(1-x^{t-1})}$, $\forall t \in \mathbb{Z}, t \geq 2$. We have $f_t(x)$ is decreasing with respect to $x$. Furthermore, for any $x \in [0, 1)$, we have*

$$f(x) \geq \sqrt[4]{f(x^4)}. \tag{44}$$

*Proof.* First of all, by definition,

$$f(x) = \frac{1-x^t}{x-x^t} = 1 + \frac{1-x}{x-x^t} = 1 + \frac{1-x}{x(1-x^{t-1})} = 1 + \frac{1}{x(1+x+\cdots+x^{t-2})}$$

is monotonously decreasing as $0 \leq x < 1$. Secondly, Eq. (44) is equivalent to

$$\frac{(1-x^t)^4}{\beta_1^4(1-x^{t-1})^4} \geq \frac{(1-x^{4t})}{x^4(1-x^{4(t-1)})}$$

$$\Longleftrightarrow \frac{(1-x^t)^3}{(1-x^{t-1})^3} \geq \frac{(1+x^t)(1+x^{2t})}{(1+x^{t-1})(1+x^{2(t-1)})}.$$

The left side of the above inequality is no smaller than 1, while the right side is no larger than 1, which completes the proof. $\qquad\square$

We are now ready to prove $\eta$ is well-defined. First of all, for every $t$, we have

$$\frac{1-\beta_1^t}{1-\beta_1} - \frac{1-\beta_1^{t-1}}{c(1-\beta_1)}\frac{1-(c\beta_1)^t}{1-(c\beta_1)^{t-1}}$$

$$= \frac{\beta_1(1-\beta_1^{t-1})}{1-\beta_1}\left(\frac{1-\beta_1^t}{\beta_1(1-\beta_1^{t-1})} - \frac{1-(c\beta_1)^t}{(c\beta_1)(1-(c\beta_1)^{t-1})}\right)$$

$$\overset{(\star)}{=} \frac{\beta_1(1-\beta_1^{t-1})}{1-\beta_1}\left(f_t(\beta_1) - f_t(c\beta_1)\right) > 0, \tag{45}$$

where Eq. $(\star)$ is by Lemma 19 and $c\beta_1 = \sqrt[4]{\beta_2} < 1$.

On the other hand, we have

$$\lim_{t\to\infty}\left(\frac{1-\beta_1^t}{1-\beta_1} - \frac{1-\beta_1^{t-1}}{c(1-\beta_1)}\frac{1-(c\beta_1)^t}{1-(c\beta_1)^{t-1}}\right) = \left(1 - \frac{1}{c}\right)\frac{1}{1-\beta_1}. \tag{46}$$

By Eq. (45) and Eq. (46), we obtain $\frac{1-\beta_1^t}{1-\beta_1} - \frac{1-\beta_1^{t-1}}{c(1-\beta_1)}\frac{1-(c\beta_1)^t}{1-(c\beta_1)^{t-1}}$ is lower bounded by some positive constant across $t$, and $\eta$ is well defined.

## C.2 Sum of gradients along the trajectory is bounded

We start with the following lemma, which indicates $\mathcal{L}(\boldsymbol{w}(t)) + \|\sqrt[4]{\varepsilon \mathbb{1}_d + \hat{\nu}(t)} \odot (\boldsymbol{w}(t) - \boldsymbol{w}(t-1))\|^2$ is a proper Lyapunov function for Adam.

*Proof of Lemma 5.* We start with the case $t = 1$. To begin with, we have $\mathcal{L}$ is $H_{\ell^{-1}(N\mathcal{L}(\boldsymbol{w}(1)))}$ smooth around $\boldsymbol{w}(1)$. By definition $H_x$ is non-increasing with respect to $x$, and since $\ell^{-1}$ is also non-increasing, we have

$$H_{\ell^{-1}(N\mathcal{L}(\boldsymbol{w}(1)))} \leq H_{\ell^{-1}(\frac{1}{1-c\beta_1}N\mathcal{L}(\boldsymbol{w}(1)))},$$

which further indicates when $\alpha$ is small enough,

$$\mathcal{L}(\boldsymbol{w}(1+\alpha)) \overset{(\star)}{\leq} \mathcal{L}(\boldsymbol{w}(1)) + \alpha\langle \nabla\mathcal{L}(\boldsymbol{w}(1)), \boldsymbol{w}(2) - \boldsymbol{w}(1)\rangle + \frac{L}{2}\alpha^2\|\boldsymbol{w}(2) - \boldsymbol{w}(1)\|$$

$$= \mathcal{L}(\boldsymbol{w}(1)) - \alpha\left\langle \nabla\mathcal{L}(\boldsymbol{w}(1)), \eta\frac{1}{\sqrt{\varepsilon\mathbb{1}_d + \hat{\nu}(1)}} \odot \nabla\mathcal{L}(\boldsymbol{w}(1))\right\rangle + \boldsymbol{o}(\alpha^2)$$

$$\leq \mathcal{L}(\boldsymbol{w}(1)) - \frac{1}{2\eta}\alpha^2\left\|\sqrt[4]{\varepsilon\mathbb{1}_d + \hat{\nu}(1)} \odot (\boldsymbol{w}(2) - \boldsymbol{w}(t))\right\|^2,$$

where in Eq. $(\star)$ we denote $L \overset{\triangle}{=} H_{\ell^{-1}(\frac{1}{1-c\beta_1}N\mathcal{L}(\boldsymbol{w}(1)))}$, and the last inequality is due to $\frac{1}{2\eta}\alpha^2$ $\left\|\sqrt[4]{\varepsilon\mathbb{1}_d + \hat{\nu}(1)} \odot (\boldsymbol{w}(2) - \boldsymbol{w}(t))\right\|^2 = \boldsymbol{o}(\alpha^2)$, and $\left\langle \nabla\mathcal{L}(\boldsymbol{w}(1)), \eta\frac{1}{\sqrt{\varepsilon\mathbb{1}_d + \hat{\nu}(1)}} \odot \nabla\mathcal{L}(\boldsymbol{w}(1))\right\rangle$ is positive.

Now if there exists an $\alpha \in (0,1)$, such that Eq. (10) fails, we denote $\alpha^* = \inf\{\alpha : Eq.(10)\ fails\ for\ 1 + \alpha\}$. We have $\alpha^* > 0$, and the equality in Eq. (10) holds for $1 + \alpha^*$. Therefore, we have for any $\alpha \in (0, \alpha^*)$,

$$\mathcal{L}(\boldsymbol{w}(1+\alpha)) \leq \mathcal{L}(\boldsymbol{w}(1+\alpha)) + \frac{1}{2\eta}\alpha^2\left\|\sqrt[4]{\varepsilon\mathbb{1}_d + \hat{\nu}(1)} \odot (\boldsymbol{w}(2) - \boldsymbol{w}(t))\right\|^2 \leq \mathcal{L}(\boldsymbol{w}(1)),$$

which by Lemma 10 leads to $\mathcal{L}$ is $H_{\ell^{-1}(N\mathcal{L}(\boldsymbol{w}(1)))}$ smooth (thus $L$ smooth) over the set $\{\boldsymbol{w}(1+\alpha) : \alpha \in [0, \alpha^*]\}$, and

$$\mathcal{L}(\boldsymbol{w}(1+\alpha^*))$$

$$\leq \mathcal{L}(\boldsymbol{w}(1)) + \alpha^*\langle \nabla\mathcal{L}(\boldsymbol{w}(1)), \boldsymbol{w}(2) - \boldsymbol{w}(1)\rangle + \frac{L}{2}(\alpha^*)^2\|\boldsymbol{w}(2) - \boldsymbol{w}(1)\|^2$$

$$= \mathcal{L}(\boldsymbol{w}(1)) - \alpha^*\left\langle \frac{1}{\eta}\sqrt{\varepsilon\mathbb{1}_d + \hat{\nu}(1)} \odot (\boldsymbol{w}(2) - \boldsymbol{w}(1)), \boldsymbol{w}(2) - \boldsymbol{w}(1)\right\rangle + \frac{L}{2}(\alpha^*)^2\|\boldsymbol{w}(2) - \boldsymbol{w}(1)\|^2$$

$$= \mathcal{L}(\boldsymbol{w}(1)) - \alpha^*\frac{1}{\eta}\left\|\sqrt[4]{\varepsilon\mathbb{1}_d + \hat{\nu}(1)} \odot (\boldsymbol{w}(2) - \boldsymbol{w}(1))\right\|^2 + \frac{L}{2}(\alpha^*)^2\left\|\frac{1}{\sqrt[4]{\varepsilon\mathbb{1}_d + \hat{\nu}(1)}} \odot \sqrt[4]{\varepsilon\mathbb{1}_d + \hat{\nu}(1)} \odot (\boldsymbol{w}(2) - \boldsymbol{w}(1))\right\|^2$$

$$\leq \mathcal{L}(\boldsymbol{w}(1)) - \alpha^*\frac{1}{\eta}\left\|\sqrt[4]{\varepsilon\mathbb{1}_d + \hat{\nu}(1)} \odot (\boldsymbol{w}(2) - \boldsymbol{w}(1))\right\|^2 + \frac{L}{2\sqrt{\varepsilon}}(\alpha^*)^2\left\|\sqrt[4]{\varepsilon\mathbb{1}_d + \hat{\nu}(1)} \odot (\boldsymbol{w}(2) - \boldsymbol{w}(1))\right\|^2$$

$$< \mathcal{L}(\boldsymbol{w}(1)) - (\alpha^*)^2\frac{1}{\eta}\left\|\sqrt[4]{\varepsilon\mathbb{1}_d + \hat{\nu}(1)} \odot (\boldsymbol{w}(2) - \boldsymbol{w}(1))\right\|^2 + \frac{L}{2\sqrt{\varepsilon}}(\alpha^*)^2\left\|\sqrt[4]{\varepsilon\mathbb{1}_d + \hat{\nu}(1)} \odot (\boldsymbol{w}(2) - \boldsymbol{w}(1))\right\|^2$$

$$\leq \mathcal{L}(\boldsymbol{w}(1)) - (\alpha^*)^2\frac{1}{2\eta}\left\|\sqrt[4]{\varepsilon\mathbb{1}_d + \hat{\nu}(1)} \odot (\boldsymbol{w}(2) - \boldsymbol{w}(1))\right\|^2, \tag{47}$$

where the second-to-last inequality is due to $\|\boldsymbol{w}(2) - \boldsymbol{w}(1)\| > 0$ (by Lemma 13) and $\alpha^* > (\alpha^*)^2$, while the last inequality is due to

$$\eta \leq \frac{\sqrt{\varepsilon}\inf_{t\geq 2}\left(\frac{1-\beta_1^t}{1-\beta_1} - \frac{1-\beta_1^{t-1}}{c(1-\beta_1)}\frac{1-(c\beta_1)^t}{1-(c\beta_1)^{t-1}}\right)}{L} \leq \frac{\sqrt{\varepsilon}\left(\frac{1-\beta_1^2}{1-\beta_1} - \frac{1-\beta_1}{c(1-\beta_1)}\frac{1-(c\beta_1)^2}{1-(c\beta_1)}\right)}{L}$$

$$= \frac{\sqrt{\varepsilon}\left(1 + \beta_1 - \frac{1+(c\beta_1)}{c}\right)}{L} = \frac{\sqrt{\varepsilon}\left(1 - \frac{1}{c}\right)}{L} < \frac{\sqrt{\varepsilon}}{L}.$$

Eq. (47) contradicts the fact that the equality in Eq. (10) holds for $1 + \alpha^*$, which completes the proof of $t = 1$.

If $t \geq 2$, following the similar routine as $t = 1$, we also prove Eq. (10) by reduction to absurdity. If there exist $t$ and $\alpha$ such that Eq. (10) fails. Denote $t^*$ as the smallest time such that there exists an $\alpha \in [0, 1)$ such that Eq. (10) fails for $t^*$ and $\alpha$. By Lemma 13, $\left\| \sqrt[4]{\varepsilon \mathbb{1}_d + \hat{\nu}(t^* - 1)} \odot (\boldsymbol{w}(t^*) - \boldsymbol{w}(t^* - 1)) \right\|^2$ is positive, and strict inequality in Eq. (10) holds for $t$ and $\alpha = 0$, which by continuity leads to

$$1 > \alpha^* \triangleq \inf\{\alpha \in [0, 1] : Eq.(10) \; fails \; for \; 1 + \alpha\} > 0.$$

Then, for any $\alpha \in [0, \alpha^*]$, we have

$$\mathcal{L}(\boldsymbol{w}(t^* + \alpha))$$
$$\leq \mathcal{L}(\boldsymbol{w}(t^* + \alpha)) + \frac{1}{2}\alpha^2 \frac{1 - \beta_1^{t^*}}{\eta(1 - \beta_1)} \left\| \sqrt[4]{\varepsilon \mathbb{1}_d + \hat{\nu}(t^*)} \odot (\boldsymbol{w}(t^* + 1) - \boldsymbol{w}(t^*)) \right\|^2$$
$$\leq \mathcal{L}(\boldsymbol{w}(t^*)) + \frac{1 - \beta_1^{t^* - 1}}{2c\eta(1 - \beta_1)} \frac{1 - (c\beta_1)^{t^*}}{1 - (c\beta_1)^{t^* - 1}} \left\| \sqrt[4]{\varepsilon \mathbb{1}_d + \hat{\nu}(t^* - 1)} \odot (\boldsymbol{w}(t^*) - \boldsymbol{w}(t^* - 1)) \right\|^2.$$

On the other hand, for any time $2 \leq s \leq t^* - 1$, we have

$$\mathcal{L}(\boldsymbol{w}(s + 1)) + \frac{1}{2}\frac{1 - \beta_1^s}{\eta(1 - \beta_1)} \left\| \sqrt[4]{\varepsilon \mathbb{1}_d + \hat{\nu}(s)} \odot (\boldsymbol{w}(s + 1) - \boldsymbol{w}(s)) \right\|^2$$
$$\leq \mathcal{L}(\boldsymbol{w}(s)) + \frac{\beta_1(1 - \beta_1^{s-1})}{2c\eta(1 - \beta_1)} \frac{1 - (c\beta_1)^s}{1 - (c\beta_1)^{s-1}} \left\| \sqrt[4]{\varepsilon \mathbb{1}_d + \hat{\nu}(s - 1)} \odot (\boldsymbol{w}(s) - \boldsymbol{w}(s - 1)) \right\|^2. \quad (48)$$

By Eq. (46), we have

$$\frac{1 - \beta_1^s}{\eta(1 - \beta_1)} > \frac{1 - \beta_1^s}{c\eta(1 - \beta_1)} = \frac{1 - \beta_1^s}{c\eta(1 - \beta_1)} \frac{1 - (c\beta_1)^{s+1}}{1 - (c\beta_1)^s} \frac{1 - (c\beta_1)^s}{1 - (c\beta_1)^{s+1}},$$

which by $\frac{(1 - \beta_1^{s-1})}{(1 - \beta_1^s)}$ further leads to

$$\mathcal{L}(\boldsymbol{w}(s)) + \frac{1 - \beta_1^{s-1}}{2c\eta(1 - \beta_1)} \frac{1 - (c\beta_1)^s}{1 - (c\beta_1)^{s-1}} \left\| \sqrt[4]{\varepsilon \mathbb{1}_d + \hat{\nu}(s - 1)} \odot (\boldsymbol{w}(s) - \boldsymbol{w}(s - 1)) \right\|^2$$
$$\geq \mathcal{L}(\boldsymbol{w}(s + 1)) + \frac{1}{2}\frac{1 - \beta_1^s}{\eta(1 - \beta_1)} \left\| \sqrt[4]{\varepsilon \mathbb{1}_d + \hat{\nu}(s)} \odot (\boldsymbol{w}(s + 1) - \boldsymbol{w}(s)) \right\|^2$$
$$> \mathcal{L}(\boldsymbol{w}(s + 1)) + \frac{1 - \beta_1^s}{2c\eta(1 - \beta_1)} \frac{1 - (c\beta_1)^{s+1}}{1 - (c\beta_1)^s} \frac{1 - (c\beta_1)^s}{1 - (c\beta_1)^{s+1}} \left\| \sqrt[4]{\varepsilon \mathbb{1}_d + \hat{\nu}(s)} \odot (\boldsymbol{w}(s + 1) - \boldsymbol{w}(s)) \right\|^2$$
$$> \frac{1 - (c\beta_1)^s}{1 - (c\beta_1)^{s+1}} \left( \mathcal{L}(\boldsymbol{w}(s + 1)) + \frac{1 - \beta_1^s}{2c\eta(1 - \beta_1)} \frac{1 - (c\beta_1)^{s+1}}{1 - (c\beta_1)^s} \left\| \sqrt[4]{\varepsilon \mathbb{1}_d + \hat{\nu}(s)} \odot (\boldsymbol{w}(s + 1) - \boldsymbol{w}(s)) \right\|^2 \right).$$
$$(49)$$

On the other hand, for $s = 1$, we have

$$\mathcal{L}(\boldsymbol{w}(1)) \geq \mathcal{L}(\boldsymbol{w}(2)) + \frac{1}{2}\frac{1 - \beta_1}{\eta(1 - \beta_1)} \left\| \sqrt[4]{\varepsilon \mathbb{1}_d + \hat{\nu}(1)} \odot (\boldsymbol{w}(2) - \boldsymbol{w}(1)) \right\|^2$$
$$\geq \frac{1 - (c\beta_1)}{1 - (c\beta_1)^2} \left( \mathcal{L}(\boldsymbol{w}(2)) + \frac{1 - \beta_1}{2c\eta(1 - \beta_1)} \frac{1 - (c\beta_1)^2}{1 - (c\beta_1)} \left\| \sqrt[4]{\varepsilon \mathbb{1}_d + \hat{\nu}(1)} \odot (\boldsymbol{w}(2) - \boldsymbol{w}(1)) \right\|^2 \right).$$
$$(50)$$

Combining Eqs. (48), (49), and (50), we have

$$
\mathcal{L}(\boldsymbol{w}(t^* + \alpha))
$$
$$
\leq \mathcal{L}(\boldsymbol{w}(t^*)) + \frac{1 - \beta_1^{t^*-1}}{2c\eta(1-\beta_1)} \frac{1 - (c\beta_1)^{t^*}}{1 - (c\beta_1)^{t^*-1}} \left\| \sqrt[4]{\varepsilon \mathbb{1}_d + \hat{\nu}(t^* - 1)} \odot (\boldsymbol{w}(t^*) - \boldsymbol{w}(t^* - 1)) \right\|^2
$$
$$
< \frac{1 - (c\beta_1)^{t^*}}{1 - (c\beta_2)^{t^*-1}} \left( \mathcal{L}(\boldsymbol{w}(t^* - 1)) + \frac{1 - \beta_1^{t^*-2}}{2c\eta(1-\beta_1)} \frac{1 - (c\beta_1)^{t^*-1}}{1 - (c\beta_1)^{t^*-2}} \left\| \sqrt[4]{\varepsilon \mathbb{1}_d + \hat{\nu}(t^* - 2)} \odot (\boldsymbol{w}(t^* - 1) - \boldsymbol{w}(t^* - 2)) \right\|^2 \right)
$$
$$
< \cdots
$$
$$
< \frac{1 - (c\beta_1)^{t^*}}{1 - (c\beta_1)^2} \left( \mathcal{L}(\boldsymbol{w}(2)) + \frac{1 - \beta_1}{2c\eta(1-\beta_1)} \frac{1 - (c\beta_1)^2}{1 - (c\beta_1)} \left\| \sqrt[4]{\varepsilon \mathbb{1}_d + \hat{\nu}(1)} \odot (\boldsymbol{w}(2) - \boldsymbol{w}(1)) \right\|^2 \right)
$$
$$
\leq \frac{1 - (c\beta_1)^{t^*}}{1 - c\beta_1} \mathcal{L}(\boldsymbol{w}(1)) < \frac{1}{1 - c\beta_1} \mathcal{L}(\boldsymbol{w}(1)).
$$

Therefore, by Lemma 10, $\mathcal{L}$ is $H_{\ell^{-1}(\frac{1}{1-c\beta_1} N \mathcal{L}(\boldsymbol{w}(1)))}$ smooth (thus $L$ smooth) over the set $\{\boldsymbol{w}(t^* + \alpha) : \alpha \in [0, \alpha^*]\}$, which further leads to

$$
\mathcal{L}(\boldsymbol{w}(t^* + \alpha^*))
$$
$$
\leq \mathcal{L}(\boldsymbol{w}(t^*)) + \alpha^* \langle \nabla \mathcal{L}(\boldsymbol{w}(t^*)), \boldsymbol{w}(t^* + 1) - \boldsymbol{w}(t^*) \rangle + \frac{L}{2}(\alpha^*)^2 \|\boldsymbol{w}(t^* + 1) - \boldsymbol{w}(t^*)\|^2
$$
$$
\overset{(\bullet)}{=} - \frac{\alpha^*}{\eta(1 - \beta_1)} \left\langle \boldsymbol{w}(t^* + 1) - \boldsymbol{w}(t^*), (1 - \beta_1^{t^*}) \sqrt{\varepsilon \mathbb{1}_d + \hat{\nu}(t^*)} \odot (\boldsymbol{w}(t^* + 1) - \boldsymbol{w}(t^*)) \right.
$$
$$
\left. - \beta_1 (1 - \beta_1^{t^*-1}) \sqrt{\varepsilon \mathbb{1}_d + \hat{\nu}(t^*)} \odot (\boldsymbol{w}(t^*) - \boldsymbol{w}(t^* - 1)) \right\rangle
$$
$$
+ \mathcal{L}(\boldsymbol{w}(t^*)) + \frac{L}{2}(\alpha^*)^2 \|\boldsymbol{w}(t^* + 1) - \boldsymbol{w}(t^*)\|^2
$$
$$
= \mathcal{L}(\boldsymbol{w}(t^*)) + \frac{L}{2}(\alpha^*)^2 \|\boldsymbol{w}(t^* + 1) - \boldsymbol{w}(t^*)\|^2 - \frac{\alpha^*(1 - \beta_1^{t^*})}{\eta(1 - \beta_1)} \left\| \sqrt[4]{\varepsilon \mathbb{1}_d + \hat{\nu}(t^*)} \odot (\boldsymbol{w}(t^* + 1) - \boldsymbol{w}(t^*)) \right\|^2
$$
$$
+ \beta_1 \frac{\alpha^*(1 - \beta_1^{t^*-1})}{\eta(1 - \beta_1)} \left\langle \boldsymbol{w}(t^* + 1) - \boldsymbol{w}(t^*), \sqrt{\varepsilon \mathbb{1}_d + \hat{\nu}(t^*)} \odot (\boldsymbol{w}(t^*) - \boldsymbol{w}(t^* - 1)) \right\rangle
$$
$$
= \mathcal{L}(\boldsymbol{w}(t^*)) + \frac{L}{2}(\alpha^*)^2 \|\boldsymbol{w}(t^* + 1) - \boldsymbol{w}(t^*)\|^2 - \frac{\alpha^*(1 - \beta_1^{t^*})}{\eta(1 - \beta_1)} \left\| \sqrt[4]{\varepsilon \mathbb{1}_d + \hat{\nu}(t^*)} \odot (\boldsymbol{w}(t^* + 1) - \boldsymbol{w}(t^*)) \right\|^2
$$
$$
+ \beta_1 \frac{\alpha^*(1 - \beta_1^{t^*-1})}{\eta(1 - \beta_1)} \left\langle \frac{\sqrt[8]{\varepsilon \mathbb{1}_d + \hat{\nu}(t^* - 1)}}{\sqrt[8]{\varepsilon \mathbb{1}_d + \hat{\nu}(t^*)}} \odot \sqrt[4]{\varepsilon \mathbb{1}_d + \hat{\nu}(t^*)} \odot (\boldsymbol{w}(t^* + 1) - \boldsymbol{w}(t^*)), \right.
$$
$$
\left. \frac{\sqrt[8]{\varepsilon \mathbb{1}_d + \hat{\nu}(t^* - 1)}}{\sqrt[8]{\varepsilon \mathbb{1}_d + \hat{\nu}(t^*)}} \odot \sqrt[4]{\varepsilon \mathbb{1}_d + \hat{\nu}(t^* - 1)} \odot (\boldsymbol{w}(t^*) - \boldsymbol{w}(t^* - 1)) \right\rangle
$$
$$
\leq \mathcal{L}(\boldsymbol{w}(t^*)) + \frac{L}{2}(\alpha^*)^2 \|\boldsymbol{w}(t^* + 1) - \boldsymbol{w}(t^*)\|^2 - \frac{\alpha^*(1 - \beta_1^{t^*})}{\eta(1 - \beta_1)} \left\| \sqrt[4]{\varepsilon \mathbb{1}_d + \hat{\nu}(t^*)} \odot (\boldsymbol{w}(t^* + 1) - \boldsymbol{w}(t^*)) \right\|^2
$$
$$
+ \beta_1 \frac{(\alpha^*)^2 (1 - \beta_1^{t^*-1})}{2\eta(1 - \beta_1)} \left\| \frac{\sqrt[8]{\varepsilon \mathbb{1}_d + \hat{\nu}(t^* - 1)}}{\sqrt[8]{\varepsilon \mathbb{1}_d + \hat{\nu}(t^*)}} \odot \sqrt[4]{\varepsilon \mathbb{1}_d + \hat{\nu}(t^*)} \odot (\boldsymbol{w}(t^* + 1) - \boldsymbol{w}(t^*)) \right\|^2
$$
$$
+ \beta_1 \frac{(1 - \beta_1^{t^*-1})}{2\eta(1 - \beta_1)} \left\| \frac{\sqrt[8]{\varepsilon \mathbb{1}_d + \hat{\nu}(t^* - 1)}}{\sqrt[8]{\varepsilon \mathbb{1}_d + \hat{\nu}(t^*)}} \odot \sqrt[4]{\varepsilon \mathbb{1}_d + \hat{\nu}(t^* - 1)} \odot (\boldsymbol{w}(t^*) - \boldsymbol{w}(t^* - 1)) \right\|^2
$$

$$\overset{(\diamond)}{\leq} \mathcal{L}(\boldsymbol{w}(t^*)) + \frac{L}{2}(\alpha^*)^2 \|\boldsymbol{w}(t^*+1) - \boldsymbol{w}(t^*)\|^2 - \frac{\alpha^*(1-\beta_1^{t^*})}{\eta(1-\beta_1)} \left\| \sqrt[4]{\varepsilon \mathbb{1}_d + \hat{\nu}(t^*)} \odot (\boldsymbol{w}(t^*+1) - \boldsymbol{w}(t^*)) \right\|^2$$

$$+ \beta_1 \frac{(\alpha^*)^2(1-\beta_1^{t^*-1})}{2\eta(1-\beta_1)} \frac{1-(c\beta_1)^{t^*}}{c\beta_1(1-(c\beta_1)^{t^*-1})} \left\| \sqrt[4]{\varepsilon \mathbb{1}_d + \hat{\nu}(t^*)} \odot (\boldsymbol{w}(t^*+1) - \boldsymbol{w}(t^*)) \right\|^2$$

$$+ \beta_1 \frac{(1-\beta_1^{t^*-1})}{2\eta(1-\beta_1)} \frac{1-(c\beta_1)^{t^*}}{c\beta_1(1-(c\beta_1)^{t^*-1})} \left\| \sqrt[4]{\varepsilon \mathbb{1}_d + \hat{\nu}(t^*-1)} \odot (\boldsymbol{w}(t^*) - \boldsymbol{w}(t^*-1)) \right\|^2$$

$$\leq \mathcal{L}(\boldsymbol{w}(t^*)) + \frac{L}{2\sqrt{\varepsilon}}(\alpha^*)^2 \| \sqrt[4]{\varepsilon \mathbb{1}_d + \hat{\nu}(t^*)} \odot (\boldsymbol{w}(t^*+1) - \boldsymbol{w}(t^*)) \|^2$$

$$- \frac{\alpha^*(1-\beta_1^{t^*})}{\eta(1-\beta_1)} \left\| \sqrt[4]{\varepsilon \mathbb{1}_d + \hat{\nu}(t^*)} \odot (\boldsymbol{w}(t^*+1) - \boldsymbol{w}(t^*)) \right\|^2$$

$$+ \beta_1 \frac{(\alpha^*)^2(1-\beta_1^{t^*})}{2\eta(1-\beta_1)} \frac{1-(c\beta_1)^{t^*}}{c\beta_1(1-(c\beta_1)^{t^*-1})} \left\| \sqrt[4]{\varepsilon \mathbb{1}_d + \hat{\nu}(t^*)} \odot (\boldsymbol{w}(t^*+1) - \boldsymbol{w}(t^*)) \right\|^2$$

$$+ \beta_1 \frac{(1-\beta_1^{t^*})}{2\eta(1-\beta_1)} \frac{1-(c\beta_1)^{t^*}}{c\beta_1(1-(c\beta_1)^{t^*-1})} \left\| \sqrt[4]{\varepsilon \mathbb{1}_d + \hat{\nu}(t^*-1)} \odot (\boldsymbol{w}(t^*) - \boldsymbol{w}(t^*-1)) \right\|^2$$

$$\overset{(\square)}{<} \mathcal{L}(\boldsymbol{w}(t^*)) - \frac{(\alpha^*)^2(1-\beta_1^{t^*})}{2\eta(1-\beta_1)} \left\| \sqrt[4]{\varepsilon \mathbb{1}_d + \hat{\nu}(t^*)} \odot (\boldsymbol{w}(t^*+1) - \boldsymbol{w}(t^*)) \right\|^2$$

$$+ \frac{(1-\beta_1^{t^*-1})}{2\eta(1-\beta_1)} \frac{1-(c\beta_1)^{t^*}}{c(1-(c\beta_1)^{t^*-1})} \left\| \sqrt[4]{\varepsilon \mathbb{1}_d + \hat{\nu}(t^*-1)} \odot (\boldsymbol{w}(t^*) - \boldsymbol{w}(t^*-1)) \right\|^2,$$

where Eq. ($\bullet$) is due to an alternative form of the Adam's update rule:

$$(1-\beta_1^{t^*})\sqrt{\varepsilon \mathbb{1}_d + \hat{\nu}(t^*)} \odot (\boldsymbol{w}(t^*+1) - \boldsymbol{w}(t^*)) - \beta_1(1-\beta_1^{t^*-1})\sqrt{\varepsilon \mathbb{1}_d + \hat{\nu}(t^*-1)} \odot (\boldsymbol{w}(t^*) - \boldsymbol{w}(t^*-1))$$
$$= -\eta(1-\beta_1)\nabla\mathcal{L}(\boldsymbol{w}(t^*)), \tag{51}$$

Inequality ($\diamond$) is due to

$$\frac{\sqrt[4]{\varepsilon \mathbb{1}_d + \hat{\nu}(t^*-1)}}{\sqrt[4]{\varepsilon \mathbb{1}_d + \hat{\nu}(t^*)}} = \sqrt[4]{\frac{\varepsilon \mathbb{1}_d + \hat{\nu}(t^*-1)}{\varepsilon \mathbb{1}_d + \hat{\nu}(t^*)}} = \sqrt[4]{\frac{\varepsilon \mathbb{1}_d + \hat{\nu}(t^*-1)}{\varepsilon \mathbb{1}_d + \frac{\beta_2 \nu(t^*-1)+(1-\beta_2)\nabla\mathcal{L}(\boldsymbol{w}(t^*))^2}{1-\beta_2^{t^*}}}}$$

$$\leq \sqrt[4]{\frac{\varepsilon \mathbb{1}_d + \hat{\nu}(t^*-1)}{\varepsilon \mathbb{1}_d + \frac{\beta_2 \nu(t^*-1)}{1-\beta_2^{t^*}}}} = \sqrt[4]{\frac{\varepsilon \mathbb{1}_d + \hat{\nu}(t^*-1)}{\varepsilon \mathbb{1}_d + \frac{\beta_2(1-\beta_2^{t^*-1})\hat{\nu}(t^*-1)}{1-\beta_2^{t^*}}}} \leq \sqrt[4]{\frac{\varepsilon \mathbb{1}_d + \hat{\nu}(t^*-1)}{\frac{\beta_2(1-\beta_2^{t^*-1})\hat{\nu}(t^*-1)}{1-\beta_2^{t^*}}\varepsilon \mathbb{1}_d + \frac{\beta_2(1-\beta_2^{t^*-1})\hat{\nu}(t^*-1)}{1-\beta_2^{t^*}}}}$$

$$= \sqrt[4]{\frac{1-\beta_2^{t^*}}{\beta_2(1-\beta_2^{t^*-1})}} \mathbb{1}_d \ (\textit{all the computings are component-wisely}),$$

and $f(c\beta_1) \geq \sqrt[4]{f((c\beta_1)^4)}$, and Inequality ($\square$) is due to

$$\frac{L}{2\sqrt{\varepsilon}} \leq \frac{\inf_{t\geq 2}\left(\frac{1-\beta_1^t}{1-\beta_1} - \frac{1-\beta_1^{t-1}}{c(1-\beta_1)}\frac{1-(c\beta_1)^t}{1-(c\beta_1)^{t-1}}\right)}{2\eta} \leq \frac{\left(\frac{1-\beta_1^{t^*}}{1-\beta_1} - \frac{1-\beta_1^{t^*-1}}{c(1-\beta_1)}\frac{1-(c\beta_1)^{t^*}}{1-(c\beta_1)^{t^*-1}}\right)}{2\eta},$$

$\alpha^* > (\alpha^*)^2$, and $\left\| \sqrt[4]{\varepsilon \mathbb{1}_d + \hat{\nu}(t^*)} \odot (\boldsymbol{w}(t^*+1) - \boldsymbol{w}(t^*)) \right\|^2 > 0$.

This contradicts to that the equality in Eq. (10) holds for $t^* + \alpha^*$.

The proof is completed. $\qquad\square$

As $\lim_{t\to\infty}\beta_1^t = 0$ and $\lim_{t\to\infty}(c\beta_1)^t = 0$, we have the following corollary based on Lemma 1.

**Corollary 6.** *Let all assumptions in Theorem 4 hold. Then, for large enough $t$, we have*

$$\mathcal{L}(\boldsymbol{w}(t+1)) + \frac{1}{2\sqrt[4]{c}\eta(1-\beta_1)} \left\| \sqrt[4]{\varepsilon \mathbb{1}_d + \hat{\nu}(t)} \odot (\boldsymbol{w}(t+1) - \boldsymbol{w}(t)) \right\|^2$$

$$\leq \mathcal{L}(\boldsymbol{w}(t)) + \frac{1}{2\sqrt[2]{c}\eta(1-\beta_1)} \left\| \sqrt[4]{\varepsilon \mathbb{1}_d + \hat{\nu}(t-1)} \odot (\boldsymbol{w}(t) - \boldsymbol{w}(t-1)) \right\|^2. \tag{52}$$

*Consequently, we have*

$$\sum_{t=1}^{\infty} \|\nabla \mathcal{L}(\boldsymbol{w}(t))\|^2 < \infty. \tag{53}$$

The proof of Corollary 6 requires the following classical lemma on the equivalence between the convergence of two non-negative sequence. The proof is omitted here and can be found in Wang et al. (2021).

**Lemma 20** (c.f. Lemma 27, Wang et al. (2021)). *Let $\{a_i\}_{i=1}^{\infty}$ be a series of non-negative reals, and $\varepsilon$ be a positive real. Then, $\sum_{i=1}^{\infty} a_i < \infty$ is equivalent to $\sum_{i=1}^{\infty} \frac{a_i}{\sqrt{\varepsilon + \sum_{s=1}^{t} a_s}} < \infty$.*

*Proof of Corollary 6.* We have

$$\lim_{t\to\infty} \frac{1-\beta_1^{t-1}}{2c\eta(1-\beta_1)} \frac{1-(c\beta_1)^t}{1-(c\beta_1)^{t-1}} = \frac{1}{2c\eta(1-\beta_1)} < \frac{1}{2\sqrt[2]{c}\eta(1-\beta_1)},$$

$$\lim_{t\to\infty} \frac{1-\beta_1^t}{2\eta(1-\beta_1)} = \frac{1}{2\eta(1-\beta_1)} > \frac{1}{2\sqrt[4]{c}\eta(1-\beta_1)},$$

which completes the proof of Eq. (52). Rearranging Eq. (52) leads to

$$\frac{\sqrt[4]{c}-1}{2\sqrt[2]{c}\eta(1-\beta_1)} \left\| \sqrt[4]{\varepsilon\mathbb{1}_d + \hat{\nu}(t)} \odot (\boldsymbol{w}(t+1) - \boldsymbol{w}(t)) \right\|^2 \le \frac{1}{2\sqrt[4]{c}\eta(1-\beta_1)} \left\| \sqrt[4]{\varepsilon\mathbb{1}_d + \hat{\nu}(t-1)} \odot (\boldsymbol{w}(t) - \boldsymbol{w}(t-1)) \right\|^2$$

$$+ \mathcal{L}(\boldsymbol{w}(t)) - \left( \mathcal{L}(\boldsymbol{w}(t+1)) + \frac{1}{2\sqrt[4]{c}\eta(1-\beta_1)} \left\| \sqrt[4]{\varepsilon\mathbb{1}_d + \hat{\nu}(t)} \odot (\boldsymbol{w}(t+1) - \boldsymbol{w}(t)) \right\|^2 \right),$$

which by iteration further leads to that for a large enough time $T_1$

$$\sum_{t=T_1}^{T_2} \frac{\sqrt[4]{c}-1}{2\sqrt[2]{c}\eta(1-\beta_1)} \left\| \sqrt[4]{\varepsilon\mathbb{1}_d + \hat{\nu}(t)} \odot (\boldsymbol{w}(t+1) - \boldsymbol{w}(t)) \right\|^2$$

$$\le \mathcal{L}(\boldsymbol{w}(T_1)) + \frac{1}{2\sqrt[4]{c}\eta(1-\beta_1)} \left\| \sqrt[4]{\varepsilon\mathbb{1}_d + \hat{\nu}(T_1-1)} \odot (\boldsymbol{w}(T_1) - \boldsymbol{w}(T_1-1)) \right\|^2$$

$$- \mathcal{L}(\boldsymbol{w}(T_2+1)) + \frac{1}{2\sqrt[4]{c}\eta(1-\beta_1)} \left\| \sqrt[4]{\varepsilon\mathbb{1}_d + \hat{\nu}(T_2)} \odot (\boldsymbol{w}(T_2+1) - \boldsymbol{w}(T_2+1)) \right\|^2$$

$$< \mathcal{L}(\boldsymbol{w}(T_1)) + \frac{1}{2\sqrt[4]{c}\eta(1-\beta_1)} \left\| \sqrt[4]{\varepsilon\mathbb{1}_d + \hat{\nu}(T_1-1)} \odot (\boldsymbol{w}(T_1) - \boldsymbol{w}(T_1-1)) \right\|^2.$$

Consequently, we obtain

$$\sum_{t=1}^{\infty} \frac{\sqrt[4]{c}-1}{2\sqrt[2]{c}\eta(1-\beta_1)} \left\| \sqrt[4]{\varepsilon\mathbb{1}_d + \hat{\nu}(t)} \odot (\boldsymbol{w}(t+1) - \boldsymbol{w}(t)) \right\|^2 < \infty. \tag{54}$$

On the other hand, for any $t$, we have

$$\left\| \sqrt[4]{\varepsilon\mathbb{1}_d + \hat{\nu}(t)} \odot (\boldsymbol{w}(t+1) - \boldsymbol{w}(t)) \right\| \left\| \sqrt[4]{\varepsilon\mathbb{1}_d + \hat{\nu}(t)} \odot \hat{\boldsymbol{w}} \right\|$$

$$\ge \left\langle \sqrt[4]{\varepsilon\mathbb{1}_d + \hat{\nu}(t)} \odot (\boldsymbol{w}(t+1) - \boldsymbol{w}(t)), \sqrt[4]{\varepsilon\mathbb{1}_d + \hat{\nu}(t)} \odot \hat{\boldsymbol{w}} \right\rangle = \left\langle \sqrt[2]{\varepsilon\mathbb{1}_d + \hat{\nu}(t)} \odot (\boldsymbol{w}(t+1) - \boldsymbol{w}(t)), \hat{\boldsymbol{w}} \right\rangle$$

$$= \langle -\eta\hat{\boldsymbol{m}}(t), \hat{\boldsymbol{w}} \rangle = -\frac{\eta(1-\beta_1)}{1-\beta_1^t} \left\langle \sum_{s=1}^{t} \beta_1^{t-s} \nabla\mathcal{L}(\boldsymbol{w}(s)), \hat{\boldsymbol{w}} \right\rangle$$

$$= -\frac{\eta(1-\beta_1)}{1-\beta_1^t} \frac{1}{N} \left\langle \sum_{s=1}^{t} \beta_1^{t-s} \sum_{\tilde{\boldsymbol{x}}_i \in \mathcal{T}(\boldsymbol{S})} \ell'(\langle \tilde{\boldsymbol{x}}_i, \boldsymbol{w}(s) \rangle)\tilde{\boldsymbol{x}}_i, \hat{\boldsymbol{w}} \right\rangle \ge -\frac{\eta(1-\beta_1)}{1-\beta_1^t} \frac{1}{N} \sum_{s=1}^{t} \beta_1^{t-s} \sum_{\tilde{\boldsymbol{x}}_i \in \mathcal{T}(\boldsymbol{S})} \ell'(\langle \tilde{\boldsymbol{x}}_i, \boldsymbol{w}(s) \rangle)$$

$$\ge -\frac{\eta(1-\beta_1)}{1-\beta_1^t} \frac{1}{N} \sum_{\tilde{\boldsymbol{x}}_i \in \mathcal{T}(\boldsymbol{S})} \ell'(\langle \tilde{\boldsymbol{x}}_i, \boldsymbol{w}(t) \rangle) \ge \frac{\eta(1-\beta_1)}{1-\beta_1^t} \|\nabla\mathcal{L}(\boldsymbol{w}(t))\|,$$

which by Eq. (54) indicates

$$\sum_{t=1}^{\infty} \left( \frac{\eta(1-\beta_1)}{1-\beta_1^t} \right)^2 \frac{\|\nabla\mathcal{L}(\boldsymbol{w}(t))\|^2}{\left\| \sqrt[4]{\varepsilon\mathbb{1}_d + \hat{\nu}(t)} \odot \hat{\boldsymbol{w}} \right\|^2} < \infty.$$

As $\lim_{t\to\infty} \left( \frac{\eta(1-\beta_1)}{1-\beta_1^t} \right)^2 = \eta^2(1-\beta_1)^2$, we then obtain

$$\sum_{t=1}^{\infty} \frac{\|\nabla\mathcal{L}(\boldsymbol{w}(t))\|^2}{\sqrt[2]{\varepsilon + \sum_{s=1}^{t} \|\nabla\mathcal{L}(\boldsymbol{w}(t))\|^2}} \leq \sum_{t=1}^{\infty} \frac{\|\nabla\mathcal{L}(\boldsymbol{w}(t))\|^2}{\sqrt[2]{\varepsilon + \sum_{s=1}^{t} (1-\beta)\beta^{t-s}\|\nabla\mathcal{L}(\boldsymbol{w}(t))\|^2}}$$

$$\leq \sqrt{\frac{1}{1-\beta}} \sum_{t=1}^{\infty} \frac{\|\nabla\mathcal{L}(\boldsymbol{w}(t))\|^2}{\sqrt[2]{\varepsilon + \frac{\sum_{s=1}^{t}(1-\beta)\beta^{t-s}\|\nabla\mathcal{L}(\boldsymbol{w}(t))\|^2}{1-\beta^t}}} \leq d\sqrt{\frac{1}{1-\beta}} \sum_{t=1}^{\infty} \frac{\|\nabla\mathcal{L}(\boldsymbol{w}(t))\|^2}{\left\| \sqrt[4]{\varepsilon\mathbb{1}_d + \frac{\sum_{s=1}^{t}(1-\beta)\beta^{t-s}\nabla\mathcal{L}(\boldsymbol{w}(t))^2}{1-\beta^t}} \right\|^2}$$

$$= d\sqrt{\frac{1}{1-\beta}} \sum_{t=1}^{\infty} \frac{\|\nabla\mathcal{L}(\boldsymbol{w}(t))\|^2}{\left\| \sqrt[4]{\varepsilon\mathbb{1}_d + \hat{\nu}(t)} \right\|^2} \leq d\|\hat{\boldsymbol{w}}\|_{\infty}^2 \sqrt{\frac{1}{1-\beta}} \sum_{t=1}^{\infty} \frac{\|\nabla\mathcal{L}(\boldsymbol{w}(t))\|^2}{\left\| \sqrt[4]{\varepsilon\mathbb{1}_d + \hat{\nu}(t)} \odot \hat{\boldsymbol{w}} \right\|^2} < \infty,$$

which by Lemma 20 completes the proof. $\qquad\square$

Based on Corollary 6, we can further prove Lemma 6, characterizing the convergent rate of loss $\mathcal{L}$ directly.

*Proof of Lemma 6.* To begin with, Eq. (51) indicates

$$\|\eta(1-\beta_1)\nabla\mathcal{L}(\boldsymbol{w}(t))\|^2$$
$$= \left\| (1-\beta_1^t)\sqrt{\varepsilon\mathbb{1}_d + \hat{\nu}(t)} \odot (\boldsymbol{w}(t+1) - \boldsymbol{w}(t)) - \beta_1(1-\beta_1^{t-1})\sqrt{\varepsilon\mathbb{1}_d + \hat{\nu}(t-1)} \odot (\boldsymbol{w}(t) - \boldsymbol{w}(t-1)) \right\|^2$$
$$\leq \left\| (1-\beta_1^t)\sqrt{\varepsilon\mathbb{1}_d + \hat{\nu}(t)} \odot (\boldsymbol{w}(t+1) - \boldsymbol{w}(t)) \right\| + \left\| \beta_1(1-\beta_1^{t-1})\sqrt{\varepsilon\mathbb{1}_d + \hat{\nu}(t-1)} \odot (\boldsymbol{w}(t) - \boldsymbol{w}(t-1)) \right\|^2$$
$$\leq \left( \left\| \sqrt{\varepsilon\mathbb{1}_d + \hat{\nu}(t)} \odot (\boldsymbol{w}(t+1) - \boldsymbol{w}(t)) \right\| + \left\| \sqrt{\varepsilon\mathbb{1}_d + \hat{\nu}(t-1)} \odot (\boldsymbol{w}(t) - \boldsymbol{w}(t-1)) \right\| \right)^2$$
$$\leq 2 \left( \left\| \sqrt{\varepsilon\mathbb{1}_d + \hat{\nu}(t)} \odot (\boldsymbol{w}(t+1) - \boldsymbol{w}(t)) \right\|^2 + \left\| \sqrt{\varepsilon\mathbb{1}_d + \hat{\nu}(t-1)} \odot (\boldsymbol{w}(t) - \boldsymbol{w}(t-1)) \right\|^2 \right) \qquad (55)$$

On the other hand, by Corollary 6,

$$\sum_{s=1}^{\infty} \|\nabla\mathcal{L}(\boldsymbol{w}(s))\|^2 < \infty,$$

which following the same routine as Corollary 5 leads to

$$\langle \boldsymbol{w}(t), \tilde{\boldsymbol{x}} \rangle \to \infty, \forall \tilde{\boldsymbol{x}} \in \tilde{\boldsymbol{S}}.$$

Therefore, by Lemma 11, there exists a large enough time $T_1$, such that $\forall t \geq T_1$,

$$\frac{1}{K}\ell(\langle \boldsymbol{w}(t), \tilde{\boldsymbol{x}} \rangle) \leq -\ell'(\langle \boldsymbol{w}(t), \tilde{\boldsymbol{x}} \rangle) \leq K\ell(\langle \boldsymbol{w}(t), \tilde{\boldsymbol{x}} \rangle), \forall \tilde{\boldsymbol{x}} \in \mathcal{T}(\boldsymbol{S}),$$

which by the separable assumption further leads to

$$\frac{\gamma}{K}\mathcal{L}(\boldsymbol{w}(t)) \leq -\frac{\gamma}{N}\sum_{\tilde{\boldsymbol{x}}\in\mathcal{T}(\boldsymbol{S})} \ell'(\langle \boldsymbol{w}(t), \tilde{\boldsymbol{x}} \rangle) \leq \frac{1}{N}\left\langle -\sum_{\tilde{\boldsymbol{x}}\in\mathcal{T}(\boldsymbol{S})} \ell'(\langle \boldsymbol{w}(t), \tilde{\boldsymbol{x}} \rangle)\tilde{\boldsymbol{x}}, \gamma\hat{\boldsymbol{w}} \right\rangle$$

$$\leq \frac{1}{N}\left\| \sum_{\tilde{\boldsymbol{x}}\in\mathcal{T}(\boldsymbol{S})} \ell'(\langle \boldsymbol{w}(t), \tilde{\boldsymbol{x}} \rangle)\tilde{\boldsymbol{x}} \right\| \|\gamma\hat{\boldsymbol{w}}\| = \|\nabla\mathcal{L}(\boldsymbol{w}(t))\|$$

$$\leq -\frac{1}{N}\sum_{\tilde{\boldsymbol{x}}\in\mathcal{T}(\boldsymbol{S})} \ell'(\langle \boldsymbol{w}(t), \tilde{\boldsymbol{x}} \rangle) \leq K\mathcal{L}(\boldsymbol{w}(t)). \qquad (56)$$

Combining Eq. (26) and the above inequality, we have

$$
\left(\frac{\eta(1-\beta_1)\gamma}{K}\right)^2 \mathcal{L}(\boldsymbol{w}(t))^2 \leq 2 \left(\left\|\sqrt{\varepsilon\mathbb{1}_d + \hat{\nu}(t)} \odot (\boldsymbol{w}(t+1) - \boldsymbol{w}(t))\right\|^2\right.
$$
$$
\left. + \left\|\sqrt{\varepsilon\mathbb{1}_d + \hat{\nu}(t-1)} \odot (\boldsymbol{w}(t) - \boldsymbol{w}(t-1))\right\|^2\right). \tag{57}
$$

On the other hand, by Eq. (54), we have

$$
\sum_{t=1}^{\infty} \left\|\sqrt[4]{\varepsilon\mathbb{1}_d + \hat{\nu}(t)} \odot (\boldsymbol{w}(t+1) - \boldsymbol{w}(t))\right\|^2 < \infty.
$$

Therefore, there exists large enough time $T_2$, such that $\forall t > T_2$,

$$
\left\|\sqrt[4]{\varepsilon\mathbb{1}_d + \hat{\nu}(t)} \odot (\boldsymbol{w}(t+1) - \boldsymbol{w}(t))\right\|^2 < 1,
$$

and thus,

$$
\left\|\sqrt[4]{\varepsilon\mathbb{1}_d + \hat{\nu}(t)} \odot (\boldsymbol{w}(t+1) - \boldsymbol{w}(t))\right\|^4 < \left\|\sqrt[4]{\varepsilon\mathbb{1}_d + \hat{\nu}(t)} \odot (\boldsymbol{w}(t+1) - \boldsymbol{w}(t))\right\|^2. \tag{58}
$$

Combining Eq. (57) and Eq. (58), there exists a positive real constant $C$, such that

$$
\mathcal{L}(\boldsymbol{w}(t))^2 \leq C \left(\left\|\sqrt{\varepsilon\mathbb{1}_d + \hat{\nu}(t)} \odot (\boldsymbol{w}(t+1) - \boldsymbol{w}(t))\right\|^2\right.
$$
$$
\left. + \left\|\sqrt{\varepsilon\mathbb{1}_d + \hat{\nu}(t-1)} \odot (\boldsymbol{w}(t) - \boldsymbol{w}(t-1))\right\|^2\right),
$$
$$
\left\|\sqrt[4]{\varepsilon\mathbb{1}_d + \hat{\nu}(t)} \odot (\boldsymbol{w}(t+1) - \boldsymbol{w}(t))\right\|^4 \leq C \left(\left\|\sqrt{\varepsilon\mathbb{1}_d + \hat{\nu}(t)} \odot (\boldsymbol{w}(t+1) - \boldsymbol{w}(t))\right\|^2\right.
$$
$$
\left. + \left\|\sqrt{\varepsilon\mathbb{1}_d + \hat{\nu}(t-1)} \odot (\boldsymbol{w}(t) - \boldsymbol{w}(t-1))\right\|^2\right).
$$

Rearranging Eq. (52) leads to

$$
\frac{\sqrt[4]{c}-1}{4\sqrt[2]{c}\eta(1-\beta_1)} \left(\left\|\sqrt[4]{\varepsilon\mathbb{1}_d + \hat{\nu}(t-1)} \odot (\boldsymbol{w}(t) - \boldsymbol{w}(t-1))\right\|^2\right.
$$
$$
\left. + \left\|\sqrt[4]{\varepsilon\mathbb{1}_d + \hat{\nu}(t-1)} \odot (\boldsymbol{w}(t) - \boldsymbol{w}(t-1))\right\|^2\right)
$$
$$
\leq \mathcal{L}(\boldsymbol{w}(t)) + \frac{\sqrt[4]{c}+1}{4\sqrt[2]{c}\eta(1-\beta_1)} \left\|\sqrt[4]{\varepsilon\mathbb{1}_d + \hat{\nu}(t-1)} \odot (\boldsymbol{w}(t) - \boldsymbol{w}(t-1))\right\|^2
$$
$$
- \left(\mathcal{L}(\boldsymbol{w}(t+1)) + \frac{\sqrt[4]{c}+1}{4\sqrt[2]{c}\eta(1-\beta_1)} \left\|\sqrt[4]{\varepsilon\mathbb{1}_d + \hat{\nu}(t)} \odot (\boldsymbol{w}(t+1) - \boldsymbol{w}(t))\right\|^2\right),
$$

which further indicates

$$
\left( \mathcal{L}(\boldsymbol{w}(t)) + \frac{\sqrt[4]{c}+1}{4\sqrt[2]{c}\eta(1-\beta_1)} \left\| \sqrt[4]{\varepsilon\mathbb{1}_d + \hat{\nu}(t-1)} \odot (\boldsymbol{w}(t) - \boldsymbol{w}(t-1)) \right\|^2 \right)^2
$$

$$
\leq 2 \left( \mathcal{L}(\boldsymbol{w}(t))^2 + \frac{\sqrt[4]{c}+1}{4\sqrt[2]{c}\eta(1-\beta_1)} \left\| \sqrt[4]{\varepsilon\mathbb{1}_d + \hat{\nu}(t-1)} \odot (\boldsymbol{w}(t) - \boldsymbol{w}(t-1)) \right\|^4 \right)
$$

$$
\leq 2C \left( 1 + \frac{\sqrt[4]{c}+1}{4\sqrt[2]{c}\eta(1-\beta_1)} \right) \left( \left\| \sqrt{\varepsilon\mathbb{1}_d + \hat{\nu}(t)} \odot (\boldsymbol{w}(t+1) - \boldsymbol{w}(t)) \right\|^2 \right.
$$

$$
\left. + \left\| \sqrt{\varepsilon\mathbb{1}_d + \hat{\nu}(t-1)} \odot (\boldsymbol{w}(t) - \boldsymbol{w}(t-1)) \right\|^2 \right)
$$

$$
\leq 2C \left( 1 + \frac{\sqrt[4]{c}+1}{4\sqrt[2]{c}\eta(1-\beta_1)} \right) \frac{4\sqrt[2]{c}\eta(1-\beta_1)}{\sqrt[4]{c}-1} \left( \mathcal{L}(\boldsymbol{w}(t)) + \frac{\sqrt[4]{c}+1}{4\sqrt[2]{c}\eta(1-\beta_1)} \right.
$$

$$
\cdot \left\| \sqrt[4]{\varepsilon\mathbb{1}_d + \hat{\nu}(t-1)} \odot (\boldsymbol{w}(t) - \boldsymbol{w}(t-1)) \right\|^2 - (\mathcal{L}(\boldsymbol{w}(t+1)))
$$

$$
\left. + \frac{\sqrt[4]{c}+1}{4\sqrt[2]{c}\eta(1-\beta_1)} \left\| \sqrt[4]{\varepsilon\mathbb{1}_d + \hat{\nu}(t)} \odot (\boldsymbol{w}(t+1) - \boldsymbol{w}(t)) \right\|^2 \right) \right).
$$

Denote $\xi(t)$ as

$$
\xi(t) \triangleq \mathcal{L}(\boldsymbol{w}(t)) + \frac{\sqrt[4]{c}+1}{4\sqrt[2]{c}\eta(1-\beta_1)} \left\| \sqrt[4]{\varepsilon\mathbb{1}_d + \hat{\nu}(t-1)} \odot (\boldsymbol{w}(t) - \boldsymbol{w}(t-1)) \right\|^2.
$$

We then have

$$
\xi(t)^2 \leq 2C \left( 1 + \frac{\sqrt[4]{c}+1}{4\sqrt[2]{c}\eta(1-\beta_1)} \right) \frac{4\sqrt[2]{c}\eta(1-\beta_1)}{\sqrt[4]{c}-1} (\xi(t) - \xi(t+1)),
$$

which leads to

$$
\xi(t) = \mathcal{O}\left( \frac{1}{t} \right), \ i.e., \ \mathcal{L}(\boldsymbol{w}(t)) = \mathcal{O}\left( \frac{1}{t} \right),
$$

$$
and \ \left\| \sqrt[4]{\varepsilon\mathbb{1}_d + \hat{\nu}(t-1)} \odot (\boldsymbol{w}(t) - \boldsymbol{w}(t-1)) \right\|^2 = \mathcal{O}\left( \frac{1}{t} \right).
$$

Due to Eq. (56), we further have $\|\nabla\mathcal{L}(\boldsymbol{w}(t))\| = \mathcal{O}(t^{-1})$, which indicates

$$
\|\boldsymbol{w}(t)\| \leq \|\boldsymbol{w}(1)\| + \sum_{s=1}^{t} \|\boldsymbol{w}(s+1) - \boldsymbol{w}(s)\| = \|\boldsymbol{w}(1)\| + \eta \sum_{s=1}^{t} \left\| \frac{\hat{m}(s)}{\sqrt{\hat{\nu}(s) + \varepsilon\mathbb{1}_d}} \right\|
$$

$$
\leq \|\boldsymbol{w}(1)\| + \frac{\eta}{\sqrt{\varepsilon}} \sum_{s=1}^{t} \|\hat{m}(s)\| = \|\boldsymbol{w}(1)\| + \frac{\eta}{\sqrt{\varepsilon}} \sum_{s=1}^{t} \frac{1}{(1-\beta^s)} \left\| \sum_{i=1}^{s} \beta^{s-i} \nabla\mathcal{L}(\boldsymbol{w}(i)) \right\|
$$

$$
\leq \|\boldsymbol{w}(1)\| + \frac{\eta}{\sqrt{\varepsilon}(1-\beta)} \sum_{s=1}^{t} \sum_{i=1}^{s} \beta^{s-i} \|\nabla\mathcal{L}(\boldsymbol{w}(i))\|
$$

$$
\leq \|\boldsymbol{w}(1)\| + \frac{\eta}{\sqrt{\varepsilon}(1-\beta)^2} \sum_{s=1}^{t} \|\nabla\mathcal{L}(\boldsymbol{w}(s))\| = \mathcal{O}(\log(t)).
$$

Therefore, for any $\tilde{\boldsymbol{x}} \in \mathcal{T}(\boldsymbol{S})$, we have $\langle \boldsymbol{w}(t), \tilde{\boldsymbol{x}} \rangle = \mathcal{O}(\log(t))$, which by $\ell$ is exponential-tailed leads to $\ell(\langle \boldsymbol{w}(t), \tilde{\boldsymbol{x}} \rangle) = \Omega(t^{-1})$, and thus $\mathcal{L}(\boldsymbol{w}(t)) = \Theta(t^{-1})$. Also, since $\mathcal{L}(\boldsymbol{w}(t)) = \mathcal{O}(t^{-1})$, we have $\langle \boldsymbol{w}(t), \tilde{\boldsymbol{x}} \rangle = \Omega(\log(t))$, which further leads to $\|\boldsymbol{w}(t)\| = \Omega(\log(t))$, and thus $\|\boldsymbol{w}(t)\| = \Theta(\log(t))$.

Finally, we have

$$
\gamma \sum_{s=1}^{t} \beta^{t-s} \|\nabla \mathcal{L}(\boldsymbol{w}(s))\| = \frac{\gamma}{N} \sum_{s=1}^{t} \beta^{t-s} \left\| \sum_{\tilde{\boldsymbol{x}} \in \mathcal{T}(\boldsymbol{S})} \ell'(\langle \boldsymbol{w}(s), \tilde{\boldsymbol{x}} \rangle) \tilde{\boldsymbol{x}} \right\|
$$

$$
\leq -\frac{\gamma}{N} \sum_{s=1}^{t} \beta^{t-s} \sum_{\tilde{\boldsymbol{x}} \in \mathcal{T}(\boldsymbol{S})} \ell'(\langle \boldsymbol{w}(s), \tilde{\boldsymbol{x}} \rangle) \leq -\frac{1}{N} \sum_{s=1}^{t} \beta^{t-s} \left\langle \sum_{\tilde{\boldsymbol{x}} \in \mathcal{T}(\boldsymbol{S})} \ell'(\langle \boldsymbol{w}(s), \tilde{\boldsymbol{x}} \rangle) \tilde{\boldsymbol{x}}, \gamma \hat{\boldsymbol{w}} \right\rangle
$$

$$
\leq \left\| \sum_{s=1}^{t} \beta^{t-s} \nabla \mathcal{L}(\boldsymbol{w}(s)) \right\| \|\gamma \hat{\boldsymbol{w}}\| = \|\boldsymbol{m}(t)\| \leq \sum_{s=1}^{t} \beta^{t-s} \|\nabla \mathcal{L}(\boldsymbol{w}(s))\|,
$$

which leads to $\|\boldsymbol{m}(t)\| = \Theta(t^{-1})$. Similarly, we have $\nu(t) = \mathcal{O}(t^{-2})$, component-wisely. As $\lim_{t \to \infty} \beta_1^t = 0$ and $\lim_{t \to \infty} \beta_2^t = 0$, we have

$$
\|\boldsymbol{w}(t) - \boldsymbol{w}(t-1)\| = \left\| \frac{\hat{\boldsymbol{m}}(t)}{\sqrt{\varepsilon \mathbb{1}_d + \hat{\nu}(t)}} \right\| = \Theta(t^{-1}).
$$

The proof is completed. □

### C.3 BOUNDING THE ORTHOGONAL PART

By Lemma 8, there exists a solution $\tilde{\boldsymbol{w}}$ as the solution of Eq. (12) with $C_2 = \frac{\eta}{(1-\beta)\sqrt{\varepsilon}}$. Define $\boldsymbol{r}(t)$ as

$$
\boldsymbol{r}(t) \triangleq \boldsymbol{w}(t) - \log(t)\hat{\boldsymbol{w}} - \tilde{\boldsymbol{w}}, \tag{59}
$$

and we only need to prove $\|\boldsymbol{r}(t)\|$ is bounded over time. We then prove Lemma 7, providing an equivalent condition of $\|\boldsymbol{r}(t)\|$ being bounded. As the GDM and SGDM case, we separate the proof into two sub-lemmas.

**Lemma 21.** *Let all conditions in Theorem 4 hold. Then, $\|\boldsymbol{r}(t)\|$ is bounded if and only if $g(t)$ is upper bounded.*

*Proof.* Following the same routine as Lemma 2 and Lemma 17, we only need to prove

$$
\lim_{t \to \infty} \left\| (1 - \beta_1^{t-1}) \sqrt{\varepsilon \mathbb{1}_d + \hat{\nu}(t-1)} \odot (\boldsymbol{w}(t) - \boldsymbol{w}(t-1)) \right\| = 0, \tag{60}
$$

and

$$
\sum_{\tau=2}^{\infty} \left| \langle \boldsymbol{r}(\tau) - \boldsymbol{r}(\tau-1), (1 - \beta_1^{\tau-1}) \sqrt{\varepsilon \mathbb{1}_d + \hat{\nu}(\tau-1)} \odot (\boldsymbol{w}(\tau) - \boldsymbol{w}(\tau-1)) \rangle \right| < \infty. \tag{61}
$$

As for Eq. (60), by Lemma 6, we have

$$
\left\| (1 - \beta_1^{t-1}) \sqrt{\varepsilon \mathbb{1}_d + \hat{\nu}(t-1)} \odot (\boldsymbol{w}(t) - \boldsymbol{w}(t-1)) \right\|
$$
$$
= \mathcal{O}(t^{-1}) = \boldsymbol{o}(1).
$$

As for Eq. (60), we have

$$
\left| \left\langle \boldsymbol{r}(\tau) - \boldsymbol{r}(\tau-1), (1 - \beta_1^{\tau-1}) \sqrt{\varepsilon \mathbb{1}_d + \hat{\nu}(\tau-1)} \odot (\boldsymbol{w}(\tau) - \boldsymbol{w}(\tau-1)) \right\rangle \right|
$$

$$
= \left| \left\langle \boldsymbol{w}(\tau) - \boldsymbol{w}(\tau-1) - \log \frac{\tau}{\tau-1} \hat{\boldsymbol{w}}, (1 - \beta_1^{\tau-1}) \sqrt{\varepsilon \mathbb{1}_d + \hat{\nu}(\tau-1)} \odot (\boldsymbol{w}(\tau) - \boldsymbol{w}(\tau-1)) \right\rangle \right|
$$

$$
\leq \left| \left\langle \log \frac{\tau}{\tau-1} \hat{\boldsymbol{w}}, (1 - \beta_1^{\tau-1}) \sqrt{\varepsilon \mathbb{1}_d + \hat{\nu}(\tau-1)} \odot (\boldsymbol{w}(\tau) - \boldsymbol{w}(\tau-1)) \right\rangle \right|
$$

$$
+ (1 - \beta_1^{\tau-1}) \left\| \sqrt[4]{\varepsilon \mathbb{1}_d + \hat{\nu}(\tau-1)} \odot (\boldsymbol{w}(\tau) - \boldsymbol{w}(\tau-1)) \right\|^2
$$

$$
\overset{(\star)}{=} \mathcal{O}(\tau^{-2}),
$$

where Eq. $(\star)$ is due to Lemma 6 and $\log(\frac{\tau}{\tau-1}) = \Theta(\tau^{-1})$.

The proof is completed. □

We conclude the proof of Theorem 4 by showing $g(t)$ is upper bounded.

**Lemma 22.** *Let all conditions in Theorem 4 hold. Then, $g(t)$ is upper bounded.*

*Proof.* $g(t)$ is upper bounded is equivalent to $\sum_{t=1}^{\infty} g(t+1) - g(t) < \infty$. We then prove this lemma by calculating $g(t+1) - g(t)$ directly.

$$g(t+1) - g(t)$$

$$= \frac{\sqrt{\varepsilon}}{2} \|\boldsymbol{r}(t+1)\|^2 + \frac{\beta_1}{1-\beta_1} \left\langle \boldsymbol{r}(t+1), (1-\beta_1^t)\sqrt{\varepsilon \mathbb{1}_d + \hat{\nu}(t)} \odot (\boldsymbol{w}(t+1) - \boldsymbol{w}(t)) \right\rangle$$

$$- \left( \frac{\sqrt{\varepsilon}}{2} \|\boldsymbol{r}(t)\|^2 + \frac{\beta_1}{1-\beta_1} \left\langle \boldsymbol{r}(t), (1-\beta_1^{t-1})\sqrt{\varepsilon \mathbb{1}_d + \hat{\nu}(t-1)} \odot (\boldsymbol{w}(t) - \boldsymbol{w}(t-1)) \right\rangle \right)$$

$$- \frac{\beta_1}{1-\beta_1} \langle \boldsymbol{r}(t+1) - \boldsymbol{r}(t), (1-\beta_1^t)\sqrt{\varepsilon \mathbb{1}_d + \hat{\nu}(t)} \odot (\boldsymbol{w}(t+1) - \boldsymbol{w}(t)) \rangle$$

$$= \frac{\sqrt{\varepsilon}}{2} \|\boldsymbol{r}(t+1) - \boldsymbol{r}(t)\|^2 + \frac{\beta_1}{1-\beta_1} \left\langle \boldsymbol{r}(t), (1-\beta_1^t)\sqrt{\varepsilon \mathbb{1}_d + \hat{\nu}(t)} \odot (\boldsymbol{w}(t+1) - \boldsymbol{w}(t)) \right.$$

$$\left. -(1-\beta_1^{t-1})\sqrt{\varepsilon \mathbb{1}_d + \hat{\nu}(t-1)} \odot (\boldsymbol{w}(t) - \boldsymbol{w}(t-1)) \right\rangle + \sqrt{\varepsilon} \langle \boldsymbol{r}(t+1) - \boldsymbol{r}(t), \boldsymbol{r}(t) \rangle$$

$$\overset{(\star)}{=} \left\langle \boldsymbol{r}(t), -(1-\beta_1^t)\sqrt{\varepsilon \mathbb{1}_d + \hat{\nu}(t)} \odot (\boldsymbol{w}(t+1) - \boldsymbol{w}(t)) - \frac{\eta}{1-\beta}\nabla\mathcal{L}(\boldsymbol{w}(t)) \right\rangle$$

$$+ \frac{\sqrt{\varepsilon}}{2} \|\boldsymbol{r}(t+1) - \boldsymbol{r}(t)\|^2 + \sqrt{\varepsilon} \langle \boldsymbol{r}(t+1) - \boldsymbol{r}(t), \boldsymbol{r}(t) \rangle,$$

where Eq. $(\star)$ is due to a simple rearranging of the update rule of Adam, i.e.,

$$\frac{\beta_1}{1-\beta_1} \left( (1-\beta_1^t)\sqrt{\varepsilon \mathbb{1}_d + \hat{\nu}(t)} \odot (\boldsymbol{w}(t+1) - \boldsymbol{w}(t)) - (1-\beta_1^{t-1})\sqrt{\varepsilon \mathbb{1}_d + \hat{\nu}(t-1)} \odot (\boldsymbol{w}(t) - \boldsymbol{w}(t-1)) \right)$$

$$= -\frac{\eta}{1-\beta_1}\nabla\mathcal{L}(\boldsymbol{w}(t)) - (1-\beta_1^t)\sqrt{\varepsilon \mathbb{1}_d + \hat{\nu}(t)} \odot (\boldsymbol{w}(t+1) - \boldsymbol{w}(t)).$$

On the one hand, as $\|\boldsymbol{r}(t+1) - \boldsymbol{r}(t)\| = \|\boldsymbol{w}(t+1) - \boldsymbol{w}(t) - \log\frac{t+1}{t}\hat{\boldsymbol{w}}\| = \mathcal{O}(t^{-1})$,

$$\sum_{t=1}^{\infty} \frac{\sqrt{\varepsilon}}{2} \|\boldsymbol{r}(t+1) - \boldsymbol{r}(t)\|^2 < \infty.$$

On the other hand,

$$\left\langle \boldsymbol{r}(t), -(1-\beta_1^t)\sqrt{\varepsilon \mathbb{1}_d + \hat{\nu}(t)} \odot (\boldsymbol{w}(t+1) - \boldsymbol{w}(t)) - \frac{\eta}{1-\beta}\nabla\mathcal{L}(\boldsymbol{w}(t)) \right\rangle$$

$$+ \sqrt{\varepsilon} \langle \boldsymbol{r}(t+1) - \boldsymbol{r}(t), \boldsymbol{r}(t) \rangle$$

$$= \left\langle \boldsymbol{r}(t), -(1-\beta_1^t)\sqrt{\varepsilon \mathbb{1}_d + \hat{\nu}(t)} \odot (\boldsymbol{w}(t+1) - \boldsymbol{w}(t)) - \frac{\eta}{1-\beta}\nabla\mathcal{L}(\boldsymbol{w}(t)) \right\rangle$$

$$+ \sqrt{\varepsilon} \left\langle \boldsymbol{w}(t+1) - \boldsymbol{w}(t) - \log\left(\frac{t+1}{t}\right)\hat{\boldsymbol{w}}, \boldsymbol{r}(t) \right\rangle$$

$$= \left\langle \boldsymbol{r}(t), -(1-\beta_1^t)\sqrt{\varepsilon \mathbb{1}_d + \hat{\nu}(t)} \odot (\boldsymbol{w}(t+1) - \boldsymbol{w}(t)) + \sqrt{\varepsilon}(\boldsymbol{w}(t+1) - \boldsymbol{w}(t)) \right\rangle$$

$$+ \left\langle \boldsymbol{r}(t), -\sqrt{\varepsilon}\log\left(\frac{t+1}{t}\right)\hat{\boldsymbol{w}} - \frac{\eta}{1-\beta}\nabla\mathcal{L}(\boldsymbol{w}(t)) \right\rangle$$

$$\overset{(\bullet)}{=} \mathcal{O}(\beta_1^t + t^{-2}) + \left\langle \boldsymbol{r}(t), -\sqrt{\varepsilon}\log\left(\frac{t+1}{t}\right)\hat{\boldsymbol{w}} - \frac{\eta}{1-\beta}\nabla\mathcal{L}(\boldsymbol{w}(t)) \right\rangle,$$

where Eq. $(\bullet)$ is due to $\hat{\nu}(t) = \mathcal{O}(t^{-2})$.

Furthermore, following exactly the same routine as Lemma 16, we have

$$\sum_{t=1}^{\infty} \left\langle \boldsymbol{r}(t), -\sqrt{\varepsilon}\log\left(\frac{t+1}{t}\right)\hat{\boldsymbol{w}} - \frac{\eta}{1-\beta}\nabla\mathcal{L}(\boldsymbol{w}(t)) \right\rangle < \infty.$$

The proof is completed. $\qquad\square$

# D  APPLICATIONS&EXTENSIONS

## D.1  APPLICATION TO THE MINI-BATCH SGDM

This section provides formal description of the implicit bias of mini-batch SGDM and its corresponding proof. To begin with, we would like to provide a formal definition of mini-batch SGDM. Mini-batch SGDM differs from SGDM by applying sampling without replacement to obtain $\boldsymbol{B}(t)$ in Eq. (1). Specifically, let $K = \frac{N}{b}$. For any $T \geq 0$, we call time series $\{KT + 1, \cdots, KT + K\}$ the $(T+1)$-th epoch, and during the $T + 1$-th epoch, the dataset $\boldsymbol{S}$ is randomly uniformly divided into $K$ parts $\{\boldsymbol{B}(KT + 1), \cdots, \boldsymbol{B}(KT + K)\}$, with $\bigcup_{t=KT+1}^{KT+K} \boldsymbol{B}(t) = \boldsymbol{S}$. The implicit bias of mini-batch SGDM is then stated as the following theorem:

**Theorem 6.** *Let Assumptions 1, 2, and 3. (S) hold. Let learning rate $\eta$ be small enough, and $\beta \in [0, 1)$. Then, for almost every dataset $\boldsymbol{S}$, mini-batch SGDM satisfies $\boldsymbol{w}(t) - \log(t)\hat{\boldsymbol{w}}$ is bounded as $t \to \infty$, and $\lim_{t\to\infty} \frac{\boldsymbol{w}(t)}{\|\boldsymbol{w}(t)\|} = \frac{\hat{\boldsymbol{w}}}{\|\hat{\boldsymbol{w}}\|}$.*

The without-replacement sampling method leads to the direction of every trajectory of mini-SGDM converge to the max-margin solution, compared to the same conclusion holds for SGDM a.s.. We prove the theorem following the same framework of GDM, by proceeding with two stages.

**Stage I.** The following lemma proves $\mathcal{L}(\boldsymbol{u}(t))$ is an Lyapunov function for mini-batch SGDM and without the a.s. condition.

**Lemma 23.** *Let all conditions in Theorem 6 hold. Then, we have*

$$\mathcal{L}(\boldsymbol{u}(t + 1)) \leq \mathcal{L}(\boldsymbol{u}(1)) - \Omega(\eta) \sum_{s=1}^{t} \|\nabla \mathcal{L}(\boldsymbol{w}(s))\|^2.$$

*Proof.* By the Taylor Expansion of $\mathcal{L}(\boldsymbol{u}(t + 1))$ at $\boldsymbol{u}(t)$, we have

$$\mathcal{L}(\boldsymbol{u}(KT + T + 1))$$
$$\leq \mathcal{L}(\boldsymbol{u}(KT + 1)) - \tilde{\eta} \left\langle \nabla \mathcal{L}(\boldsymbol{u}(KT + 1)), \sum_{t=1}^{K} \nabla \mathcal{L}_{\boldsymbol{B}(t+KT)}(\boldsymbol{w}(t + KT)) \right\rangle$$
$$+ \frac{H\tilde{\eta}^2}{2} \left\| \sum_{t=1}^{K} \mathcal{L}_{\boldsymbol{B}(t+KT)}(\boldsymbol{w}(t + KT)) \right\|^2. \tag{62}$$

On the other hand, for any $t \in \{2, \cdots, K\}$, we have

$$\boldsymbol{w}(KT + t) - \boldsymbol{w}(KT + 1) = \eta \sum_{s=1}^{t} \left( \sum_{\ell=1}^{KT+s} \beta^{KT+s-\ell} \nabla \mathcal{L}_{\boldsymbol{B}(\ell)}(\boldsymbol{w}(\ell)) \right)$$

$$= \eta \sum_{s=1}^{t} \left( \sum_{\ell=KT+1}^{KT+s} \beta^{KT+s-\ell} \nabla \mathcal{L}_{\boldsymbol{B}(\ell)}(\boldsymbol{w}(\ell)) \right) + \eta \sum_{s=1}^{t} \left( \sum_{\ell=1}^{KT} \beta^{KT+s-\ell} \nabla \mathcal{L}_{\boldsymbol{B}(\ell)}(\boldsymbol{w}(\ell)) \right)$$

$$= \eta \sum_{\ell=1}^{t} \frac{1 - \beta^{t-\ell+1}}{1 - \beta} \nabla \mathcal{L}_{\boldsymbol{B}(KT+\ell)}(\boldsymbol{w}(KT + \ell)) + \eta \frac{\beta(1 - \beta^t)}{1 - \beta} \sum_{\ell=1}^{KT} \beta^{KT-\ell} \nabla \mathcal{L}_{\boldsymbol{B}(\ell)}(\boldsymbol{w}(\ell))$$

$$= \eta \sum_{\ell=1}^{t} \frac{1 - \beta^{t-\ell+1}}{1 - \beta} \nabla \mathcal{L}_{\boldsymbol{B}(KT+\ell)}(\boldsymbol{w}(KT + \ell)) - \eta \sum_{\ell=1}^{t} \frac{1 - \beta^{t-\ell+1}}{1 - \beta} \nabla \mathcal{L}_{\boldsymbol{B}(KT+\ell)}(\boldsymbol{w}(KT + 1))$$

$$+ \eta \frac{\beta(1 - \beta^t)}{1 - \beta} \sum_{\ell=1}^{KT} \beta^{KT-\ell} \nabla \mathcal{L}_{\boldsymbol{B}(\ell)}(\boldsymbol{w}(\ell)) + \eta \sum_{\ell=1}^{t} \frac{1 - \beta^{t-\ell+1}}{1 - \beta} \nabla \mathcal{L}_{\boldsymbol{B}(KT+\ell)}(\boldsymbol{w}(KT + 1)),$$

which by $\eta$ is small enough further indicates

$$
\begin{aligned}
&\|\boldsymbol{w}(KT + t) - \boldsymbol{w}(KT + 1)\| \\
\leq& \eta \left\| \sum_{\ell=1}^{t} \frac{1 - \beta^{t-\ell+1}}{1 - \beta} \nabla \mathcal{L}_{\boldsymbol{B}(KT+\ell)}(\boldsymbol{w}(KT + \ell)) - \sum_{\ell=1}^{t} \frac{1 - \beta^{t-\ell+1}}{1 - \beta} \nabla \mathcal{L}_{\boldsymbol{B}(KT+\ell)}(\boldsymbol{w}(KT + 1)) \right\| \\
&+ \eta \left\| \frac{\beta(1 - \beta^t)}{1 - \beta} \sum_{\ell=1}^{KT} \beta^{KT-\ell} \nabla \mathcal{L}_{\boldsymbol{B}(\ell)}(\boldsymbol{w}(\ell)) \right\| + \eta \left\| \sum_{\ell=1}^{t} \frac{1 - \beta^{t-\ell+1}}{1 - \beta} \nabla \mathcal{L}_{\boldsymbol{B}(KT+\ell)}(\boldsymbol{w}(KT + 1)) \right\| \\
=& \mathcal{O}(\eta) \sum_{\ell=2}^{t} \|\boldsymbol{w}(KT + \ell) - \boldsymbol{w}(KT + 1)\| + \mathcal{O}(\eta) \left( \sum_{\ell=1}^{KT} \beta^{KT-\ell} \|\nabla \mathcal{L}_{\boldsymbol{B}(\ell)}(\boldsymbol{w}(\ell))\| \right) \\
&+ \mathcal{O}(\eta) \|\nabla \mathcal{L}(\boldsymbol{w}(KT + 1))\|.
\end{aligned}
$$

Applying the same analysis to $\|\boldsymbol{w}(KT + t - 1) - \boldsymbol{w}(KT + 1)\|$ recursively, we finally obtain

$$
\begin{aligned}
&\|\boldsymbol{w}(KT + t) - \boldsymbol{w}(KT + 1)\| \\
\leq& \mathcal{O}(\eta) \left( \sum_{\ell=1}^{KT} \beta^{KT-\ell} \|\nabla \mathcal{L}_{\boldsymbol{B}(\ell)}(\boldsymbol{w}(\ell))\| \right) + \mathcal{O}(\eta) \|\nabla \mathcal{L}(\boldsymbol{w}(KT + 1))\|.
\end{aligned} \tag{63}
$$

Applying Eq. (63) to the $\|\nabla \mathcal{L}_{\boldsymbol{B}(\ell)}(\boldsymbol{w}(\ell))\|$ in Eq. (63) ($\forall \ell \in [1, KT]$) iterative and choosing $\eta$ to be small enough, we further have

$$
\begin{aligned}
&\|\boldsymbol{w}(KT + t) - \boldsymbol{w}(KT + 1)\| \\
\leq& \mathcal{O}(\eta) \left( \sum_{\ell=0}^{T-1} \sqrt{\beta^{K(T-\ell)}} \|\nabla \mathcal{L}_{\boldsymbol{B}(K\ell+1)}(\boldsymbol{w}(K\ell + 1))\| \right) + \mathcal{O}(\eta) \|\nabla \mathcal{L}(\boldsymbol{w}(KT + 1))\| \\
=& \mathcal{O}(\eta) \left( \sum_{\ell=0}^{T} \sqrt{\beta^{K(T-\ell)}} \|\nabla \mathcal{L}_{\boldsymbol{B}(K\ell+1)}(\boldsymbol{w}(K\ell + 1))\| \right).
\end{aligned}
$$

Therefore,

$$
\begin{aligned}
&\sum_{t=1}^{K} \nabla \mathcal{L}_{\boldsymbol{B}(t+KT)}(\boldsymbol{w}(t + KT)) \\
=& \sum_{t=1}^{K} \nabla \mathcal{L}_{\boldsymbol{B}(t+KT)}(\boldsymbol{w}(t)) + \mathcal{O} \left( \eta \left( \sum_{\ell=0}^{T} \sqrt{\beta^{K(T-\ell)}} \|\nabla \mathcal{L}_{\boldsymbol{B}(K\ell+1)}(\boldsymbol{w}(K\ell + 1))\| \right) \right) \\
=& K \nabla \mathcal{L}(\boldsymbol{w}(t)) + \mathcal{O} \left( \eta \left( \sum_{\ell=0}^{T} \sqrt{\beta^{K(T-\ell)}} \|\nabla \mathcal{L}_{\boldsymbol{B}(K\ell+1)}(\boldsymbol{w}(K\ell + 1))\| \right) \right).
\end{aligned} \tag{64}
$$

Similarly, one can obtain

$$
\begin{aligned}
&\nabla \mathcal{L}(\boldsymbol{u}(KT + 1)) \\
=& \nabla \mathcal{L}(\boldsymbol{w}(KT + 1)) + \mathcal{O}(\|\boldsymbol{w}(KT + 1) - \boldsymbol{w}(KT)\|) \\
=& \nabla \mathcal{L}(\boldsymbol{w}(KT + 1)) + \mathcal{O} \left( \eta \left( \sum_{\ell=0}^{T} \sqrt{\beta^{K(T-\ell)}} \|\nabla \mathcal{L}_{\boldsymbol{B}(K\ell+1)}(\boldsymbol{w}(K\ell + 1))\| \right) \right).
\end{aligned} \tag{65}
$$

Applying Eq. (64) and Eq. (65) back to the Taylor Expansion (Eq. (62)), we have

$$
\begin{aligned}
&\mathcal{L}(\boldsymbol{u}(KT+T+1))\\
&\leq \mathcal{L}(\boldsymbol{u}(KT+1)) - \Omega(\eta)\langle \nabla\mathcal{L}(\boldsymbol{w}(KT+1)), \nabla\mathcal{L}(\boldsymbol{w}(KT+1))\rangle\\
&\quad + \mathcal{O}\left(\eta^2\left(\sum_{\ell=0}^{T}\sqrt{\beta^{K(T-\ell)}}\left\|\nabla\mathcal{L}_{\boldsymbol{B}(K\ell+1)}(\boldsymbol{w}(K\ell+1))\right\|\right)^2\right)\\
&\leq \mathcal{L}(\boldsymbol{u}(KT+1)) - \Omega(\eta)\langle \nabla\mathcal{L}(\boldsymbol{w}(KT+1)), \nabla\mathcal{L}(\boldsymbol{w}(KT+1))\rangle\\
&\quad + \mathcal{O}\left(\eta^2\left(\sum_{\ell=0}^{T}\sqrt{\beta^{K(T-\ell)}}\left\|\nabla\mathcal{L}_{\boldsymbol{B}(K\ell+1)}(\boldsymbol{w}(K\ell+1))\right\|^2\right)\right).
\end{aligned}
$$

Summing the above inequality over $T$ and setting $\eta$ small enough leads to the conclusion.

The proof is completed. $\qquad\square$

## D.2 EXTENSION TO THE MULTI-CLASS CLASSIFICATION PROBLEM

Here we use several notations and lemmas from (Soudry et al., 2018). We define $\boldsymbol{w} = \mathrm{vec}(\boldsymbol{W})$, $\hat{\boldsymbol{w}} = \mathrm{vec}(\hat{\boldsymbol{W}})$, $\boldsymbol{e}_i \in \mathbb{R}^C$ ($i \in \{1, \cdots, C\}$) satisfying $(\boldsymbol{e}_i)_j = \delta_{ij}$, and $\boldsymbol{A}_i = \boldsymbol{e}_i \otimes \mathbb{I}_{d_X}$, where $\mathbb{I}_{d_X}$ is the identity matrix with dimension $d_X$. We still consider the normalized data, i.e., $\|\boldsymbol{x}\| \leq 1$, $\forall(\boldsymbol{x}, \boldsymbol{y}) \in \boldsymbol{S}$. Then, the individual loss of sample $(\boldsymbol{x}, \boldsymbol{y})$ can be then represented as

$$
\ell(\boldsymbol{y}, \boldsymbol{W}\boldsymbol{x}) = \log\frac{e^{\langle\boldsymbol{w}, \boldsymbol{A_y}\boldsymbol{x}\rangle}}{\sum_{i=1}^{C}e^{\langle\boldsymbol{w}, \boldsymbol{A_i}\boldsymbol{x}\rangle}}.
$$

Furthermore, the gradient of training error at $\boldsymbol{W}$ has the form

$$
\nabla\mathcal{L}(\boldsymbol{w}) = \frac{1}{N}\sum_{(\boldsymbol{x}, \boldsymbol{y})\in\boldsymbol{S}}\sum_{i=1}^{C}\frac{1}{\sum_{j=1}^{C}e^{\langle\boldsymbol{w}, (\boldsymbol{A}_j-\boldsymbol{A}_i)\boldsymbol{x}\rangle}}(\boldsymbol{A}_i-\boldsymbol{A_y})\boldsymbol{x}.
$$

and the Hessian matrix of $\mathcal{L}$ can be represented as

$$
\mathcal{H}\mathcal{L}(\boldsymbol{w}) = \frac{1}{N}\sum_{(\boldsymbol{x}, \boldsymbol{y})\in\boldsymbol{S}}\sum_{i=1}^{C}\frac{\sum_{j=1}^{C}e^{\langle\boldsymbol{w}, (\boldsymbol{A}_j-\boldsymbol{A}_i)\boldsymbol{x}\rangle}}{\left(\sum_{j=1}^{C}e^{\langle\boldsymbol{w}, (\boldsymbol{A}_j-\boldsymbol{A}_i)\boldsymbol{x}\rangle}\right)^2}(\boldsymbol{A}_i-\boldsymbol{A_y})\boldsymbol{x}((\boldsymbol{A}_j-\boldsymbol{A}_i)\boldsymbol{x})^\top,
$$

one can then easily verify all absolute value of the eigenvalues of $\mathcal{H}\mathcal{L}(\boldsymbol{w})$ is no larger than 2, which indicates $\mathcal{L}$ is 2-globally smooth.

On the other hand, the separable assumption leads to $\langle\hat{\boldsymbol{w}}, (\boldsymbol{A_y}-\boldsymbol{A}_i)\boldsymbol{x}\rangle > 0$, $\forall \boldsymbol{y} \neq i$, which further indicates

$$
\langle\nabla\mathcal{L}(\boldsymbol{w}), \hat{\boldsymbol{w}}\rangle > 0.
$$

Let $\gamma = \frac{1}{\|\hat{\boldsymbol{w}}\|}$, following the similar routine as the binary case, we have for a random subset of $\boldsymbol{S}$ sampled uniformly without replacement with size $b$, we have

$$
\|\nabla\mathcal{L}(\boldsymbol{w})\|^2 \leq \mathbb{E}_{\boldsymbol{B}(t)}\|\nabla\mathcal{L}_{\boldsymbol{B}(t)}(\boldsymbol{w})\|^2 \leq \frac{2N}{\gamma b^2}\|\nabla\mathcal{L}(\boldsymbol{w})\|^2. \tag{66}
$$

Similarly, we have for any positive real series $\{a_t\}_{t=t_1}^{t_2}$,

$$
\gamma\sum_{t=t_1}^{t_2}a(t)\|\nabla\mathcal{L}(\boldsymbol{w}(t))\| \leq \left\|\sum_{t=t_1}^{t_2}a(t)\nabla\mathcal{L}(\boldsymbol{w}(t))\right\| \leq \sum_{t=t_1}^{t_2}a(t)\|\nabla\mathcal{L}(\boldsymbol{w}(t))\|. \tag{67}
$$

The proofs of Stage I can then be obtained with Lyapunov functions unchanged and by replacing the corresponding lemmas using Eq. (66) and Eq. (67).

As for the proofs of Stage II, the Lyapunov functions are still the same, while we only need to prove the sum of $\langle\boldsymbol{r}(t), -\frac{\eta}{1-\beta}\nabla\mathcal{L}(\boldsymbol{w}(t)) - \log\frac{t+1}{t}\hat{\boldsymbol{w}}\rangle$ (for GDM, $\langle\boldsymbol{r}(t), -\frac{\eta}{1-\beta}\nabla\mathcal{L}_{\boldsymbol{B}(t)}(\boldsymbol{w}(t)) - $

$\frac{N}{bt} \sum_{i:\tilde{\boldsymbol{x}}_i \in \mathcal{T}(\boldsymbol{B}(t) \cap \boldsymbol{S}_s)} \boldsymbol{v}_i \tilde{\boldsymbol{x}}_i \rangle$ for SGDM, $\langle \boldsymbol{r}(t), -\sqrt{\varepsilon} \log\left(\frac{t+1}{t}\right) \hat{\boldsymbol{w}} - \frac{\eta}{1-\beta} \nabla \mathcal{L}(\boldsymbol{w}(t)) \rangle$ for Adam). For the multi-class case using GDM, We present the following lemma from (Soudry et al., 2018), while the other two cases can be proved similarly:

**Lemma 24** (Part of the proof of Lemma 20, (Soudry et al., 2018))**.** *If* $\langle \boldsymbol{w}(t), (\boldsymbol{A_y} - \boldsymbol{A_i})\boldsymbol{x} \rangle \to \infty$ *as* $t \to \infty$, $\forall (\boldsymbol{x}, \boldsymbol{y}) \in \boldsymbol{S}$ *and* $\forall i \neq \boldsymbol{y}$, *we have the sum of* $\langle \boldsymbol{r}(t), -\frac{\eta}{1-\beta} \nabla \mathcal{L}(\boldsymbol{w}(t)) - \log \frac{t+1}{t} \hat{\boldsymbol{w}} \rangle$ *is upper bounded.*

The proof of Theorem 5 is then completed.

# E    EXPERIMENTS

This section collects several experiments supporting our theoretical results.

## E.1    EXPERIMENTS OF LINEAR MODEL

### E.1.1    COMPARING THE TRAINING BEHAVIOR OF GD, GDM, AND ADAM (W/S)

The experiments in this section is designed to verify Theorem 2, i.e., with proper learning rates, gradient descent with momentum converges to the max margin solution, which is the same as gradient descent. We use the synthetic dataset as (Figure 1, (Soudry et al., 2018)) and run GD, GDM and Adam (w/s) over it with different learning rates $\eta = 0.1, 0.01$ and different random seeds (for random initialization and random samples despite the support sets $\{((1.5, 0.5), 1), ((0.5, 1.5), 1), ((-1.5, -0.5), -1), ((-1.5, -0.5), -1)\}$). Both the angle between the output parameter and max margin solution and the training accuracy are plotted in Figure 1. Specifically, we plot the results with learning rate (1). $\eta_{GD} = \eta_{GDM} = \eta_{Adam} = 0.001$, (2). $\eta_{GD} = \eta_{GDM} = \eta_{Adam} = 0.1$. We plot the training accuracy and the angle between the output parameter and the max margin solution for each setting respectively. The results are shown in Figure 1. The observations can be summarized as follows:

- When learning rate is small enough ($\eta = 0.1, 0.001$), both GD, GDM, and Adam converge to the max margin solution, which supports our theoretical results;
- (Similarity between GD and GDM) The training behaviors of GD and GDM are highly similar.
- (The acceleration effect of Adam) Adam achieves smaller angle with the max margin solution under the same number of iterations.

### E.1.2    COMPARING THE TRAINING BEHAVIOR OF SGD AND SGDM

We also run the stochastic optimizers (SGD and SGDM) on the same synthetic dataset as Figure 1. The learning rate is same as Figure 1, namely 0.001 and 0.1. The results of the experiment are plotted in Figure 2. It can be observed that when learning rate is small enough, SGD and SGDM have the similar training behavior, both converging to the max margin solution.

### E.1.3    ADAM ON ILL-POSED DATASET IN (SOUDRY ET AL., 2018)

In Figure 3 of (Soudry et al., 2018), an ill-posed synthetic dataset is proposed to support the argument "Adam does not converge to max margin solution", which contradicts to the theoretical results of this paper. We re-conduct the experiment of Figure 3 in (Soudry et al., 2018) with the same ill-posed synthetic dataset with different learning rates and different random seeds as Figure 3. Figure 3. (f) is similar to Figure 3 in (Soudry et al., 2018), where with learning rate $\eta = 0.1$ and random seed 1, the angle of GD to the max margin solution is smaller than Adam all the time. However, it can be observed from the amplified figure that the angle of GD keeps still and above 0 all the time, meaning that GD doesn't converge to the max margin solution under this setting. However, the angle of Adam to the max margin solution still keeps decreasing and it's unreasonable to claim "Adam doesn't converge to the max margin solution" in this case (the same issue exists in Figure 3 in (Soudry et al., 2018)). Also, as we mentioned at the beginning of this section, this dataset is ill-posed, which is due to the imbalance between the two components of the data (for all data $((x_1, x_2), y)$ in the dataset,

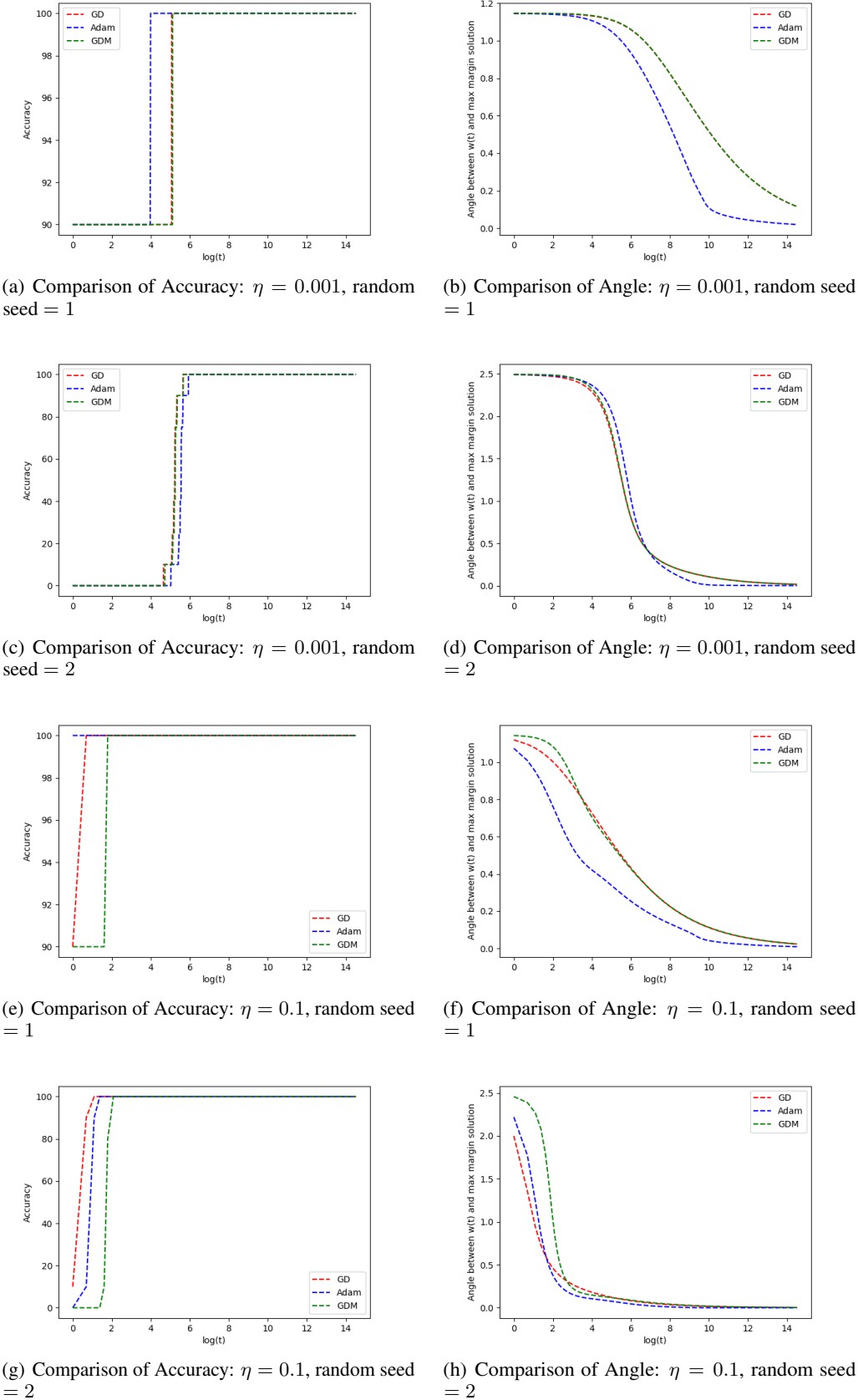

(a) Comparison of Accuracy: $\eta = 0.001$, random seed $= 1$

(b) Comparison of Angle: $\eta = 0.001$, random seed $= 1$

(c) Comparison of Accuracy: $\eta = 0.001$, random seed $= 2$

(d) Comparison of Angle: $\eta = 0.001$, random seed $= 2$

(e) Comparison of Accuracy: $\eta = 0.1$, random seed $= 1$

(f) Comparison of Angle: $\eta = 0.1$, random seed $= 1$

(g) Comparison of Accuracy: $\eta = 0.1$, random seed $= 2$

(h) Comparison of Angle: $\eta = 0.1$, random seed $= 2$

Figure 1: Comparison of GD, GDM, and Adam on the synthetic dataset in (Soudry et al., 2018). In (a-g). the GD curve coincides with the GDM curve.

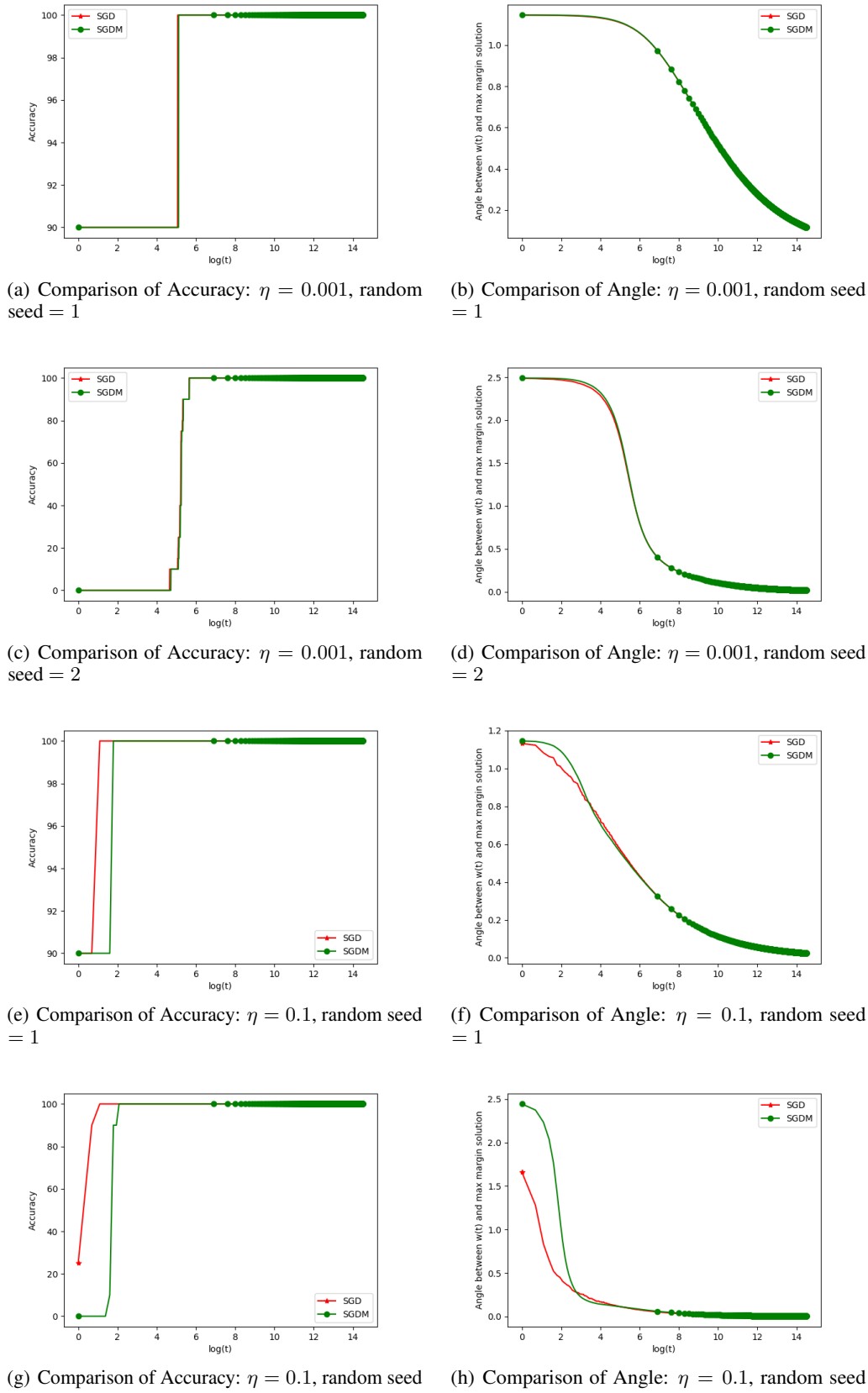

(a) Comparison of Accuracy: $\eta = 0.001$, random seed $= 1$

(b) Comparison of Angle: $\eta = 0.001$, random seed $= 1$

(c) Comparison of Accuracy: $\eta = 0.001$, random seed $= 2$

(d) Comparison of Angle: $\eta = 0.001$, random seed $= 2$

(e) Comparison of Accuracy: $\eta = 0.1$, random seed $= 1$

(f) Comparison of Angle: $\eta = 0.1$, random seed $= 1$

(g) Comparison of Accuracy: $\eta = 0.1$, random seed $= 2$

(h) Comparison of Angle: $\eta = 0.1$, random seed $= 2$

Figure 2: Comparison of SGD and SGDM on the synthetic dataset in (Soudry et al., 2018).

$|x_1|$ is always smaller than 2, while $|x_2|$ is larger than 10 (and even larger than 30 despite two data in the dataset)), which requires smaller learning rate. To tackle this problem, we need to tune down the learning rate. By Figure 3. (b),(d) and Figure 4. (b), after scaling down the learning rate, both GD's angle and Adam's angle keep decreasing.

### E.2 EVIDENCE IN THE DEEP NEURAL NETWORKS

We conduct an experiment on the MNIST dataset using the four layer convolutional networks used in (Lyu & Li, 2019; Wang et al., 2021) (first proposed by (Madry et al., 2018)) to verify whether SGD and SGDM still behave similarly in (homogeneous) deep neural networks. The learning rates of the optimizers are all set to be the default in Pytorch. The results can be seen in Figure 5. It can be observed that (1). SGDM achieves similar test accuracy compared to SGD while (2). SGDM converges faster than SGD.

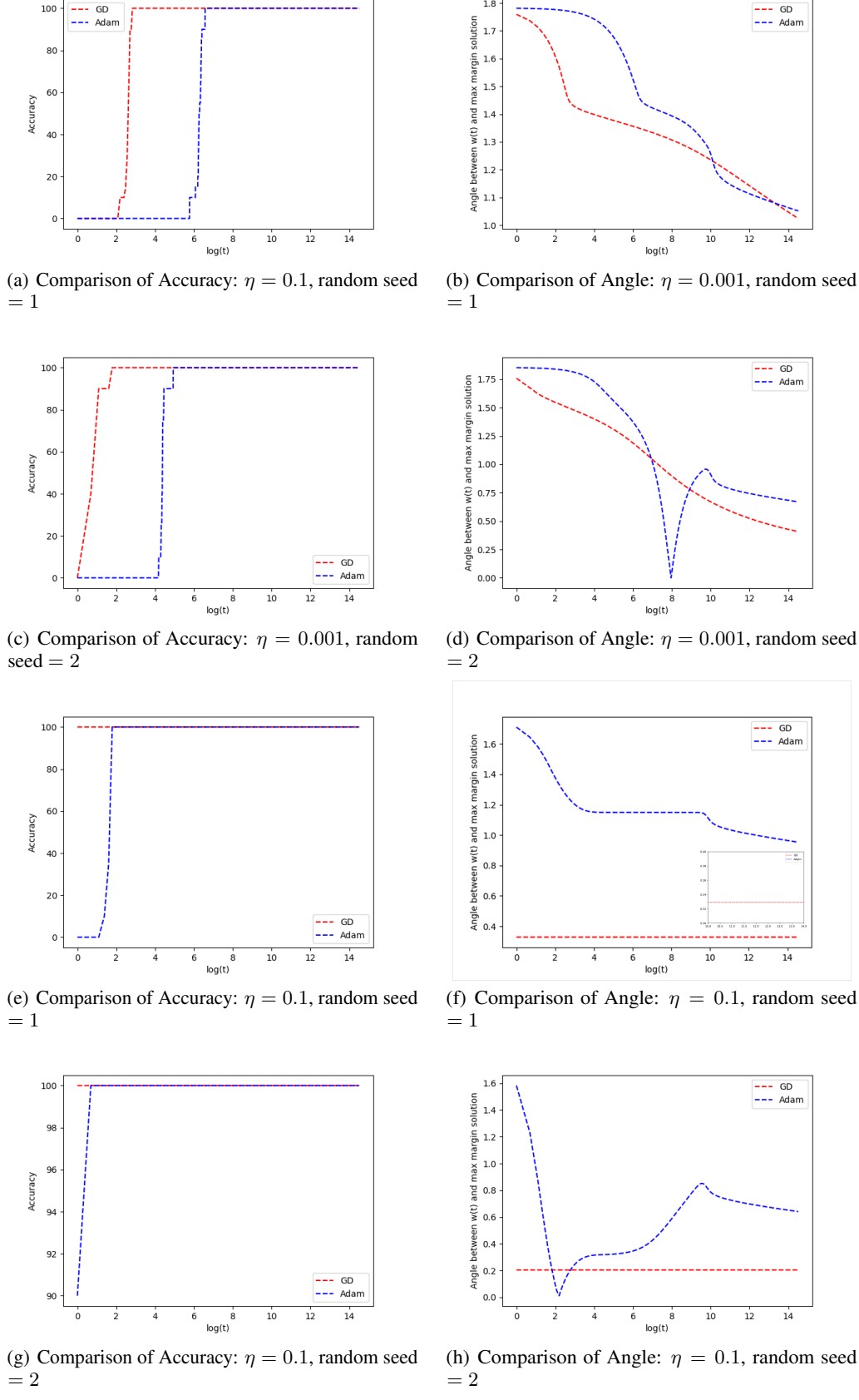

(a) Comparison of Accuracy: $\eta = 0.1$, random seed $= 1$

(b) Comparison of Angle: $\eta = 0.001$, random seed $= 1$

(c) Comparison of Accuracy: $\eta = 0.001$, random seed $= 2$

(d) Comparison of Angle: $\eta = 0.001$, random seed $= 2$

(e) Comparison of Accuracy: $\eta = 0.1$, random seed $= 1$

(f) Comparison of Angle: $\eta = 0.1$, random seed $= 1$

(g) Comparison of Accuracy: $\eta = 0.1$, random seed $= 2$

(h) Comparison of Angle: $\eta = 0.1$, random seed $= 2$

Figure 3: Comparison of GD and Adam on the ill-posed synthetic dataset in (Soudry et al., 2018).

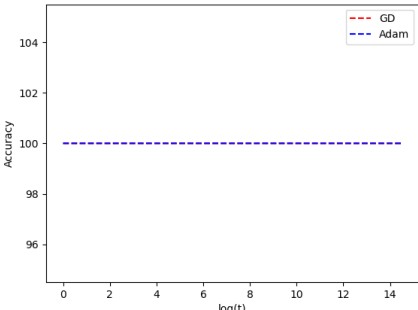 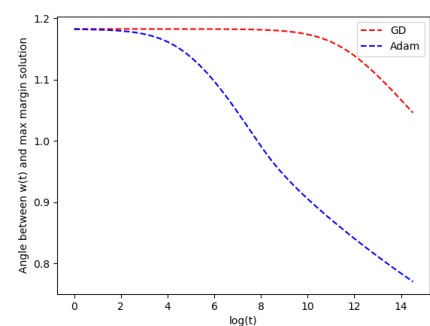

(a) Comparison of Accuracy: $\eta = 0.1$, random seed $= 3$

(b) Comparison of Angle: $\eta = 0.001$, random seed $= 3$

Figure 4: Comparison of GD and Adam on the ill-posed synthetic dataset in (Soudry et al., 2018) (continue).

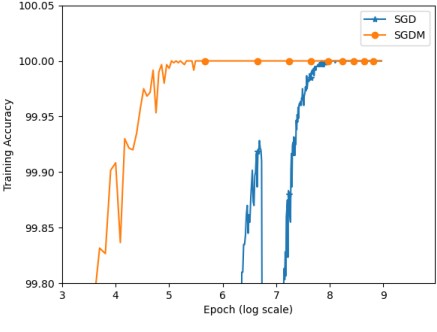 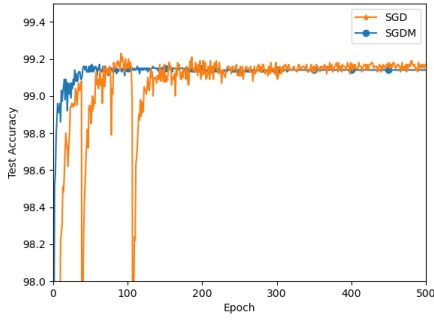

(a) Comparison of Training Accuracy

(b) Comparison of Test Accuracy

Figure 5: Comparison of SGD and SGDM on MNIST with four layer CNN.

