# OpenReview forum: "Momentum Doesn't Change The Implicit Bias"
_ICLR.cc/2022/Conference — ICLR 2022 Submitted_

### Official Review · Reviewer_5GXX · 2021-10-28

**Correctness:** 4
**Technical Novelty And Significance:** 4
**Empirical Novelty And Significance:** Not applicable
**Recommendation:** 8
**Confidence:** 4

**Main Review:**

I think this is a good paper. Although the model is very simple, the effect of momentum on implicit bias was not analyzed theoretically previously even in such a simple model.
The authors provide an analysis of SGD and ADAM with momentum using new Lyapunov functions, but I didn't check the proofs thoroughly.

However, I think the title is quite exaggerated. Claiming that “momentum doesn’t change the implicit bias” based on the simple linear model is not enough, and I think more evidence is needed to state that, at least empirically (e.g. for deep linear networks).
Also, I think it would be good to complement the paper with empirical results, even if some empirical results already appear in previous works. Specifically, since the theoretical results are only for simple linear model, it would be interesting to see empirically the effect of momentum for more complex models, e.g. deep linear/ReLU networks.

Questions:
-	What are the practical implication of the results ?  I think in practice momentum (with good hyperparameters) usually improves generalization, so I would expect that momentum has some effect on the implicit bias (at least in more complex networks).
-	It is shown in Figure 3 of Soudry et al. 2018 that ADAM does not converge to $L_2$ max-margin. This seems to contradict the results for ADAM in this paper. Can you comment on that ?
-	What is the effect of $\varepsilon$ in ADAM ?

Minor:
it will be good to define “almost every dataset”.


**Summary Of The Paper:**

In this paper the authors analyze the implicit bias of SGD with momentum and ADAM with momentum. For linear classification with exponential-tailed loss and separable data, the authors show that both algorithms converge to the $L_2$ max-margin solution, similarly to vanilla GD.

**Summary Of The Review:**

Overall, I think the paper is interesting. The authors push forward the understanding of implicit bias by analyzing the effect of momentum. The tools in this paper might be used to analyze more complex models. Therefore, I recommend for acceptance.

---

> ### Author Response · Authors · 2021-11-22
> **Thanks for your positive feedbacks!**
>
> We thank the reviewer for the positive feedback. Your concerns are correspondingly addressed  as follows:
>
> **Q1**: I think the title is quite exaggerated. More evidence is needed to state that, at least empirically (e.g. for deep linear networks).
>
> **A1**: Thanks for the comments. We have changed the title into "Does Momentum Change the Implicit Bias on Separable Data?". We also conduct an experiment on MNIST with SGD and SGDM (please refer to Figure 5 in the updated version). It can be observed that the momentum helps SGDM converge faster than SGD, while help SGDM obtain similar test accuracy.
>
> **Q2**: It is shown in Figure 3 of [Soudry et al. 2018] that ADAM does not converge to  max-margin. This seems to contradict the results for ADAM in this paper. Can you comment on that ?
>
> **A2**:
>  Good question and thanks for asking! We re-conduct the experiment in Figure 3, [Soudry et al. 2018] (as we can not find the code of Figure 3 in [Soudry et al. 2018] in their github project), and the results is shown in Appendix E.1.3 in the updated version. After running the experiments across different learning rates and different random seeds (we use the He initialization), we believe the Figure 3 in [Soudry et al. 2018] is not sufficient to support the claim "Adam doesn't converge to the $L^2$ max margin solution". The reason are as follows:
>
> (1). The result in [Figure 3, [Soudry et al. 2018]] is similar to our Figure 3. (f) (learning rate $0.1$ and random seed $1$), in which GD always have a smaller angle (to the max margin solution) before the experiment ends. However, as it can be observed from the amplified figure, the angle of GD keeps still and above $0$, which indicates GD doesn't converge to the max margin solution under this setting, while the angle of Adam to the max margin solution keeps decreasing and it's unreasonable to claim "Adam doesn't converge (to the max margin solution) while GD does" in this case. This is also supported by changing the random seed from $1$ to $2$ in Figure 3. (f), and the angle of GD also keeps still but far above $0$, while angle of Adam keeps decreasing and is smaller than that of GD when the experiment ends. The reason behind this phenomenon is that the dataset is ill-posed (second component is way larger than the first component of every data), and we need to decrease the learning rate.
>
> (2). After decreasing the learning rate, both angle of GD and Adam (to the max margin solution) is decreasing and stays in the same magnitude, which can be observed in Figure 3. (b) and (d), and Figure 4. (b).
>
>
> **Q3**: What is the effect of $\varepsilon$ in Adam.
>
> **A3**: Thanks for asking. The benefit of  $\varepsilon$ in Adam is two-fold: (1). for early time in the training, $\varepsilon$ is small compared to the second moment estimator $\hat{\nu}(t)$ in the adaptor, which accelerate the training; (2). for latter time in the training, gradient becomes small and $\varepsilon$ becomes dominant, which helps the Adam converge to the max margin solution.

---

> > ### Comment · Reviewer_5GXX · 2021-11-24
> > **Response**
> >
> > Thank you for the feedback. I also read the other reviews and decided to keep my score as is.

---

### Official Review · Reviewer_D63g · 2021-10-29

**Correctness:** 4
**Technical Novelty And Significance:** 3
**Empirical Novelty And Significance:** Not applicable
**Recommendation:** 6
**Confidence:** 3

**Main Review:**

The main result of the paper that shows the same behavior in (S)GD and SGDM is particularly interesting. However there is incongruence with the result of [Soudry et al. 2018] where the authors showed that Adam does not converge to the L2 max margin.

The authors need to add "on linearly separable datasets", or some variation, in the title. The title of the paper is extremely misleading and (for what we know) can be false. We do not know whether momentum changes or not the implicit bias in general.
Both, the title and abstract, do not mention that the analysis is limited to linearly separable binary classification problems. This is deceitful and has to be change. (My current recommendation score is low for this reason.)

In page 6 there are a large number of typos in the equations. The authors should revise. In the equations after "of the dynamic as" there are many "t" that should be replaced by "s" (unless I am mistaken, 3 of them are wrong). The final term dw(1)/dt is imprecise and should be replaced by, dw(t)/dt|_{t=1}.
At the end of the section, the summatory has a nabla term should not be there.

I do not have an expertise in this field, therefore it is hard to assess the analysis of the related works nor I can give a comparison with the state of the art. I can observe that citing only 23 papers is strange. For instance, some results that are not reported are: [Neyshabur et al., 2015] that (as far as I know) introduced the concept of implicit bias, [Gunasekar et al 2018] that studied implicit biases for a variety of algorithms, [Mannelli, Urbani 2021] that found a similar conclusion a non-convex task. In general papers in ICLR cite among 40-60 papers, that may be community-dependent, but I would guess that some of the relevant literature have not been cited.

The author should comment on the empirical difference between the speed SGDM with respect to standard SGD. I would be happy to see the result of same simulations. In particular something like figure 3 in [Soudy et al. 2018].

**Summary Of The Paper:**

The authors study the convergence of stochastic gradient descent momentum (SGDM) in a problem with of binary classification with separable data.
Interestingly, the authors observed that SGDM and GD converge to the same solution (i.e. the max margin solution) with the same convergence rate O(1/t).
The paper proposed a variation of [Soudry et al. 2018] solution, that allows for the analysis of the more complex SGDM.

**Summary Of The Review:**

The paper found interesting result that in my opinion are significant. However the paper has two major issues that should be considered before acceptance.

---

> ### Author Response · Authors · 2021-11-22
> **Thanks for your comments!**
>
> We thank the reviewer for instructive comments, and we have revised paper accordingly with color red. The concerns are dually addressed as follows:
>
> **Q1**: The theory result on Adam is inconsistent with the experiment in (Figure 3, [Soudry et al. 2018]).
>
> **A1**: Good question and thanks for asking! We re-conduct the experiment in Figure 3, [Soudry et al. 2018] (as we can not find the code of Figure 3 in [Soudry et al. 2018] in their github project), and the results is shown in Appendix E.1.3 in the updated version. After running the experiments across different learning rates and different random seeds (we use the He initialization), we believe the Figure 3 in [Soudry et al. 2018] is not sufficient to support the claim "Adam doesn't converge to the $L^2$ max margin solution". The reason are as follows:
>
> (1). The result in [Figure 3, [Soudry et al. 2018]] is similar to our Figure 3. (f) (learning rate $0.1$ and random seed $1$), in which GD always have a smaller angle (to the max margin solution) before the experiment ends. However, as it can be observed from the amplified figure, the angle of GD keeps still and above $0$, which indicates GD doesn't converge to the max margin solution under this setting, while the angle of Adam to the max margin solution keeps decreasing and it's unreasonable to claim "Adam doesn't converge (to the max margin solution) while GD does" in this case. This is also supported by changing the random seed from $1$ to $2$ in Figure 3. (f), and the angle of GD also keeps still but far above $0$, while angle of Adam keeps decreasing and is smaller than that of GD when the experiment ends. The reason behind this phenomenon is that the dataset is ill-posed (second component is way larger than the first component of every data), and we need to decrease the learning rate.
>
> (2). After decreasing the learning rate, both angle of GD and Adam (to the max margin solution) is decreasing and stays in the same magnitude, which can be observed in Figure 3. (b) and (d), and Figure 4. (b).
>
> **Q2**: The authors need to add "on linearly separable datasets". The title is misleading.
>
> **A2**: Thanks for the advice. We have added "on linearly separable datasets" in the abstract, and change the title to "DOES MOMENTUM CHANGE THE IMPLICIT BIAS ON
> SEPARABLE DATA?
> " (please refer to the updated version). We hope this adjustment can address your concerns.
>
> **Q3**: In page 6 there are a large number of typos in the equations: (1). three $t$ should be changed to $s$; (2). $dw(1)/dt$ is not precise; (3). There is an additional $\nabla$ term.
>
> **A3**: Thanks for the pointing out these typos. These typos have been fixed in the updated version.
>
> **Q4**: Only 23 papers are cited. Some works are not reported ([Neyshabur et al., 2015],  [Gunasekar et al 2018], [Mannelli, Urbani 2021]).
>
> **A4**: Thanks for the comments. We apologize that in the original version only the most related works are cited. We have added other related works including the ones mentioned by the reviewer.
>
> **Q5**: Additional experiments as Figure 3 in [Soudry et al. 2018].
>
> **A5**: Thanks for the advice and we have added the experiments of SGD and SGDM on the dataset used for Figure 1 in [Soudry et al. 2018] over different settings of learning rates and random seeds (please refer to Figure 2 in the updated version), and the results indicates that the training processes of  SGD and SGDM are similar regardless of the learning rates and random seeds.

---

### Official Review · Reviewer_4RNH · 2021-11-01

**Correctness:** 4
**Technical Novelty And Significance:** 3
**Empirical Novelty And Significance:** 1
**Recommendation:** 5
**Confidence:** 4

**Main Review:**

This paper provides theoretical results for the implicit bias of momentum methods. This topic is interesting and important. The writing overall is good to me, but there are two comments: the authors should (i) cite paper [1] in the Preliminary section since most contents of this section are borrowed from [1]; (ii) formally define "implicit bias", since it would help general readers to easily understand this paper.

Although the authors make a contribution in the theoretical side, the theoretical results are still far from many applications in practice since this paper makes many assumptions such linear model, separable data, and smooth loss. I am wondering if some of assumptions such as separable data can be removed? That is to say, if the analysis can be extended to non-separable data?

Comparing with the learning rates for GD [1] and SGD [2], the learning rates in Theorem 2 (for GDM) and Theorem 3 (for SGDM) are smaller respectively, which allows narrower choices of hyper-parameter tuning for learning rate. Does this mean GD/SGD is better than GDM/SGDM? If so, why do we need momentum?

Although the authors provided the theoretical analysis for the implicit bias of momentum methods, it is strongly recommended and needed that the authors demonstrate their theoretical findings (for example, similar experiments like in [1] and [2]) and compare the momentum methods to other methods such as GD/SGD (for example, different choices of learning rate). The lack of experimental results makes the main contribution of this paper unclear and incomplete.

Reference

[1] The Implicit Bias of Gradient Descent on Separable Data, JMLR 2018

[2] Implicit bias of SGD: Stochastic gradient descent on separable data: Exact convergence with a fixed learning rate, AISTATS 2019

**Summary Of The Paper:**

This paper theoretically studies the implicit bias of momentum methods such as GDM/SGDM and Adam under some commonly-used assumptions and shows that momentum doesn't change the implicit bias. One of the main contributions lies on the new theoretical analysis that constructs new Lyapunov functions.

**Summary Of The Review:**

1. The authors provided new theoretical results for implicit bias of momentum methods.

2. The learning rate of GDM/SGDM is smaller than that of GD/SGD, which allows narrower choices.

3. No numerical results.

---

> ### Author Response · Authors · 2021-11-22
> **Thanks for your comments! This is response part I.**
>
> We thank the reviewer for the comments. The concerns are respectively addressed as follows:
>
> **Q1**:  Cite paper [Soudry et al. 2018] in the Preliminary section.
>
> **A1**: Thanks and we have revised the paper accordingly.
>
> **Q2**: Formally define "implicit bias".
>
> **A2**: The implicit bias, which is the regularization implicitly brought by the optimizer into the output parameters, is  introduced in the first sentence of the introduction section in the original version as "It is widely believed that the optimizers have implicit bias in terms of selecting output parameters among all the local minima on the landscape.". However, defining the implicit bias more formally is difficult, as the form of implicit bias can be different across optimizers ([Qian \& Qian 2019]) and across tasks ([Ali et al. 2020]).
>
> **Q3**: The theoretical results are still far from many applications in practice since this paper makes many assumptions such linear model, separable data, and smooth loss.
>
> **A3**: We respectfully argue that although the theoretical results focus on the linear classification problem with separable data, our results shed light on the analysis of momentum based optimizer on the deep homogeneous neural network with separable data and exponential-tailed loss, as the previous theoretical results of GD and adaptive optimizers in linear classification problems have  already been extended to the deep homogeneous neural networks ([Lyu \& Li 2021] and [Wang et al. 2021]) (it's also detailed discussed in the "Gap Between The Linear Model and Deep Neural Networks." part of Section 7).
>
> Also, the assumption of exponential-tailed loss is actually mild, as the cross-entropy loss, as a special case of the exponential-tailed loss, is widely adopted in classification problems.
>
> **Q4**: Can the separable data assumption be removed?
>
> **A4**: Good question and thanks for asking. Separable data is a more challenging setting to analyze because it means the linear model is overparameterized and there are infinite global minima.
> If the data is no longer separable, the empirical risk is then strongly convex on any compact set and has a finite global minimum by [Ji et al., 2019]. On the other hand, if the data is separable, which is analyzed in this paper, there is no finite minimum and the parameter will diverge to infinity. Therefore, the result for the separable data no longer holds for the non-separable data.
>
> **Q5**: The learning rate required by the theorems are narrower than their without momentum counterpart. Does this mean GD and SGD is better than GDM and SGDM?
>
> **A5**: No. The narrower learning rate range is due to the requirement of theoretical analysis to deal with the additional momentum term. The required learning rate for GD is smaller than $\frac{2}{H_{S_0}}$, while for GDM the learning rate need to be smaller than $\frac{2(1-\beta)}{H_{S_0}}$, which comes naturally as by the following intuition:
>
> The momentum at step $t$ is calculated as $m(t)=\nabla \mathcal{L}(w(t))+\beta m(t-1)$, and if $\nabla \mathcal{L}(w(t))$ is a constant vector $a$ independent of $t$, $m(t)$ will have the form $\frac{1-\beta^t}{1-\beta}a$, which has the limit $\frac{1}{1-\beta}a=\frac{1}{1-\beta}\nabla \mathcal{L}(w(t))$ and using GDM in this case is equivalent to scale up the update in GD by $\frac{1}{1-\beta}$. Therefore, to get the same theoretical guarantee, we need to scale down the learning rate $\eta$ by $1-\beta$.
>
> Lower learning rate in the analysis of momentum based method is also common in existing literature (e.g., [Ghadimi et al. 2015] and [Sun et al. 2019]). We conduct the experiments of GD and GDM on the same dataset as [Soudry et al. 2018], and it turns out the behavior of GD and GDM are quite similar regardless of the learning rate.
>
> As for the benefit of momentum, although both the theory (this paper) and experiment ([Soudry et al. 2018] and [Nacson et al. 2019]) indicate the (S)GDM behaves quite similar to the (S)GD in the linear classification problem, we conjecture there is two-fold benefit for deep homogeneous neural networks: (1).
> at the early training stage, the momentum term can accelerate the training, which is beyond the scope of this paper; (2). at the later training stage (especially, after neural networks achieve 100\% training accuracy), the momentum based optimizers still drive the neural network to the solution with good generalization guarantee as the ones without momentum, as existing literature ([Lyu \& Li 2019, Wang et al. 2021]) indicates the conclusion of linear model can be extended to deep homogeneous neural networks in this stage.  To demonstrate this effect empirically, we conduct an experiment on MNIST with SGD and SGDM (please refer to Figure 5 in the updated version). It can be observed that the momentum helps SGDM converge faster than SGD, while help SGDM obtain similar test accuracy.

---

> ### Author Response · Authors · 2021-11-22
> **This is response part II.**
>
> **Q6**: Experiments.
>
> **A6**: Thanks for the advice! We will add the experiments to demonstrate the similarity between (S)GD and (S)GDM with the same learning rate, to support our theoretical result of Adam, and to show SGD and SGDM behave similarly in terms of test accuracy in deep neural networks. Please refer to Appendix E for details.
>
> **References**
>
> Ghadimi et al., Global convergence of
> the heavy-ball method for convex optimization. In 2015 European control conference (ECC), pp.
> 310–315. IEEE, 2015.
>
> Soudry et al., "The implicit bias of gradient descent on separable data." The Journal of Machine Learning Research 19.1 (2018): 2822-2878.
>
> Ji et al., "The implicit bias of gradient descent on nonseparable data." Conference on Learning Theory. PMLR, 2019.
>
> Sun et al., Non-ergodic
> convergence analysis of heavy-ball algorithms. In Proceedings of the AAAI Conference on
> Artificial Intelligence, volume 33, pp. 5033–5040, 2019.
>
> Lyu \& Li. "Gradient Descent Maximizes the Margin of Homogeneous Neural Networks." International Conference on Learning Representations. 2020.
>
> Wang et al., "The implicit bias for adaptive optimization algorithms on homogeneous neural networks." International Conference on Machine Learning. PMLR, 2021.

---

> ### Author Response · Authors · 2021-11-30
> **Look forward to your response!**
>
> Dear Reviewer 4RNH,
>
> Corresponding to your review, we have posted our replies and revised the paper accordingly. Does our replies address your concerns, or are there other aspects that you think would help improve the paper? Thanks!
>
> Best,
> Paper 926 Authors

---

### Official Review · Reviewer_g2J5 · 2021-11-03

**Correctness:** 4
**Technical Novelty And Significance:** 2
**Empirical Novelty And Significance:** Not applicable
**Recommendation:** 5
**Confidence:** 4

**Main Review:**

The eventual goal of this manuscript (analyzing Adam) is interesting and of importance to some degree. However, the current manuscript seems still a bit far from the final goal as only a deterministic version is analyzed, but such an algorithm is quite meaningless as a deterministic version of Adam is never being used, and using the same name Adam has the suspicion of misleading readers that the stochastic algorithm (as Adam has always been a stochastic algorithm from the beginning, which is also indicated in the title of the original paper of it) is analyzed. Moreover, there are many errors, typos, and undefined symbols in the proof, although most are notational issues that should be easily fixable, I also found that there are crucial errors that invalidate the proofs. In general, viewing from the messiness of the proofs, I would not consider the current manuscript finished, and would like to urge the authors to at least have a thorough check of all their contents before any submission, which should be a minimum common practice.

My major comments are as follows:
1. The whole picture of the manuscript seems to be quite unfocused. If the authors are able to include the analysis of Adam, then surely this is a self-contained story. However, without this piece in the puzzle, the description flow becomes quite divergent and more like just putting all known partial results together to form a paper. Instead, if the manuscript went with either GDM plus SGDM, or GDM plus Eq. (2), the story will look more like a cohesive one.

2. Throughout the manuscript, the authors referred Eq. (2) as Adam. This is very misleading, as Adam is a stochastic algorithm, while (2) is a deterministic algorithm. The analysis of this algorithm is quite meaningless to me, as it is not being used anywhere (although it is analyzed in Wang et al (2021), being analyzed in a theoretical paper only doesn't justify the usefulness of this algorithm). To me, this part of analysis is quite pointless unless it really can lead to any insight for analyzing the original Adam, but this is surely doubtful.
The naming also gives readers the illusion that the real Adam is analyzed and thus falsely boosted the contributions of the manuscript. This obvious limitation is not mentioned at all in the manuscript as well, which is not quite meritorious. If the authors would like to go with the current form, such an incompleteness of their partial results should at least be mentioned in the conclusions.

3. I also found that the proofs contain more than acceptable typos, undefined symbols, and small or huge errors everywhere. I'll list the major ones here and leave the fixable ones to the minor comments.
A. Definition 2:the vector \tilde w is undefined, and so is gamma. The authors also seem to assume gamma = 1 sometimes, but to drop such an assumption sometimes throughout the analysis.
B. Lemma 12: The lines between (15) and (16) are erroneous. The correctness of first inequality (the one before (*)) is unclear to me.
C. Lemma 14: In case 1, the last line of (22) doesn't hold. In particular, on the right-hand side you have O(alpha^2)\|w(t^*+1) - w(t^*)\|^2 (not o(alpha^2) as claimed), while on the right-hand side, the coefficient should actually be O(alpha^2) (instead of small-o) viewing from the previous line. This one is fatal as the same argument is also used in the analysis of SGDM.

Minor comments:
1. The authors are overloading the same symbol for multiple meanings in multiple places, and this is really confusing. For example, c is used in Assumption 2, Lemma 1, and Corollary 2, but they are for different things. There are also inconsistent notations, like sometimes w(1) and sometimes w_0 indicates the initial point.
2. Assumption 2: \ell' > 0 should be \ell' < 0.
3. Lemma 8: \tilde S_s is undefined.
4. Corollary 2: \hat w should be \tilde w.
5. Lemma 10: x_0 should be s_0.
6. Lemma 14: The equality after (**) should be \leq. The next line it should be O(\alpha^2) instead of small-o. The line after (23) has a reference to (by Lemma) without a number.
In case 2: (\dot) should be \leq instead of =. In its previous line, alpha should be alpha^*.
7. Corollary 1: the 2nd equality is wrong. Note that \| \hat w\| = \gamma. All later terms should have \gamma^2 cancelled out.
8. Corollary 4: there are 2 consecutive \infty in the first equation.
9. Lemma 15: we will prove \|r(t)\| is bounded by reduction to absurdity should be by contradiction. In the sequel, a sentence assuming r(t) is unbounded is needed.
10. Lemma 16: The line before (30) is wrong. t^{-1} - log((t+1)/t) = O(t^{-2}) by Taylor's expansion. The equality after (30) wrongly added 1/N to the term of \hat w and led to wrong expressions in all consequential equalities. The last line of this set of equalities missed 1/N for the second inner product.
11. Lemma 3: The equality of (35) should be \leq. The line before (37) misses a \beta^2 in the last term (although (37) is still correct).
12. Lemma 17: In the inequalities after "On the other hand..." , the 3rd line should be using \hat w. The penultimate inequality in the following set of inequalities doesn't seem to be directly interchangeable because the inner one involves the product of 2 sums, but fortunately the final conclusion is still right.
13. Lemma 18: In the equality set before defining A_6 and A_7, 1/b is missing starting from the 3rd line, and \tilde w(t) should be \tilde w. The penultimate line on p. 31 should be 1 - \exp(...). (*) should be \leq instead of =. On the next page, the fact of e^{-x) x \leq e^{-1} is used much earlier at the page top in another set of inequalities, but this is only explained in a later place for another set of inequalities. For the case of r(t)^T \tilde x_i \geq 0, the 2nd inequality should be equality. Here throughout \tilde w(t) should be \tilde w. The line explaining Eq. (\dotp) should be 0.5 - \epsilon instead of 0.5+\epsilon. In the set of inequalities after the line "Specifically, if...", the exponent of t on the right-hand side of the ineq should include min(\mu_-,0.5) instead of just \mu_-. The explanation for the \square doesn't quite make sense to me: -(r(t)+n(t))^T \tilde x_i doesn't seem to approach 0. In the following inequalities, starting from the 4th line, the term of \exp(-(r(t)+n(t))^T \tilde x_i) becomes missing. In the 6th line of the same set of inequalities, +n(t)^T \tilde x_i should be - \mu_- n(t)^T \tilde x_i. I don't see why adding 4  in the exponent of this set of inequalities is needed, seems like the result still holds without this 4.



**Summary Of The Paper:**

This manuscript attempts to analyze the asymptotic behavior of heavy-ball method, stochastic gradient with momentum, and an algorithm that combines the heavy-ball method with a coordinate-wise preconditioning similar to that used in Adagrad and Adam for a class of problems that are essentially logistic regression without any regularization, under the assumption that the provided data set is linearly separable.

**Summary Of The Review:**

The manuscript claimed to provide an analysis for Adam but actually didn't and only analyzed a much simplified (non-stochastic) algorithm that is not being used anywhere. The proofs contain many errors and some are vital.

I'd like to thank the authors for a detailed explanation and revising the manuscript.
After the authors' response and revision, I think the proof of lemma 14 is now correct. However, I still think that the analysis of the algorithm described in eq (2) is not very meaningful, as it looks to me more like inventing an algorithm that is not used anywhere and deriving some (however novel or difficult) analysis for it.

---

> ### Author Response · Authors · 2021-11-17
> **Thanks for your comments! Concerns about the proof are addressed here.**
>
> We would like to sincerely thank the reviewer for your careful review and instructive comments. The typos in the paper are all fixed with color red. After reading the review, we think your biggest concern is on the "crucial errors that invalidate the proofs", and we would like to answer it at first. we believe that there are some misunderstandings and we illustrate them as follows:
>
> **Q (a):** Definition 2: the vector $\tilde{w}$ is undefined, and so is gamma. The authors also seem to assume $\gamma=1$ sometimes, but to drop such an assumption sometimes throughout the analysis.
>
> **A (a):**  (1) The $\tilde{w}$ should be $\hat{w}$ instead in Definition 2. Thank you for pointing out this typo. The support vector is defined as "Correspondingly, $\tilde{x}_i$ is called a non-support vector if $ \langle \tilde{x}_i, \hat{w} \rangle>1$". (2) The margin $\gamma$ has been defined in Definition 1, which is the inverse of the norm of $\hat{w}$. (3) We don't assume that $\gamma=1$ in the paper, so could you please specify where we drop the $\gamma$?
>
> **Q (b):** Lemma 12: The lines between (15) and (16) are erroneous. The correctness of first inequality (the one before (15)) is unclear to me.
>
> **A (b):** The inequality is correct, which can be derived as follows: the term $\frac{1}{Nb} \left(\sum_{\tilde{x}\in \mathcal{T}(S)}  \ell' (\langle w,\tilde{x}\rangle) \right)^2$ equals to
> \begin{equation*}
>     \frac{1}{Nb} \sum_{\tilde{x}\in \mathcal{T}(S)}  \ell' (\langle w,\tilde{x}\rangle) ^2+  \frac{1}{Nb} \sum_{\tilde{x},\tilde{x}'\in \mathcal{T}(S),\tilde{x}\ne\tilde{x}'}  \ell' (\langle w,\tilde{x}\rangle) \ell' (\langle w,\tilde{x}'\rangle)
> \end{equation*}
> by expansion. Also, by $\ell'<0$ and $\frac{b-1}{N-1}\le 1$, we have
> \begin{equation*}
>      \frac{1}{Nb} \sum_{\tilde{x},\tilde{x}'\in \mathcal{T}(S),\tilde{x}\ne\tilde{x}'}  \ell' (\langle w,\tilde{x}\rangle) \ell' (\langle w,\tilde{x}'\rangle)\ge \frac{b-1}{N(N-1)b} \sum_{\tilde{x},\tilde{x}'\in \mathcal{T}(S),\tilde{x}\ne\tilde{x}'}  \ell' (\langle w,\tilde{x}\rangle) \ell' (\langle w,\tilde{x}'\rangle),
> \end{equation*}
> which leads to the inequality.
>
> **Q (c):** Lemma 14: In case 1, the last line of Eq. (22) doesn't hold.
>
> **A (c):** There was a typo ($\boldsymbol{o}(\alpha^2)$ should be $\boldsymbol{o}(\alpha)$ in Eq. (22), thanks for pointing it out), but this doesn't influence the correctness of this inequality. {We have made a slight adjustment on Eqn.(22) to make it more clear.} Specifically, to prove the Inequality $(\diamond)$ in Eq. (22), it suffices to show the term $-\frac{(1-\alpha)\beta}{2\eta}\Vert (w(t^*)-w(t^*-1)\Vert^2+\frac{\alpha\beta}{2\eta}\Vert w(t^*+1)-w(t^*)\Vert^2
>     +\left(\frac{H_{s_0}\alpha^2}{2}-\frac{\alpha}{\eta}\right)\Vert w(t^*+1)-w(t^*)\Vert^2$ in Eq. (22) (in the updated version) to be smaller than $ -\frac{\beta}{2\eta}\alpha^2 \Vert w(t^*+1)-w(t^*)\Vert^2
>     -\frac{(1-\beta)(1-{ C_1})\alpha^2}{\eta}\Vert w(t^*+1)-w(t^*)\Vert^2$ in Eq. (22) (in the updated version). However, the former term has a negative limit as $\alpha\rightarrow 0$, while the latter term has a zero limit as $\alpha\rightarrow 0$. Therefore, this inequality holds when $\alpha$ is small enough.
>
> Also, we would like to mention that the proof of the theorem for the SGDM case doesn't follow this routine, as in the SGDM case we don't use the interpolation in non-integer time $w(t+\alpha)$ ($\alpha\in (0,1)$).

---

> > ### Comment · Reviewer_g2J5 · 2021-11-29
> > **Thanks for the response**
> >
> > I'd like to thank the authors for their detailed response and revision to handle the technical issues in the proof.
> > The proofs are now mostly ok, although the part between (15) and (16) still remains unfixed.
> > The authors provided explanations for that, but the text of
> > "where Eq. (*) is due to $\forall z \\in  \\langle \\tilde x, - \\hat w \\rangle \\geq 1$" still looks typos to me:
> > why is there a $z$ here and what does that sentence even mean?
> >
> > I agree with all the arguments about the difficulty in analyzing Adam, but those anyway don't make section meaningful to me. This remains my major concern, so my recommendation is (weak) rejection.

---

> > > ### Author Response · Authors · 2021-11-30
> > > **Clarification on the result of Adam**
> > >
> > > We thank the reviewer for the further response. Here are our corresponding replies:
> > >
> > > **Q1**: The text of "where Eq. (*) is due to "$\forall z\in \langle \tilde{x},-\hat{w}\rangle\ge 1$" still looks typos to me.
> > >
> > > **A1**: This is indeed a typo and thanks for pointing it out. The correct version should be "$\forall \tilde{x}\in \mathcal{T}(S), \langle \tilde{x},-\hat{w}\rangle\ge 1$". We have revised the paper accordingly which can be seen in the following link: https://www.dropbox.com/s/rerhrbmdh6yaxvy/Momentum_Doesn_t_Change_The_Implicit_Bias%20%281%29.pdf?dl=0
> > >
> > > **Q2**: The section of deterministic Adam is still meaningless.
> > >
> > > **A2**: We respectfully argue that the analysis of deterministic Adam is meaningful as it sheds light on the analysis towards stochastic Adam, just as the analysis of GD [Sourdry et al. 2018] laid the foundation of SGD [Nacson et al. 2018]. The theoretical analysis of momentum+stochastic is very hard, while this work for Adam, at least, solves one part of the two.
> > >
> > > **References**
> > >
> > > [Soudry et al. 2018] The Implicit Bias of Gradient Descent on Separable Data
> > >
> > > [Nacson et al. 2018] Stochastic Gradient Descent on Separable Data: Exact Convergence with a Fixed Learning Rate

---

> ### Author Response · Authors · 2021-11-17
> **Other concerns are addressed here.**
>
> **Q1:** The eventual goal of this manuscript (analyzing Adam) is interesting and of importance to some degree. However, the paper only analyzes the deterministic version of Adam, which is not used in practice.
>
> **A1:** We believe  that there is misunderstanding on the purpose of this paper. The  main purpose of this paper is to investigate the  influence of "Momentum" on the implicit bias of the momentum-based optimizers, which has not been well studied in current literature. Along this line, we study momentum starting from GDm, and we further study the combination of momentum and stochastic sampling and the combination of momentum and preconditioning.
>  We never claim that our eventual goal is to analyze Adam, and the results for GDm and SGDm are also novel (the first of this kind) and are as important as the Adam analysis.  However, we agree that  it will be better to distinguish the deterministic and stochastic version of Adam,
> and we have changed the description from "Adam" to "Adam (w/s)" (the deterministic version) in  the updated paper.
>
>  Moreover, we'd like to discuss the technical challenges in analyzing  stochastic version of Adam, which includes momentum, stochastic sampling and preconditioning.
>
> The influence of the preconditioning in stochastic Adam makes it no longer able to be viewed as a variant of the heavy-ball algorithm,  which makes its stochastic Adam can not be analyzed in the proposed framework. For the stochastic algorithm which is not belong to the family of heavy-ball algorithm, its theoretical foundation is still an open problem and there is no existing theoretical framework constructed for it.
> We will also include this discussion into the paper and study this setting in future work.
>
> **Q2:** The current paper seems to be without focus. Instead, if the manuscript went with either GDM plus SGDM, or GDM plus Eq. (2), the story will look more like a cohesive one.
>
> **A2:** Thanks for the suggestion on the organization of our paper. As discussed in **A1**, we  study momentum starting from GDm, and we further study the combination of momentum and stochastic sampling and the combination of momentum and preconditioning. Moreover, we will add the discussion of "momentum + stochastic sampling + preconditioning" (i.e., stochastic adam)
> in Section 6 together with the technical difficulty to analyze the stochastic Adam. We hope  this adjustment can address your concerns on the organization.
>
>
>
> **Minor Issues:**
>
> 1. :Thanks for pointing out. We have changed $c$ in Lemma 1 to $C_1$, and $c$ in Corollary 2 to $C_2$. The initialization point is $w(1)$ and there is a misunderstanding: we use $w(0)=w(1)$ as an auxiliary point to provide a unified update rule for GDM and SGDM (equivalent to $m(0)=0$) as the update rule at $t=1$ to get $w(2)$ for GDM and SGDM needs the information at $t=0$. Also, the $w_0$ in Lemma 10 is not $w(0)$, it can be any point in the parameter space.
>
> 3.: $S_s$ is the support set of $S$ and is defined in Definition 2.
>
> 6.:
> Here Eq. ($\bullet$) in case 2 should be equality, as it directly applies the definition of  $C_1$ (in the updated version): $H_{s_0}=2\frac{1-\beta}{\eta}C_1$. Other typos are revised accordingly.
>
> 7.: The conclusion of Corollary 1 is correct. There is a typo in the second equality in the proof of Corollary 1 and a factor $\frac{1}{\Vert \hat{w}\Vert^2}$ was dropped. The proof is correct after adding the $\frac{1}{\Vert \hat{w}\Vert^2}$ back.
>
> 10.: Here is a typo and the first equality wrongly adds $\frac{1}{N}$. However, the following equalities are correct as $\tilde{w}$ is defined to satisfy $\frac{\eta}{(1-\beta)N} e^{-\langle \tilde{x}_i,\tilde{w}\rangle}=v_i$.
>
> 11.: The line before (37) is correct. The reason is that $$\mathbb{E}\_{\mathcal{F}\_{s+1}}\left\Vert  \nabla \mathcal{L}\_{B(s)}(w(s))\right\Vert^2\le \frac{1}{\gamma^2}\mathbb{E}\_{\mathcal{F}\_{s}}\left\Vert  \nabla \mathcal{L}(w(s))\right\Vert^2$$ by Lemma 12.
>
> 13.:
> (a). For the case of $r(t)^T \tilde x_i \geq 0$, the 2nd inequality shouldn't be equality, as here we use the condition $\langle r(t),\tilde{x}_i\rangle\ge 0$.
>
> (b). Here the equation in the explanation of Eq, ($\square$) is under the condition " $0>\langle r(t), \tilde{x}\_i\rangle\ge -t^{-0.5\min(\mu\_{-},0.5)}$", and therefore $-(r(t)+n(t))^T \tilde{x}_i$ approaches 0. We understand this part is a bit confusing, and restate the reason in the paper (please refer to the current version for details).
>
> (c). $4$ comes from the estimation $ \langle r(t)+n(t),\tilde{x}_i\rangle\ge -4$ and thus $e^{-\langle r(t)+n(t),\tilde{x}\_i\rangle}\le e^{4\mu\_{-}}$ (which leads to the first inequality after "If $-2\le\cdots$").
>
>
> 2., 4., 5., 8., 9., 12.: Thanks for the detailed review and we have modified the paper accordingly!

---

> ### Author Response · Authors · 2021-11-29
> **Does our response address your concern?**
>
> Dear Reviewer g2J5,
>
> We have posted our responses and revised the paper accordingly. Does our response address your concerns, or are there other aspects that you think would help improve the paper? Thanks and look forward to your response!
>
> Best,
> Paper 926 Authors

---

### Author Response · Authors · 2021-11-23
**Revision Summary**

Dear ICLR 2022 PCs, ICLR Paper 926 SACs, ICLR Paper 926 ACs, and ICLR Paper 926 Reviewers,

Thank you for organizing the ICLR 2022 and handling our paper. During the second stage of discussion, we have revised the paper based on the comments of the reviewers, and the revision is summarized as follows:

(1). We add numerical experiments both on the linear model with synthetic data and on the deep neural network with MNIST. These experiements support our theoretical results.

(2). We change the title into "DOES MOMENTUM CHANGE THE IMPLICIT BIAS ON SEPARABLE DATA?", and highlight our analysis is on the linear model with separable data in the abstract.

(3). We add discussions on the difficulty of analyzing stochastic Adam, while highlight the Adam analyzed in our paper is without stochasticity (we change the name from "Adam" to "Adam (without stochasticity)" in the paper).

(4). The typos in the main body of the paper and in the appendix are fixed accordingly.

(5). By taking the suggestions from Reviewer D63g, additional papers are cited in the related works.

---

### Decision · Program_Chairs · 2022-01-20

**Decision:**

Reject

**Comment:**

Following a recent line of work on the implicit bias of learning algorithms, the authors consider optimization methods that incorporate momentum. The reviewers found the topic timely and interesting, and generally appreciated the novelty of the technical contributions in the work. However, several critical issues concerning the presentation quality and the positioning of the paper have been raised. In addition, some of the reviewers felt that parts of the paper were somewhat rushed and potentially misleading (mainly, those concerning deterministic\stochastic ADAM and the complexity of the models considered in the paper), and others believed that the experimental section should be made more solid to properly corroborate the theoretical analysis provided in the paper. The authors are encouraged to incorporate the instructive feedback provided by the reviewers in future revisions of the paper.